# 🌀 DEEPSCIENTIST: ADVANCING FRONTIER-PUSHING SCIENTIFIC FINDINGS PROGRESSIVELY

**Yixuan Weng**[1][*]**Minjun Zhu**[2,1][*]**Qiujie Xie**[2,1]**, Qiyao Sun**[1]**, Zhen Lin**[1]**, Sifan Liu**[1]**, Yue Zhang**[1][†]
[1]School of Engineering, Westlake University    [2]Zhejiang University
wengsyx@gmail.com, {zhuminjun, zhangyue}@westlake.edu.cn
**Project:** https://deepscientist.cc
**Code&Logs:** https://github.com/ResearAI/DeepScientist

## ABSTRACT

While previous AI Scientist systems can generate novel findings, they often lack the focus to produce scientifically valuable contributions that address pressing human-defined challenges. We introduce DeepScientist, a system designed to overcome this by conducting goal-oriented, fully autonomous scientific discovery over month-long timelines. It formalizes discovery as a Bayesian Optimization problem, using a cumulative Findings Memory to intelligently balance the exploitation of promising avenues with the exploration of novel hypotheses. Consuming over 20,000 GPU hours, the system generated about 5,000 unique ideas and experimentally validated approximately 1100, ultimately surpassing human-designed 2025 state-of-the-art (SOTA) methods on three frontier AI tasks by 183.7%, 1.9%, and 7.9%. Crucially, this was achieved by autonomously redesigning core methodologies, not merely recombining existing techniques. In a striking demonstration, the system achieved progress on AI text detection in just two weeks that is comparable to three years of cumulative human research. This work provides the first large-scale evidence of an AI achieving discoveries that progressively surpass human SOTA on scientific tasks, producing valuable findings that genuinely push the frontier forward.

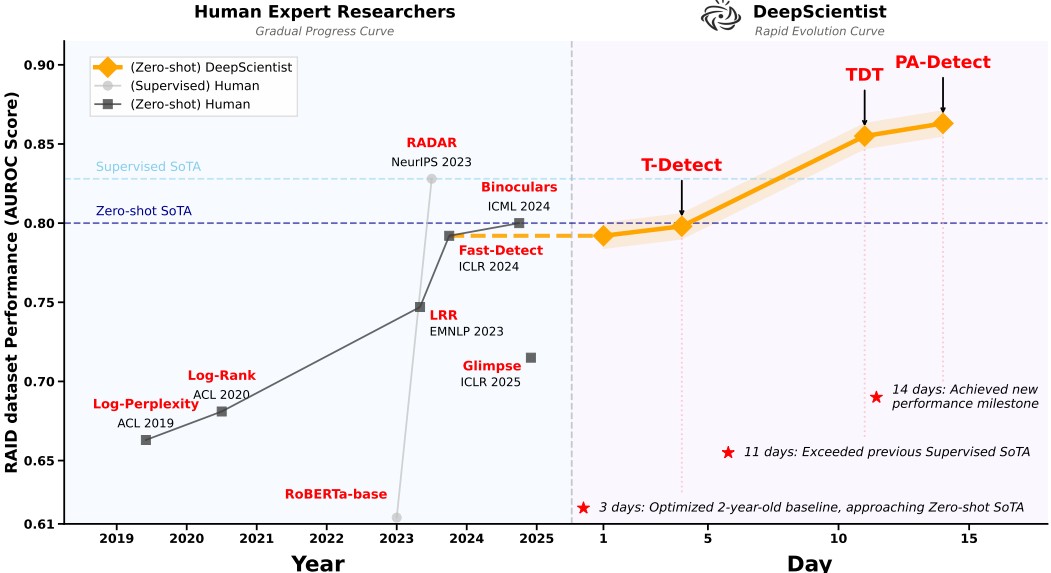

Figure 1: Research timelines for AI text detection on RAID. **Left:** Three years of human research . **Right:** DeepScientist achieves comparable progress in just two weeks. Moreover, all DeepScientist methods demonstrate higher throughput than the previous SOTA, Binoculars.

---

[*]Equal contribution.
[†]Correspondence to: Yue Zhang (zhangyue@westlake.edu.cn)

# 1 INTRODUCTION

Scientific discovery is inherently a process of **continuous exploration** and **trial-and-error**, where vast amounts of time and effort are invested to push the boundaries of human knowledge forward by a small step. This principle of persistent, incremental advancement is visible across the history of technology. For example, the decades-long optimization of semiconductor manufacturing has seen the feature size of transistors systematically reduced from micrometers to single-digit nanometers (Moore, 1965). Similarly, the efficiency of photovoltaic cells has been continuously advanced over half a century, with myriad material and architectural innovations pushing conversion rates from nascent single-digit percentages ever closer to their theoretical limits (Green, 1993). These historical trajectories underscore a process where human scientists engage in decades of **goal-directed**, **iterative** work to advance the SoTA artifacts continuously. In this work, we ask whether an AI-driven system can participate in such long-horizon, goal-directed scientific progress on modern tasks, and how its behavior compares to that of human scientists.

Recently, the emergence of Large Language Models (LLMs) has propelled **automated scientific discovery**, where LLM-based AI Scientist systems take the lead in exploration (Xie et al., 2025). With their powerful capacity for long-form generation and comprehension, LLMs enable **end-to-end, full-cycle automation in scientific discovery.** This has inspired influential work such as AI SCIENTIST-V2 (Yamada et al., 2025), whose scientific artifacts have been published in top-tier conference workshops. However, in the absence of clearly defined scientific goals, current AI Scientist systems often fall into the trap of blindly recombining existing knowledge and methods. As a result, their research outputs frequently appear naive under human evaluation and lack genuine scientific value (Zhu et al., 2025b). **AI Scientists are yet to solve human challenges.**

To solve real-world challenges, we formally model the full cycle of scientific discovery as a **goal-driven Bayesian Optimization problem**, where the primary objective is to discover methods that reliably improve a given evaluation metric over a strong human-designed baseline under a fixed compute budget. Building on this formulation, we introduce **DeepScientist**, an LLM-based agent system explicitly designed to operate on modern, resource-intensive AI research problems, rather than on small-scale symbolic or synthetic tasks. Architecturally, DeepScientist departs from the common "one-shot pipeline" or "single-idea infinite trial-and-error" paradigm by implementing a three-stage iterative workflow (hypothesis generation, implementation & evaluation, and analysis & abstraction) that is tightly coupled with a persistent *Findings Memory* accumulating both successful and failed attempts over month-long runs. Within this framework, a Bayesian surrogate model and acquisition function reason over thousands of past experiments to select the next hypotheses to test, allowing the system to intelligently balance **exploitation** (deepening investigations into promising high-value directions) with **exploration** (venturing into under-explored areas to acquire new knowledge). Through large-scale parallel exploration informed by this Bayesian-optimization perspective and shared memory, DeepScientist can generate innovative hypotheses and ultimately yield both valuable new methods and validation-proven scientific findings.

We select three frontier scientific tasks (*Agent Failure Attribution*, *LLM Inference Acceleration*, and *AI Text Detection* ), take their state-of-the-art methods (ICML 2025 Spotlight, ACL 2025 Outstanding, ICLR 2024) as starting points, and ask DeepScientist to conduct continuous research. As shown in Figures 1 and 3, within a month-long cycle of exploration, validation, and iteration on 16 H800 GPUs, **DeepScientist exceeds their respective human SOTA methods by 183.7% (Accuracy), 1.9% (Tokens/second), and 7.9% (AUROC) through autonomously redesigning core methodologies, rather than simply combining existing techniques** (Section 4.1). To understand how such progress emerged, we analyze DeepScientist's discovery logs, and formed a small program committee to review the generated papers (Section 4.2). These logs show that the system generated over 5,000 unique ideas, of which only 1,100 are selected for experimental validation, and just 21 ultimately lead to scientific innovations (Section 4.3). Moreover, through the scaling experiment on computational resource, we discover a near-linear relationship between the resources allocated and the output of valuable scientific discoveries.

To our knowledge, this is **the first large-scale empirical demonstration of an automated system that continuously advances scientific frontiers on complex AI tasks**, rivaling human researchers under comparable compute budgets. However, these results also highlight a critical reality: AI's exploratory capacity is immense, but genuine success is scarce. Consequently, effective validation,

filtering, and the strategic reuse of failed attempts have emerged as the new bottleneck for automated science. The core question for the field is no longer whether AI can innovate, but how to efficiently guide its powerful yet dissipative exploratory process. We hope our insights, alongside the released logs and codebase, will inspire the community to build more reliable AI Scientist systems, accelerating scientific discovery at scale.

## 2 RELATED WORK

**Replication and Optimization.** A significant body of research focuses on engineering tasks that operate within established scientific frameworks. This includes replication-oriented works like PaperBench (Starace et al., 2025) and Paper2Agent (Miao et al., 2025), which aim to reproduce existing papers. Other works, such as Agent Laboratory (Schmidgall et al., 2025b) and MLE-Bench (Chan et al., 2024), tackle early-stage machine learning engineering problems. Similarly, systems like AlphaTensor (Fawzi et al., 2022), ASI-Arch (Liu et al., 2025) and AlphaEvolve (Novikov et al., 2025) use massive trial-and-error with known engineering methods to improve the performance of codebases (Romera-Paredes et al., 2024; Shojaee et al.). The common goal of these efforts is engineering-driven optimization within an established scientific paradigm, enhancing existing systems without questioning their foundational assumptions. DeepScientist, in contrast, pursues scientific discovery by explicitly targeting the core limitations of strong methods on modern AI tasks: its objective is not merely to refine existing implementations, but to propose and validate new methodological directions that establish improved SOTA performance.

**Semi-Automated Scientific Assistance.** The path toward automating scientific discovery begin not with replacing the scientist, but with assisting them, leading to the development of a paradigm of specialized AI tools for individual research tasks. Systems like CycleResearcher (Weng et al., 2025) handle writing, DeepReview (Zhu et al., 2025a) manages reviewing, and co-scientists (Gottweis et al., 2025; Penadés et al., 2025; Swanson et al., 2025; Baek et al., 2025) aid in hypothesis generation. These powerful tools address only isolated fragments of the scientific process, leaving the crucial loop of learning from failure and exploration to humans. In contrast, DeepScientist is an autonomous agent of inquiry, managing the entire end-to-end research cycle and closing the loop by learning from its own experiments and self-directing its research path.

**Automated Scientific Discovery.** Building on the capabilities of specialized assistants, a line of research pursues full research automation (Xie et al., 2025). Pioneering efforts, such as the AI Scientist systems (Lu et al., 2024; Yamada et al., 2025) and subsequent work (Intology, 2025; Jiabin et al., 2025; Miyai et al., 2025), successfully demonstrate that an AI system can manage the full research cycle and produce novel findings. However, these systems are typically evaluated on relatively synthetic or narrowly scoped problems, and their exploratory strategies are not anchored to clearly specified scientific goals or strong human baselines. This can lead to undirected discoveries that, while novel, are often perceived as having limited scientific value in practice. In contrast, DeepScientist is designed to operate on modern, high-cost AI tasks with competitive human SOTA methods and to treat discovery as a goal-driven Bayesian optimization problem over these baselines. Its exploration is explicitly tied to identified limitations of the human SOTA—using failure attribution and a persistent *Findings Memory* to prioritize hypotheses that are both novel and measurably impactful.

## 3 DEEPSCIENTIST: A PROGRESSIVE SYSTEM FOR DISCOVERING SOTA-SURPASSING FINDINGS

DeepScientist is an LLM-based multi-agent system equipped with an **open-knowledge system**, a continuously accumulating **Findings Memory**, which is composed of both frontier human knowledge (e.g. papers and codes) and the system's own historical findings. This memory is constructed and updated fully automatically during the system's operation, without any manual editing, and each record stores the hypothesis, implementation details, evaluation metrics, and logs of both successful and failed experiments (Zhou et al., 2025). This memory intelligently guides subsequent explorations, ensuring a sustained and focused push at the scientific frontier. The entire system's core task is to find the optimal program $I^*$ from a space of all possible candidate research programs $\mathcal{I}$

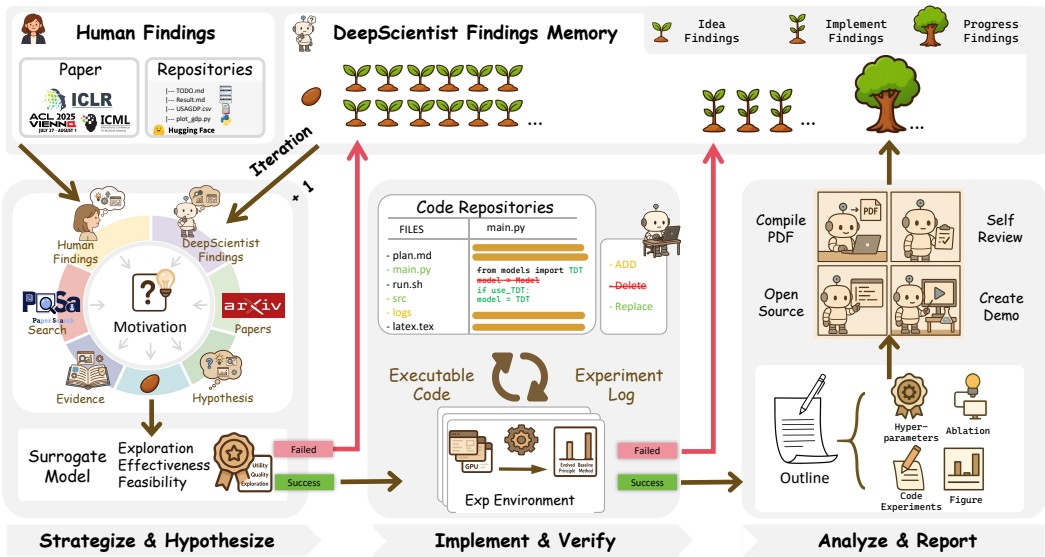

Figure 2: The autonomous, closed-loop discovery process of DeepScientist. The system iterates through a three-stage cycle, learning from both human knowledge and its own experiments.

that maximizes an unknown and extremely expensive-to-evaluate true scientific value function $f(\cdot)$. The architecture of DeepScientist is detailed in Appendix D.

**Exploration Strategy.** Scientific discovery differs from previously considered tasks like early-stage machine learning (Schmidgall et al., 2025a), algorithm discovery (Novikov et al., 2025), or scientific software development (Aygün et al., 2025). Each exploratory step within it requires immense resources. For instance, solving a frontier LLM problem requires approximately $1 \times 10^{16}$ FLOPs for each implementation (Figure 4.c). This necessitates an efficient exploration strategy rather than brute-force search. Compared to prior AI Scientist systems that either follow a single one-shot "idea → experiment → paper" pipeline or perform near-unlimited trial-and-error around a single idea, DeepScientist adopts an explicit iterative loop that combines Bayesian value estimation with a persistent *Findings Memory*. To address this, DeepScientist's discovery process is structured as an iterative Bayesian Optimization loop (Figure 2), through the following **three-stage** cycle:

**Stage I: Strategize & Hypothesize.** Each research cycle begins by analyzing the Findings Memory ($\mathcal{M}_t$), a list-style database containing thousands of structured records. Each record represents a unique scientific finding, which is categorized according to its stage of development. To overcome the LLM's context length constraints, we use a separate retrieval model (Wolters et al., 2024) when needed to select the Top-K Findings as input. In practice, the retrieved subset of Findings Memory for a single task typically fits within a long-context window of about $2 \times 10^5$ tokens, which is sufficient to contextualize the planner LLM without loss of relevant information. The vast majority of records begin as Idea Findings—unverified hypotheses. During this first stage, the system identifies limitations in existing knowledge and generates a new collection of hypotheses ($\mathcal{P}_{\text{new}}$), and then they evaluated by a low-cost Surrogate Model ($g_t$). The surrogate model (an LLM) is first contextualized with the entire Findings Memory. In implementation, this is realized by feeding the surrogate with the retrieved Top-K records from $\mathcal{M}_t$ together with the candidate hypothesis, so that it can reason over representative past successes and failures. It then approximates the true value function $f$ and, for each candidate finding $I \in \mathcal{P}_{\text{new}}$, produces a structured valuation vector $V = \langle v_u, v_q, v_e \rangle$, quantifying its estimated utility, quality, and exploration value as integer scores on a scale of 0 to 100. Each new hypothesis and its valuation vector is then used to initialize a new record in the Findings Memory as an "Idea Finding".

**Stage II: Implement & Verify.** This stage serves as the primary filter in the Findings Memory. To decide which of the numerous "Idea Findings" warrants the significant resource investment to be advanced in a real-world experiment, the system employs an Acquisition Function ($\alpha$). Specifically, it uses the classic Upper Confidence Bound (UCB) algorithm to select the most promising record.

Table 1: Overview of the three different human SOTA methods we selected.

| Task | Method | Venue | Benchmark | Github Star |
|---|---|---|---|---|
| Agents Failure Attribution | All at Once | ICML 2025 Spotlight | Who&When | 302 |
| LLM Inference Accel. | TokenRecycling | ACL 2025 Outstanding | MBPP | 323 |
| AI Text Detection | FastDetectGPT | ICLR 2024 | RAID | 414 |

The UCB formula maps the valuation vector $V$ to balance the trade-off between exploiting promising avenues (represented by $v_u$ and $v_q$) and exploring uncertain ones (represented by $v_e$):

$$I_{t+1} = \arg\max_{I \in \mathcal{P}_{\text{new}}} \big( \underbrace{w_u v_u + w_q v_q}_{\text{Exploitation Term } \mu(I)} + \kappa \cdot \underbrace{v_e}_{\text{Exploitation Term } \sigma(I)} \big), \tag{1}$$

where $w_u$ and $w_q$ are hyperparameters and $\kappa$ controls the intensity of exploration. we adopt a simple, task-agnostic configuration with $w_u = w_q = \kappa = 1$, and do not tune these hyperparameters across the three tasks; this choice reflects an assumption of equal importance among expected utility, quality, and exploration, and ablations. The highest-scoring finding $I_{t+1}$ is selected for validation, and its record is promoted to the status of an Implement Finding. A coding agent then performs a repository-level implementation to execute the experiment. This agent operates within a sandboxed environment with full permissions, allowing it to read the complete code repository and access the internet for literature and code searches. Its objective is to implement the new hypothesis on top of the existing SOTA method's repositories. The agent typically begins by planning the task, then reads the code to understand its structure, and finally implements the changes to produce the experimental logs and results. The experiment logs and results, $f(I_{t+1})$, are used to update the corresponding record, enriching it with empirical evidence and thus closing the learning loop.

**Stage III: Analyze & Report.** The final and most selective stage of the Findings Memory is triggered only by a successful validation. When an "Implement Finding" succeeds in surpassing the baseline, its record is promoted to a Progress Finding. This transformation is implemented by a series of specialized agents capable of utilizing a suite of MCP (Hou et al., 2025) tools. These agents first autonomously design and execute a series of deeper analytical experiments (e.g., ablations, evaluations on new datasets), leveraging MCP tools to manage the experimental lifecycle, data collection, and result parsing. Subsequently, a synthesis agent employs the same toolset to collate all experimental results, analytical insights, and generated artifacts into a coherent, reproducible research paper. The resulting deeply validated record is written back into the Findings Memory as a high-confidence Progress Finding, and, like all other records, will be retrieved and reused in subsequent cycles, allowing the system to learn from both its successes and its failures.

## 4 EXPERIMENTS

As detailed in Table 1, we select three distinct SOTA methods (published in 2024 and 2025) as starting points, chosen for their frontier status, community interest, and human supervisability. Each SOTA method is manually reproduced, and we preserve execution logs and test scripts to allow DeepScientist to focus on research advancement. DeepScientist is provided with two servers, each with 8 Nvidia H800 GPUs. To maximize utilization, we launch a separate system instance for each GPU, employing the Gemini-2.5-Pro model for core logic and the Claude-4-Opus model for its robust code-generation capabilities. Three human experts supervise the process to verify outputs and filter out hallucinations. For more implementation details, please see Appendix F.

### 4.1 DEEPSCIENTIST ACHIEVEMENTS ON THREE RESEARCH DOMAINS

We evaluate DeepScientist on three frontier AI research tasks where strong human-designed SOTA methods already exist, and ask whether the system can discover methods that further advance these frontiers (Figure 3). For each task, we briefly recall the problem and baseline, then summarize the method discovered by DeepScientist and its improvement over the human SOTA.

**Agents Failure Attribution.** The goal of Agent Failure Attribution is, given a failed episode in an LLM-based multi-agent system, to identify which agent and which step were decisively responsible for the failure, which is crucial for debugging complex agent pipelines. The human SOTA

| Method | Agent Failure Attribution | | LLM Inference Acceleration | AI Text Detection | |
| --- | --- | --- | --- | --- | --- |
| | Handcraft (Acc.) | Algorithm-Gen (Acc.) | Tokens/second | AUROC | Latency |
| Human SoTA method | 12.07% (All at Once) | 16.67% (All at Once) | 190.25 (Token Recycling) | 0.800 (Binoculars) | 117ms (Binoculars) |
| DeepScientist's method | **29.31%** (A2P) | **47.46%** (A2P) | **193.90** (ACRA) | **0.863** (PA-TDT) | **60ms** (PA-TDT) |
| **Improvement** | △+142.8% (+17.24) | △+183.7% (+30.79) | △+1.9% (+3.65) | △+7.9% (+0.063) | △+190%↓ (-57) |

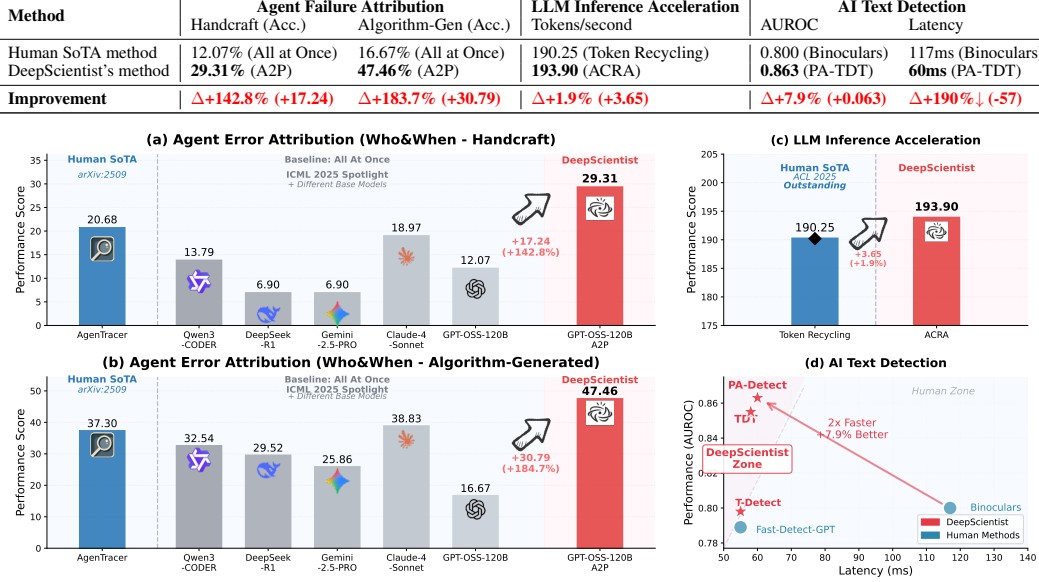

Figure 3: Performance evaluation of DeepScientist across three research domains: (a-b) Agent Failure Attribution on Who&When benchmark in handcraft and algorithm-generated settings; (c) LLM Inference Acceleration on MBPP dataset; (d) AI Text Detection with performance-latency tradeoff analysis. DeepScientist consistently outperform human-designed SoTA approaches across all tasks.

"All at once" method (Zhang et al., 2025c) feeds the entire failure log to a judge LLM and asks it to directly predict the responsible agent and step, but this approach relies on pattern recognition over static logs and lacks explicit counterfactual reasoning, making step-level attribution and chain-like failures challenging. Starting from the baseline "All at once" method, DeepScientist identified that the current approach lacks the counterfactual reasoning capabilities essential for attribution. Through a process of trial, error, and synthesizing new findings—discovering the effectiveness of hypothetical prediction and simulated attempts—it ultimately proposed the *A2P* method. Named for its Abduction-Action-Prediction process, its core innovation elevates failure attribution from pattern recognition to causal reasoning, filling the critical gap in counterfactual capabilities by predicting if a proposed fix would have led to success. Concretely, A2P first hypothesizes hidden causes behind a suspicious action, then proposes a counterfactual fix, and finally simulates several future steps under this intervention to test whether the task would have succeeded. As shown in Figure 3.(a-b), A2P achieved scores of 29.31 and 47.46 in the "handcraft" and "algorithm-generated" settings of the Who&When benchmark, respectively, setting a new state-of-the-art (SOTA). In this task, DeepScientist validated that a structured, zero-shot causal reasoning framework can be superior to less principled methods. As of September 2025, the training-free A2P method maintains its SOTA position, outperforming even 7B models trained on synthetic data. (Zhang et al., 2025a).

**LLM Inference Acceleration** is a highly optimized field aiming to maximize throughput and reduce latency during LLM inference. The human SOTA baseline Token Recycling (TR) (Xia et al., 2024) reuses rejected candidate tokens produced during decoding via a tree-structured speculative decoding scheme, but effectively treats decoding as a near–first-order Markov process and primarily exploits local transition patterns, limiting its ability to consistently leverage longer-range regularities. In this process, the system actively made many different attempts, such as using a Kalman Filter (Zarchan, 2005) to dynamically adjust an adjacency matrix to address the original method's lack of a memory function. Although most of these attempts failed, the system-generated *ACRA* method ultimately advanced the MPBB (Austin et al., 2021) from a human SOTA of 190.25 to 193.90 tokens/second by identifying stable suffix patterns, as shown in Figure 3. ACRA assumes that LLM decoding exhibits recurrent, variable-length stable suffixes: it maintains a suffix-indexed history, finds the longest stable suffix matching the current context, and, only when a stability gate is passed, uses the associated next-token statistics to override the first layer of draft tokens; otherwise, it falls back to the original TR scheme. Scientifically, this innovation is significant because it uses this extra contextual information to dynamically adjust the decoding guess, effectively grafting a long-term

Table 2: Evaluation of AI-generated papers produced by various AI Scientist systems. Scores represent the average ratings given by DeepReviewer-14B (Zhu et al., 2025a) across the number ("Num") of available papers. Note: Publicly available papers may be curated and therefore may not fully represent the typical output of each system.

| AI Scientist Systems | Number | Soundness | Presentation | Contribution | Rating | Accept Rate |
|---|---|---|---|---|---|---|
| AI SCIENTIST | 10 | 2.08 | 1.80 | 1.75 | 3.35 | 0% |
| HKUSD AI Researcher | 7 | 1.75 | 1.46 | 1.57 | 2.57 | 0% |
| AI SCIENTIST-V2 | 3 | 1.67 | 1.50 | 1.50 | 2.33 | 0% |
| CycleResearcher-12B | 6 | 2.25 | 1.75 | 2.13 | 3.75 | 0% |
| Zochi | 2 | 2.38 | 2.38 | 2.25 | 4.63 | 0% |
| DeepScientist (Ours) | 5 | **2.90** | **2.90** | **2.90** | **5.90** | **60%** |

memory onto the process and breaking the context-collapsing of standard decoders. This discovery highlights the system's primary goal: the creation of new, human-unknown knowledge rather than mere engineering optimization. For instance, one could likely achieve greater performance gains by combining ACRA with an established technique like layer skipping (Wang et al., 2022) or PageAttention (Kwon et al., 2023), but this would represent an engineering effort, not a scientific one. The exploration assessment within our process avoids such combinations of existing knowledge.

**AI Text Detection** is a binary classification task where (Dugan et al., 2024), given a text that may contain content from an LLM (and possibly additional noise), the goal is to determine if it was produced by a human or an LLM (Li et al., 2022; Ghosal et al., 2023; Su et al.; Bao et al., a;b; Hu et al., 2023). This capability underpins applications such as exam integrity, content moderation, and model misuse detection. Recent human-designed SOTA detectors such as Fast-Detect GPT and Binoculars (Hans et al., 2024) exploit differences in perplexity, burstiness, or style between human and model distributions, but these global-statistic approaches assume relatively stationary gaps and often degrade when modern LLMs actively mimic human style or texts are paraphrased or lightly edited. To validate its capacity for sustained advancement, DeepScientist made numerous attempts that included addressing the Boundary-Aware Extension problem and exploring approaches like Volatility-Aware and Wavelet Subspace Energy methods. The final results show a dramatic acceleration in scientific discovery: in a rapid evolution over just two weeks, the system produced three distinct, progressively superior methods. This began with *T-Detect* fixing core statistics with a robust t-distribution, then evolved conceptually with *TDT* and *PA-TDT*, which treat text as a signal and use wavelet and phase congruency analysis to pinpoint anomalies. Taken together, these methods shift the perspective from global distributional differences to the non-stationary, time–frequency structure of AI-generated text, showing that localized changes in energy and phase carry the key evidence for detection. Scientifically, this shift reveals the "non-stationarity" of AI-generated text, alleviating the information bottleneck in prior paradigms that average away localized evidence. As shown in Figure 1 and 3(d), this entire discovery trajectory demonstrates DeepScientist's ability for advancing frontier-pushing scientific findings progressively, establishing a new SOTA with a 7.9% higher AUROC while also doubling the inference speed.

## 4.2 ASSESSING THE QUALITY OF AI-GENERATED RESEARCH PAPER

Table 3: Evaluation of DeepScientist's papers produced by human experts. Values are presented as mean (variance) from three reviewers. Inter-rater reliability for Rating: Krippendorff's $\alpha = 0.739$.

| Paper | Confidence | Soundness | Presentation | Contribution | Rating |
|---|---|---|---|---|---|
| HUMAN Avg. (ICLR 2025) | - | 2.59 | 2.36 | 2.62 | 5.08 |
| 1. T-DETECT | 4.33 (0.33) | 2.00 (1.00) | **2.67** (0.33) | **2.67** (0.33) | 5.00 (0.00) |
| 2. TDT | 4.67 (0.33) | **3.00** (0.00) | **3.00** (0.00) | **3.00** (0.00) | **5.67** (0.33) |
| 3. PA-TDT | 4.00 (0.00) | 1.67 (0.33) | 2.00 (1.00) | 2.00 (1.00) | 4.33 (1.33) |
| 4. A2P | 4.00 (0.00) | **3.00** (0.00) | **3.00** (0.00) | **2.67** (0.33) | **5.67** (0.33) |
| 5. ACRA | 3.33 (0.33) | 1.67 (0.33) | 2.00 (1.00) | 1.67 (0.33) | 4.33 (1.33) |
| DeepScientist Avg. | 4.07 | 2.27 | **2.53** | 2.40 | 5.00 |

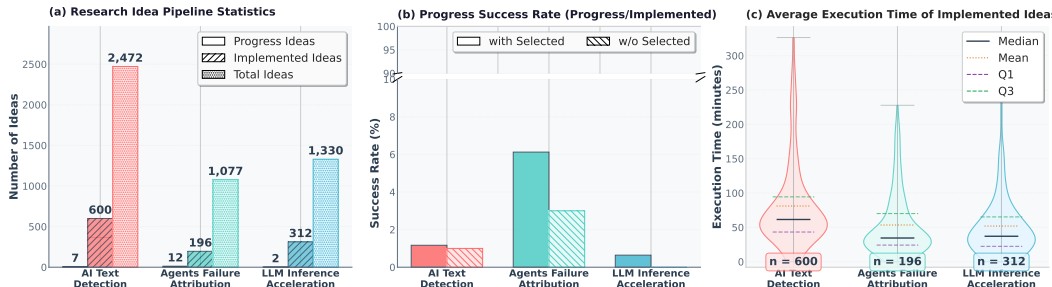

Figure 4: DeepScientist's experimental statistics. (a) The research pipeline from generated ideas to validated progress. (b) Success rates comparing our selection strategy against a baseline. (c) Distribution of wall-clock execution times for all implemented trials.

**Experimental Setup.** To assess the quality of the final output, we evaluate the five research papers autonomously generated by DeepScientist's end-to-end process. Our evaluation protocol is twofold. First, to benchmark against existing work, we employ DeepReviewer (Zhu et al., 2025a), an AI agent that simulates the human peer-review process with an external search capability, comparing Deep-Scientist's output against 28 publicly available papers from other AI Scientist systems. Second, for a more rigorous assessment, we convene a dedicated program committee consisting of three active LLM researchers: two volunteers who have served as ICLR reviewers and one senior volunteer who has been invited to be an ICLR Area Chair. The generated papers are available in Appendix F.

**Automated Review Against Other AI Scientist Systems.** As shown in Table 2, the results from the LLM-based automatic evaluation indicate that the system's outputs are recognized for their scientific novelty and value. When benchmarked against 28 publicly available papers from other AI Scientist systems using DeepReviewer, DeepScientist is the only system whose papers achieve a 60% simulated acceptance rate under the same reviewing protocol.

**Human Expert Evaluation.** The evaluation from our human program committee, shown in Table 3, reveals a strong and consistent consensus: DeepScientist's outputs are particularly strong in ideation, the most challenging and often rate-limiting step in human-led research. Full details on the review protocol are provided in Appendix B, and the core ideas within each paper are praised for their genuine novelty, ingenuity, and scientific contributions. The quality of these innovations is further demonstrated by the review scores: the system's average rating (5.00) closely mirrors the average of all ICLR 2025 submissions (5.08), with two of its papers significantly exceeding this (5.67).

## 4.3 ANALYSIS OF THE ITERATIVE TRAJECTORY OF AUTONOMOUS EXPLORATION

**Experimental Setup.** The findings in this section are derived from a series of post-hoc analyses conducted on the complete operational data generated by DeepScientist across the three frontier tasks. This data includes the full set of execution logs and the Findings Memory, providing the basis for all subsequent statistical analysis. To visualize the conceptual search space (Figure 5), we embed the complete description of each generated finding using the Qwen3-Embedding-8B model. To assess scalability (Figure 6), we conduct a dedicated one-week experiment where N identified limitations of a single SOTA method are assigned to N parallel GPU instances. These instances explore solutions independently but share their findings to a central database, which are synchronized globally every five cycles to accommodate the asynchronous nature of the discovery process. Finally, to better understand the low success rate, our program committee experts perform a detailed causal attribution analysis on a sample of 300 failed implementations.

**Our analysis of DeepScientist's experimental logs reveals the sheer scale of the trial-and-error process inherent in autonomous scientific discovery.** Even in our relatively fast-executing domains, achieving progress required hundreds of trials per task. As show in Figure 4, the execution time distributions show that while individual experiments may be quick, the sheer volume of trial-and-error necessary to uncover a successful idea is substantial. This suggests a clear application boundary for current autonomous science: for tasks with rapid feedback loops, such as aspects of chip design, delegating massive-scale experimentation to AI is a powerful strategy. However, for high-cost endeavors like pre-training foundation models or pharmaceutical synthesis, the low suc-

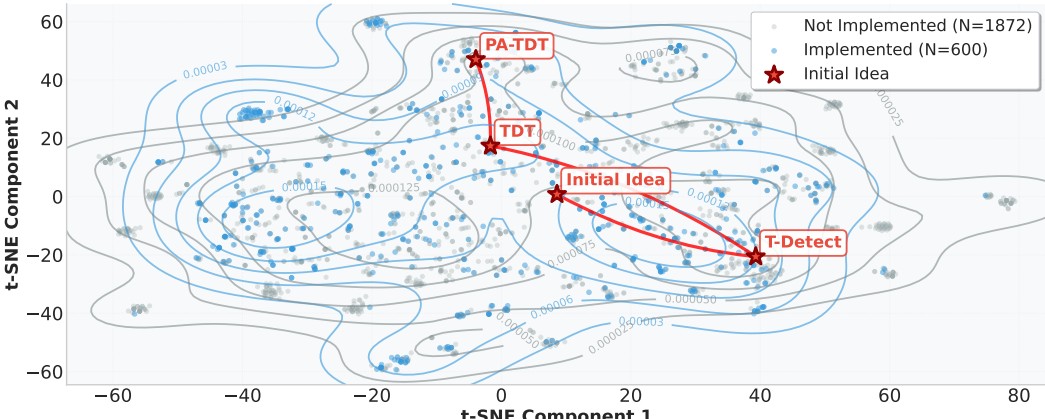

Figure 5: Visualization of the conceptual search space for the AI text detection task. The plot shows a t-SNE visualization of the semantic embeddings for all 2,472 generated ideas. Markers identify the initial SOTA method (Initial Idea) and the three final SOTA-surpassing methods (Progress Ideas).

cess rate makes such an approach currently impractical, mandating continued reliance on human-led ideation. The autonomous research process is characterized by a vast exploratory funnel where promising ideas are exceptionally rare. Across the three tasks, DeepScientist generate over 5,000 unique ideas, yet only about 1,100 are deemed worthy of experimental validation by the system's selection mechanism, and a mere 21 ultimately result in scientific progress. These 21 are promoted to *Progress Findings* (ideas that surpass the then-current SOTA in Stage II), and the fully automated Stage III pipeline turns them into full drafts and filters them with an LLM-based reviewer, yielding 5 final papers. An ablation study underscores the criticality of this selection process: without it, randomly sampling 100 ideas for each task and testing them yields a success rate of effectively zero. The low success rate is not merely a matter of failed hypotheses; analysis by human experts on a sample of failed trials reveals that approximately 60% were terminated prematurely due to implementation errors, while the vast majority of the remaining 40% simply offered no performance improvement or caused a regression. This highlights that the probability of an LLM-generated idea being both correct in its premise and flawless in its implementation is exceedingly low. In other words, the executor largely determines whether ideas can be executed at all and how often they fail, while the planner determines how far the system can advance under a fixed budget. The success of this work, therefore, is not a product of brute-force computation but of search efficiency. A naive approach of fully testing all ∼5000 promising candidates would have required over 100,000 GPU hours, whereas our targeted exploration achieved its breakthroughs using only ∼20,000.

**DeepScientist's discovery process follows a purposeful and progressive trajectory.** The semantic distribution of ideas generated for the AI text detection task, as shown in Figure 5, reveals the characteristics of this sophisticated strategy. While the system generates thousands of diverse ideas across a vast conceptual landscape, its path to success is not random but is a series of focused, logical advancements. This indicates a capacity to progressively deepen its understanding: after achieving an initial breakthrough with *T-Detect*, the system effectively establishes a SoTA, identifies its subsequent limitations, and reorients its search towards a new goal. This dynamic exploration is exemplified by the conceptual shift towards *TDT* and *PA-TDT*, which build upon the previous success by leveraging new positional and temporal information. This ability to build upon its own discoveries, turning each successful finding into a new starting point for identifying and solving the next set of limitations, demonstrates a powerful capacity for scientific exploration.

**Scaling Laws in DeepScientist's Scientific Discovery.** To investigate the relationship between computational scale and the rate of scientific progress, we evaluated the number of "Progress Findings" generated by DeepScientist within a fixed one-week period as a function of available parallel resources in Figure 6. In this setup, the system first identified a set of limitations in the baseline method, and each parallel exploration path was tasked with resolving a distinct limitation, with all paths periodically synchronizing their results into a shared Findings Memory . Our results indicate a promising scaling trend. While minimal resources yielded no breakthroughs, the rate of discovery began to increase effectively as we scaled to 4 GPUs and beyond, growing from one SOTA-

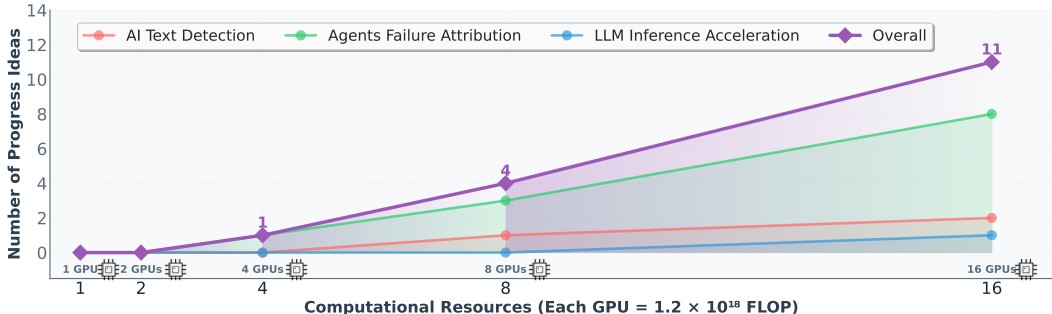

Figure 6: Scaling analysis of autonomous scientific discovery. The plot illustrates the relationship between parallel computational resources (number of GPUs) and the number of SOTA-surpassing "Progress Findings" found by DeepScientist across all tasks within a one-week period.

surpassing finding with 4 GPUs to eleven with 16 GPUs . This appears to establish a near-linear relationship between the resources allocated and the output of valuable scientific discoveries. We hypothesize this efficiency stems from more than just parallel trial-and-error; it is a direct result of the shared knowledge architecture. Mechanistically, this knowledge-driven gain should also apply when scaling with time (serial execution); indeed, our preliminary 4-week single-GPU tests confirmed this, yielding new progress approximately every 8-14 days. While serial exploration may be more sample-efficient due to real-time memory updates (whereas our parallel setup synchronizes periodically ), the immense value of parallel scaling lies in its wall-clock time advantage—compressing months of discovery into a single week. This distinction highlights that parallel scaling demonstrates the scalability of the knowledge-sharing mechanism, not just its effectiveness. As each parallel path explores, it enriches the shared Findings Memory. This creates a synergistic effect where the collective intelligence of the system grows (Schmidgall & Moor, 2025; Zhang et al., 2025b) , allowing each independent path to benefit from the successes and, just as importantly, the failures of others. This suggests that effectively scaling autonomous science is not just a matter of increasing brute-force computation, but of fostering a richer, interconnected knowledge base that accelerates discovery across all concurrent efforts.

## 4.4 DISCUSSION

The results from DeepScientist suggest a new paradigm in scientific exploration, defined not by infallibility but by massive scale and efficiency. The system's 1-5% progress rate mirrors the reality of frontier research, successfully compressing years of human exploration into weeks . The primary path forward is systematically improving this efficiency. Our analysis identifies key bottlenecks: enhancing the robustness of implementation (as 60% of failed trials stemmed from implementation errors, not flawed hypotheses ) and improving scientific rigor (as human evaluations praised the system's conceptual novelty but noted a lack of deep validation ). This highlights a powerful opportunity for human-AI synergy, where humans provide high-level strategic direction while the AI handles rapid, exhaustive exploration . Our scaling analysis confirms this path is viable, showing a near-linear relationship between parallel resources and discoveries, driven not by brute-force computation, but by a shared knowledge base that accelerates discovery across all concurrent efforts . Future work should focus on these efficiencies, develop simulated discovery environments, and bridge the gap to the physical sciences through robotics.

## 5 CONCLUSION

This work presents the first large-scale empirical evidence that an autonomous AI can achieve progressively, SOTA-surpassing progress on modern scientific frontiers. We introduced DeepScientist, a goal-oriented system achieving end-to-end autonomy from ideation to real progress, which learns by synthesizing human knowledge with its own findings from iteration of trials. Results across multiple domains serves to accelerate the progress of real-world scientific discovery, providing a crucial foundation. Our findings can signal a foundational shift in AI research, heralding an era where the pace of discovery is no longer solely dictated by the cadence of human thought.

ACKNOWLEDGEMENT

This publication has been supported by the National Natural Science Foundation of China (NSFC) Key Project under Grant Number 62336006 and 625B2152, and the 2025 Dean's Special "PhD Student Project" of the School of Engineering, Westlake University.

We are grateful to Professor Linyi Yang for his insightful discussions on this paper. This work is inspired by pioneering efforts in automated scientific discovery, including AI Scientist (Lu et al., 2024; Yamada et al., 2025) and AlphaEvolve (Novikov et al., 2025).

ETHICS STATEMENT

The development of DeepScientist, an autonomous system capable of advancing scientific frontiers, carries profound ethical responsibilities. Our primary goal is to accelerate discovery for the benefit of humanity, but we recognize the potential for misuse. The most significant risks include the application of this technology to advance dangerous research and the potential degradation of the academic ecosystem. We have implemented specific, robust measures to address these concerns proactively.

A primary concern is the dual-use risk, where the system could be co-opted to accelerate research in harmful domains, such as developing novel toxins or malicious software. To assess and mitigate this, we conducted red-teaming exercises specifically targeting the generation of computer viruses. We tasked the system, powered by leading foundation models (including GPT-5, Gemini-2.5-Pro, and Claude-4.1-Opus in our testbed), with this malicious objective. In all instances, the underlying models exhibited robust safety alignment, refusing to proceed with the research. They correctly identified the task as illegal and harmful, and autonomously terminated the research cycle, demonstrating that foundation model safety protocols provide a critical defense layer.

We are also deeply conscious of the potential negative impact on the academic ecosystem. It is crucial to state that all results from DeepScientist presented in this paper, including code and experimental findings, have undergone rigorous human verification. Recognizing that others might neglect this critical oversight, we are adopting a selective open-sourcing policy to mitigate the risk of proliferating unreliable publications. We will open-source the core components that drive continuous discovery, as we believe their potential to accelerate progress for the community outweighs the risks. However, we will deliberately refrain from open-sourcing the "Analyze & Report" module. This decision is made to prevent the automated generation of seemingly credible but scientifically unverified papers, thereby safeguarding the integrity of the academic record.

Ultimately, we envision DeepScientist as a powerful tool to augment, not replace, human intellect and judgment. To enforce this vision, our open-source components will be released under a license based on MIT, but with explicit addendums that codify our ethical framework. This license will strictly prohibit any use of the software for harmful research. Furthermore, it will legally require that a human user must supervise the entire operational process of DeepScientist and assumes full and final responsibility for all its outputs. By embedding these requirements directly into our terms of use, we aim to foster a research environment where AI-driven discovery proceeds with the necessary human accountability and ethical oversight.

REFERENCES

Jacob Austin, Augustus Odena, Maxwell Nye, Maarten Bosma, Henryk Michalewski, David Dohan, Ellen Jiang, Carrie Cai, Michael Terry, Quoc Le, et al. Program synthesis with large language models. *arXiv preprint arXiv:2108.07732*, 2021.

Eser Aygün, Anastasiya Belyaeva, Gheorghe Comanici, Marc Coram, Hao Cui, Jake Garrison, Renee Johnston Anton Kast, Cory Y. McLean, Peter Norgaard, Zahra Shamsi, David Smalling, James Thompson, Subhashini Venugopalan, Brian P. Williams, Chujun He, Sarah Martinson, Martyna Plomecka, Lai Wei, Yuchen Zhou, Qian-Ze Zhu, Matthew Abraham, Erica Brand, Anna Bulanova, Jeffrey A. Cardille, Chris Co, Scott Ellsworth, Grace Joseph, Malcolm Kane, Ryan Krueger, Johan Kartiwa, Dan Liebling, Jan-Matthis Lueckmann, Paul Raccuglia, Xuefei, Wang, Katherine Chou, James Manyika, Yossi Matias, John C. Platt, Lizzie Dorfman, Shibl Mourad, and

Michael P. Brenner. An ai system to help scientists write expert-level empirical software, 2025. URL https://arxiv.org/abs/2509.06503.

Jinheon Baek, Sujay Kumar Jauhar, Silviu Cucerzan, and Sung Ju Hwang. Researchagent: Iterative research idea generation over scientific literature with large language models. In *Proceedings of the 2025 Conference of the Nations of the Americas Chapter of the Association for Computational Linguistics: Human Language Technologies (Volume 1: Long Papers)*, pp. 6709–6738, 2025.

Guangsheng Bao, Yanbin Zhao, Juncai He, and Yue Zhang. Glimpse: Enabling white-box methods to use proprietary models for zero-shot llm-generated text detection. In *The Thirteenth International Conference on Learning Representations*, a.

Guangsheng Bao, Yanbin Zhao, Zhiyang Teng, Linyi Yang, and Yue Zhang. Fast-detectgpt: Efficient zero-shot detection of machine-generated text via conditional probability curvature. In *The Twelfth International Conference on Learning Representations*, b.

Jun Shern Chan, Neil Chowdhury, Oliver Jaffe, James Aung, Dane Sherburn, Evan Mays, Giulio Starace, Kevin Liu, Leon Maksin, Tejal Patwardhan, et al. Mle-bench: Evaluating machine learning agents on machine learning engineering. *arXiv preprint arXiv:2410.07095*, 2024.

Cristina Cornelio, Sanjeeb Dash, Vernon Austel, Tyler R. Josephson, Joao Goncalves, Kenneth L. Clarkson, Nimrod Megiddo, Bachir El Khadir, and Lior Horesh. Combining data and theory for derivable scientific discovery with AI-Descartes. *Nature Communications*, 14(1):1777, April 2023. ISSN 2041-1723. doi: 10.1038/s41467-023-37236-y. URL https://doi.org/10.1038/s41467-023-37236-y.

Cristina Cornelio, Takuya Ito, Ryan Cory-Wright, Sanjeeb Dash, and Lior Horesh. The need for verification in ai-driven scientific discovery, 2025. URL https://arxiv.org/abs/2509.01398.

Ryan Cory-Wright, Cristina Cornelio, Sanjeeb Dash, Bachir El Khadir, and Lior Horesh. Evolving scientific discovery by unifying data and background knowledge with ai hilbert. *Nature Communications*, 15:5922, July 2024. doi: 10.1038/s41467-024-50074-w. URL https://doi.org/10.1038/s41467-024-50074-w.

Liam Dugan, Alyssa Hwang, Filip Trhlík, Andrew Zhu, Josh Magnus Ludan, Hainiu Xu, Daphne Ippolito, and Chris Callison-Burch. RAID: A shared benchmark for robust evaluation of machine-generated text detectors. In *Proceedings of the 62nd Annual Meeting of the Association for Computational Linguistics (Volume 1: Long Papers)*, pp. 12463–12492, Bangkok, Thailand, August 2024. Association for Computational Linguistics. URL https://aclanthology.org/2024.acl-long.674.

Alhussein Fawzi, Matej Balog, Aja Huang, Thomas Hubert, Bernardino Romera-Paredes, Mohammadamin Barekatain, Alexander Novikov, Francisco J R. Ruiz, Julian Schrittwieser, Grzegorz Swirszcz, et al. Discovering faster matrix multiplication algorithms with reinforcement learning. *Nature*, 610(7930):47–53, 2022.

Alexander Fleming. Penicillin. *British medical journal*, 2(4210):386, 1941.

Soumya Suvra Ghosal, Souradip Chakraborty, Jonas Geiping, Furong Huang, Dinesh Manocha, and Amrit Bedi. A survey on the possibilities & impossibilities of AI-generated text detection. *Transactions on Machine Learning Research*, 2023. ISSN 2835-8856. URL https://openreview.net/forum?id=AXtFeYjboj. Survey Certification.

Juraj Gottweis, Wei-Hung Weng, Alexander Daryin, Tao Tu, Anil Palepu, Petar Sirkovic, Artiom Myaskovsky, Felix Weissenberger, Keran Rong, Ryutaro Tanno, et al. Towards an ai co-scientist. *arXiv preprint arXiv:2502.18864*, 2025.

Martin A Green. Silicon solar cells: evolution, high-efficiency design and efficiency enhancements. *Semiconductor science and technology*, 8(1):1, 1993.

Abhimanyu Hans, Avi Schwarzschild, Valeriia Cherepanova, Hamid Kazemi, Aniruddha Saha, Micah Goldblum, Jonas Geiping, and Tom Goldstein. Spotting llms with binoculars: Zero-shot detection of machine-generated text. In *International Conference on Machine Learning*, pp. 17519–17537. PMLR, 2024.

Xinyi Hou, Yanjie Zhao, Shenao Wang, and Haoyu Wang. Model context protocol (mcp): Landscape, security threats, and future research directions. *arXiv preprint arXiv:2503.23278*, 2025.

Xiaomeng Hu, Pin-Yu Chen, and Tsung-Yi Ho. Radar: Robust ai-text detection via adversarial learning. *Advances in neural information processing systems*, 36:15077–15095, 2023.

Intology. Zochi technical report. *arXiv*, 2025.

Tang Jiabin, Xia Lianghao, Li Zhonghang, and Huang Chao. Ai-researcher: Autonomous scientific innovation, 2025. URL https://arxiv.org/abs/2505.18705.

Woosuk Kwon, Zhuohan Li, Siyuan Zhuang, Ying Sheng, Lianmin Zheng, Cody Hao Yu, Joseph Gonzalez, Hao Zhang, and Ion Stoica. Efficient memory management for large language model serving with pagedattention. In *Proceedings of the 29th symposium on operating systems principles*, pp. 611–626, 2023.

Bin Li, Yixuan Weng, Qiya Song, and Hanjun Deng. Artificial text detection with multiple training strategies. *arXiv preprint arXiv:2212.05194*, 2022.

Yixiu Liu, Yang Nan, Weixian Xu, Xiangkun Hu, Lyumanshan Ye, Zhen Qin, and Pengfei Liu. Alphago moment for model architecture discovery. *arXiv preprint arXiv:2507.18074*, 2025.

Chris Lu, Cong Lu, Robert Tjarko Lange, Jakob Foerster, Jeff Clune, and David Ha. The ai scientist: Towards fully automated open-ended scientific discovery. *arXiv preprint arXiv:2408.06292v3*, 2024. URL https://www.arxiv.org/abs/2408.06292v3.

Jiacheng Miao, Joe R. Davis, Jonathan K. Pritchard, and James Zou. Paper2agent: Reimagining research papers as interactive and reliable ai agents, 2025. URL https://arxiv.org/abs/2509.06917.

Atsuyuki Miyai, Mashiro Toyooka, Takashi Otonari, Zaiying Zhao, and Kiyoharu Aizawa. Jr. ai scientist and its risk report: Autonomous scientific exploration from a baseline paper. *arXiv preprint arXiv:2511.04583*, 2025.

Gordon Moore. Moore's law. *Electronics Magazine*, 38(8):114, 1965.

Alexander Novikov, Ngân Vu, Marvin Eisenberger, Emilien Dupont, Po-Sen Huang, Adam Zsolt Wagner, Sergey Shirobokov, Borislav Kozlovskii, Francisco JR Ruiz, Abbas Mehrabian, et al. Alphaevolve: A coding agent for scientific and algorithmic discovery. Technical report, Technical report, Google DeepMind, 05 2025. URL https://storage. googleapis . . . , 2025.

José R Penadés, Juraj Gottweis, Lingchen He, Jonasz B Patkowski, Alexander Shurick, Wei-Hung Weng, Tao Tu, Anil Palepu, Artiom Myaskovsky, Annalisa Pawlosky, et al. Ai mirrors experimental science to uncover a novel mechanism of gene transfer crucial to bacterial evolution. *bioRxiv*, pp. 2025–02, 2025.

Ori Press, Brandon Amos, Haoyu Zhao, Yikai Wu, Samuel K Ainsworth, Dominik Krupke, Patrick Kidger, Touqir Sajed, Bartolomeo Stellato, Jisun Park, et al. Algotune: Can language models speed up general-purpose numerical programs? *arXiv preprint arXiv:2507.15887*, 2025.

Bernardino Romera-Paredes, Mohammadamin Barekatain, Alexander Novikov, Matej Balog, M Pawan Kumar, Emilien Dupont, Francisco JR Ruiz, Jordan S Ellenberg, Pengming Wang, Omar Fawzi, et al. Mathematical discoveries from program search with large language models. *Nature*, 625(7995):468–475, 2024.

Samuel Schmidgall and Michael Moor. Agentrxiv: Towards collaborative autonomous research. *arXiv preprint arXiv:2503.18102*, 2025.

Samuel Schmidgall, Yusheng Su, Ze Wang, Ximeng Sun, Jialian Wu, Xiaodong Yu, Jiang Liu, Zicheng Liu, and Emad Barsoum. Agent laboratory: Using llm agents as research assistants. *arXiv preprint arXiv:2501.04227v1*, 2025a. URL https://www.arxiv.org/abs/2501.04227v1.

Samuel Schmidgall, Yusheng Su, Ze Wang, Ximeng Sun, Jialian Wu, Xiaodong Yu, Jiang Liu, Zicheng Liu, and Emad Barsoum. Agent laboratory: Using llm agents as research assistants. *arXiv preprint arXiv:2501.04227*, 2025b.

Parshin Shojaee, Kazem Meidani, Shashank Gupta, Amir Barati Farimani, and Chandan K Reddy. Llm-sr: Scientific equation discovery via programming with large language models. In *The Thirteenth International Conference on Learning Representations*.

Giulio Starace, Oliver Jaffe, Dane Sherburn, James Aung, Jun Shern Chan, Leon Maksin, Rachel Dias, Evan Mays, Benjamin Kinsella, Wyatt Thompson, et al. Paperbench: Evaluating ai's ability to replicate ai research. *arXiv preprint arXiv:2504.01848*, 2025.

Jinyan Su, Terry Yue Zhuo, Di Wang, and Preslav Nakov. Detectllm: Leveraging log rank information for zero-shot detection of machine-generated text. In *The 2023 Conference on Empirical Methods in Natural Language Processing*.

Kyle Swanson, Wesley Wu, Nash L Bulaong, John E Pak, and James Zou. The virtual lab of ai agents designs new sars-cov-2 nanobodies. *Nature*, pp. 1–3, 2025.

Jue Wang, Ke Chen, Gang Chen, Lidan Shou, and Julian McAuley. Skipbert: Efficient inference with shallow layer skipping. In *Proceedings of the 60th Annual Meeting of the Association for Computational Linguistics (Volume 1: Long Papers)*, pp. 7287–7301, 2022.

Yixuan Weng, Minjun Zhu, Guangsheng Bao, Hongbo Zhang, Jindong Wang, Yue Zhang, and Linyi Yang. Cycleresearcher: Improving automated research via automated review. In *The Thirteenth International Conference on Learning Representations*, 2025. URL https://openreview.net/forum?id=bjcsVLoHYs.

Christopher Wolters, Xiaoxuan Yang, Ulf Schlichtmann, and Toyotaro Suzumura. Memory is all you need: An overview of compute-in-memory architectures for accelerating large language model inference, 2024. URL https://arxiv.org/abs/2406.08413.

Heming Xia, Zhe Yang, Qingxiu Dong, Peiyi Wang, Yongqi Li, Tao Ge, Tianyu Liu, Wenjie Li, and Zhifang Sui. Unlocking efficiency in large language model inference: A comprehensive survey of speculative decoding. In Lun-Wei Ku, Andre Martins, and Vivek Srikumar (eds.), *Findings of the Association for Computational Linguistics ACL 2024*, pp. 7655–7671, Bangkok, Thailand and virtual meeting, August 2024. Association for Computational Linguistics. doi: 10.18653/v1/2024.findings-acl.456. URL https://aclanthology.org/2024.findings-acl.456.

Qiujie Xie, Yixuan Weng, Minjun Zhu, Fuchen Shen, Shulin Huang, Zhen Lin, Jiahui Zhou, Zilan Mao, Zijie Yang, Linyi Yang, et al. How far are ai scientists from changing the world? *arXiv preprint arXiv:2507.23276*, 2025.

Yutaro Yamada, Robert Tjarko Lange, Cong Lu, Shengran Hu, Chris Lu, Jakob Foerster, Jeff Clune, and David Ha. The ai scientist-v2: Workshop-level automated scientific discovery via agentic tree search. *arXiv preprint arXiv:2504.08066*, 2025.

Paul Zarchan. *Progress in astronautics and aeronautics: fundamentals of Kalman filtering: a practical approach*, volume 208. Aiaa, 2005.

Aohan Zeng, Xin Lv, Qinkai Zheng, Zhenyu Hou, Bin Chen, Chengxing Xie, Cunxiang Wang, Da Yin, Hao Zeng, Jiajie Zhang, et al. Glm-4.5: Agentic, reasoning, and coding (arc) foundation models. *arXiv preprint arXiv:2508.06471*, 2025.

Guibin Zhang, Junhao Wang, Junjie Chen, Wangchunshu Zhou, Kun Wang, and Shuicheng Yan. Agentracer: Who is inducing failure in the llm agentic systems?, 2025a. URL https://arxiv.org/abs/2509.03312.

Pengsong Zhang, Heng Zhang, Huazhe Xu, Renjun Xu, Zhenting Wang, Cong Wang, Animesh Garg, Zhibin Li, Arash Ajoudani, and Xinyu Liu. Scaling laws in scientific discovery with ai and robot scientists. *arXiv preprint arXiv:2503.22444*, 2025b.

Shaokun Zhang, Ming Yin, Jieyu Zhang, Jiale Liu, Zhiguang Han, Jingyang Zhang, Beibin Li, Chi Wang, Huazheng Wang, Yiran Chen, and Qingyun Wu. Which agent causes task failures and when? on automated failure attribution of LLM multi-agent systems. In *Forty-second International Conference on Machine Learning*, 2025c. URL https://openreview.net/forum?id=GazlTYxZss.

Huichi Zhou, Yihang Chen, Siyuan Guo, Xue Yan, Kin Hei Lee, Zihan Wang, Ka Yiu Lee, Guchun Zhang, Kun Shao, Linyi Yang, and Jun Wang. Memento: Fine-tuning llm agents without fine-tuning llms, 2025. URL https://arxiv.org/abs/2508.16153.

Minjun Zhu, Yixuan Weng, Linyi Yang, and Yue Zhang. Deepreview: Improving llm-based paper review with human-like deep thinking process. *arXiv preprint arXiv:2503.08569*, 2025a.

Minjun Zhu, Qiujie Xie, Yixuan Weng, Jian Wu, Zhen Lin, Linyi Yang, and Yue Zhang. Ai scientists fail without strong implementation capability. *arXiv preprint arXiv:2506.01372*, 2025b.

## A    USE OF LARGE LANGUAGE MODELS

Large Language Models are a foundational component of the DeepScientist system and were integral to every stage of the research presented in this paper. The core reasoning, hypothesis generation, and experimental analysis were driven by Gemini-2.5-Pro, while all code implementation, including writing, testing, and debugging, was performed by Claude-4-Opus.

The LLM agents autonomously conducted the entire scientific workflow. For the five SOTA-surpassing findings detailed in this work, the complete research process—from the initial identification of a research gap and the formulation of a novel idea, through literature search, code implementation, and the design of analytical experiments, to the final writing of the research papers—was performed by the LLM-based system. The final research papers generated through this autonomous process are provided in Appendix F.

The role of the human authors was strictly limited to supervision, verification, and calibration of the system. We provided the initial SOTA methods as a starting point, monitored the system's progress, and verified the correctness of the final reported results. However, all novel scientific ideas, code, analyses, and written text were generated by the LLMs.

## B    HUMAN EXPERT REVIEW

### B.1    REVIEW PROCESS AND CRITERIA

To ensure a rigorous and impartial evaluation of the generated papers, we convened a small, dedicated program committee. The committee was composed of two active researchers who served as volunteer reviewers for ICLR 2025, and one senior researcher who had previously been invited to serve as an ICLR Area Chair. All committee members possess significant expertise in the field of Large Language Models. The entire review process, with the exception of a rebuttal phase, was designed to meticulously emulate the official standards of ICLR 2025. Each of the five papers generated by our system was assigned to the three reviewers for a thorough and independent assessment. The average review time for each paper was 55 minutes, during which reviewers were required to provide not only scores but also detailed written feedback, including a summary of the paper's strengths and weaknesses.

The evaluation was conducted on a custom-deployed review website where reviewers could not see each other's scores or feedback, ensuring that all initial assessments were made independently. The review form was structured to gather concise yet comprehensive feedback. First, reviewers were asked to state their **Confidence** in their review on a scale of 1 to 5. The core of the evaluation consisted of three sub-scores, each rated on a 1 to 4 scale: **Soundness**, assessing the technical correctness and experimental rigor; **Presentation**, evaluating the clarity and quality of the writing;

and **Contribution**, measuring the significance and novelty of the work. Finally, reviewers provided a holistic **Rating** on a scale of 1 to 10, where a score of 5 represented a 'borderline reject' and a score of 6 represented a 'borderline accept'.

After the three reviewers submitted their independent evaluations for a paper, the volunteer acting as Area Chair would then read all submitted reviews. Drawing upon their experience from the ICLR review process, the Area Chair synthesized the feedback, weighed the arguments presented by the reviewers, and made a final executive decision on whether the paper should be accepted or rejected in the context of our study. This final decision was recorded as the definitive outcome for each paper's evaluation.

### B.2 SUMMARY OF REVIEWER FEEDBACK

Across the five generated papers, a clear consensus emerged from the human reviewers: Deep-Scientist consistently excels at the ideation stage of research. The committee unanimously lauded the methods for their genuine novelty and tangible contributions, noting that each paper proposed a unique approach that meaningfully advanced the state-of-the-art in its respective subfield. This feedback validates the system's core strength as a powerful engine for identifying relevant research gaps and generating innovative, impactful solutions, confirming that it can successfully ideate beyond mere incremental improvements.

However, this strength in ideation was systematically undermined by a recurring pattern of weaknesses in scientific execution and rigor. The most critical and frequent concern was a lack of empirical soundness; reviewers consistently noted that DeepScientist failed to design comprehensive validation plans, citing insufficient evaluation on standard benchmarks and a lack of in-depth analytical experiments (e.g., ablations, motivation studies) to justify its claims. This was compounded by a failure to properly contextualize its contributions, with papers often omitting comparisons to essential baselines or failing to discuss closely related work, thereby weakening the perceived significance of the results.

This feedback pinpoints the primary bottleneck in current autonomous systems: a profound gap between the ability to generate novel concepts and the capacity for rigorous scientific execution and articulation. The observed weaknesses in experimental design directly reflect the low-success-rate problem discussed previously; the system struggles not just to implement ideas correctly, but to validate them convincingly. To bridge this gap, future work must endow these systems with a deeper, procedural understanding of the scientific method itself. This requires moving beyond simple implementation and reporting capabilities towards two key areas: First, developing agents explicitly trained in experimental design, capable of planning comprehensive evaluations that anticipate and address potential scientific critiques. Second, enhancing the system's ability for analytical reasoning, enabling it to not just describe results but to interpret their significance, formulate compelling arguments, and engage in the kind of deep, reflective discussion that characterizes high-impact research.

## C ADDRESSING THE BOTTLENECKS IN AUTONOMOUS SCIENTIFIC DISCOVERY

Artificial intelligence is reshaping the paradigm of scientific exploration through its ability to generate hypotheses at a massive scale; however, this has also pushed "verification" to the center stage, making it a critical bottleneck. Our research empirically reveals the severity of this challenge: on frontier scientific tasks, the success rate of ideas generated by AI systems that ultimately lead to substantial progress is typically below 3%, meaning the vast majority of computational resources are consumed exploring low-value hypotheses. This inefficient "needle in a haystack" model is the core obstacle preventing AI Scientists from evolving from "novel tools" to "efficient discoverers." (Cornelio et al., 2025) Therefore, to further accelerate the process of scientific discovery, future research must focus on constructing a systematic solution to overcome this bottleneck. As shown in Figure 7, future AI Scientist systems need to evolve synergistically in three key directions: optimizing the quality of initial hypotheses (Optimize Hypothesis Quality), enhancing filtering capabilities during the process (Enhance Filtering), and improving the quality of implementation and verification at the final stage (Improve Implementation Quality).

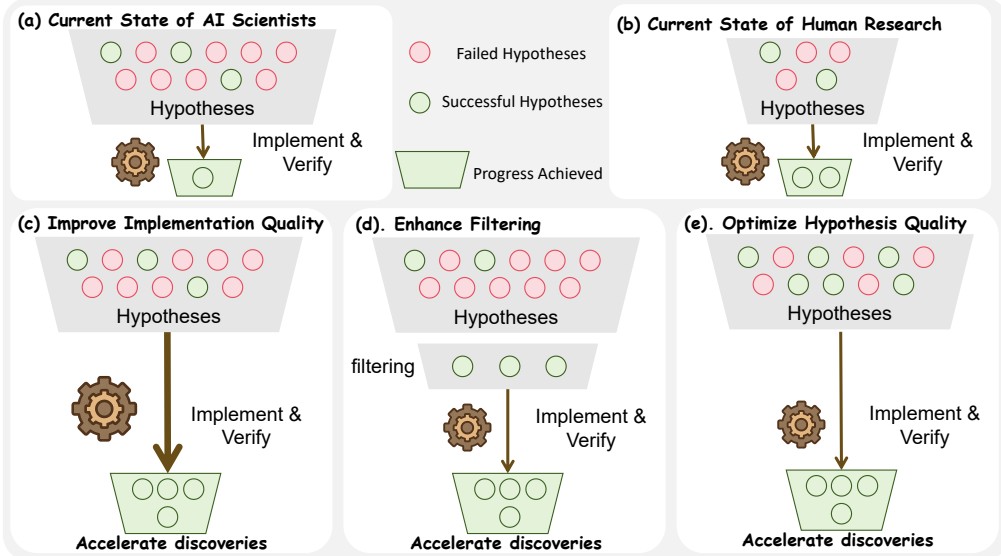

Figure 7: Three strategies for improving the efficiency of autonomous scientific discovery. (a) and (b) illustrate the low success rate currently faced by both AI and human research. Future directions will need to accelerate the discovery process through the synergy of three approaches: (c) improving implementation success rates, (d) adding an efficient filtering stage before implementation, and (e) optimizing the quality of initial hypotheses from the source.

One of the core future research directions is to develop AI systems capable of generating higher-quality, more reliable hypotheses (as shown in Figure 7e), equipped with more precise filtering mechanisms to predict their success rate (as shown in Figure 7d). Methods that rely purely on a data-driven approach, while capable of discovering patterns, often produce outputs that lack a theoretical foundation and are prone to generating "hallucinations" that contradict known scientific theories. Future systems must move beyond this by more deeply integrating background knowledge and theory. For instance, the direction represented by "derivable models" (such as AI-Descartes (Cornelio et al., 2023) and AI-Hilbert (Cory-Wright et al., 2024)), which incorporate scientific axioms as constraints during the hypothesis generation phase, offers a promising path to improving hypothesis quality. Furthermore, systems must have the ability to learn from their own exploratory history. By establishing mechanisms similar to a "Findings Memory," a system can systematically record and analyze every success and failure, thereby avoiding redundant exploration of ineffective paths in subsequent iterations and gradually developing a more insightful scientific intuition. Building on this foundation, developing more advanced, low-cost surrogate models and acquisition functions to more accurately predict the scientific value of an idea will be key to enhancing filtering efficiency and conserving verification resources.

Concurrently, an often-overlooked yet crucial future research direction is to significantly improve the quality and reliability of AI systems in the engineering implementation and verification stages (as shown in Figure 7c). Even the most brilliant scientific concept can never have its value confirmed if it cannot be accurately and flawlessly translated into an executable experiment. Our analysis indicates that up to 60% of exploratory failures stem from implementation-level errors, which represents a massive waste of resources and directly impedes scientific progress. History has repeatedly warned us that a lack of rigorous verification can lead to catastrophic consequences, whether in NASA missions or medical practice. Therefore, building a scalable and reliable automated verification platform is an essential path forward. This requires not only more powerful code-generation and self-debugging agents to reduce implementation errors but also standardized sandbox environments and automated testing procedures to ensure the stability and reproducibility of experimental results. Ensuring the absolute reliability of the verification process is the final and most critical line of defense in transforming AI-generated "plausible ideas" into "solid scientific evidence."

Looking ahead, to truly accelerate scientific discovery, it is necessary to integrate the aforementioned strategies into an organic whole, advancing AI Scientists from "random explorers" to "goal-oriented strategists." This is not about replacing humans with AI, but about pioneering a more efficient paradigm of human-AI collaboration. In this model, human scientists are responsible for defining grander, more valuable scientific goals and providing high-level strategic guidance, while the AI system serves as a powerful "exploration engine," executing efficient trial-and-error and verification cycles at an unprecedented scale and speed under human direction. To realize this vision, the community must also address a series of challenges, such as building benchmarks that can truly evaluate innovation and designing mechanisms that encourage diverse exploration to avoid the homogenization of research paradigms, thereby preserving the potential for serendipitous discoveries like Alexander Fleming's discovery of penicillin (Fleming, 1941).

# D    METHOD DETAILS

## D.1    IMPLEMENTATION OF THE STRATEGIZE & HYPOTHESIZE STAGE

The Strategize & Hypothesize stage of each discovery cycle is operationalized within our system as a multi-agent workflow that mirrors a structured research and development process. This entire process is centered around the **'Findings Memory'** ($\mathcal{M}_t$), which is implemented as a large, list-style database (`IdeaDatabase` in the codebase) designed to persistently store thousands of structured records. Each record represents a unique finding at a specific stage of its lifecycle. The workflow is executed by a cohort of specialized agents, orchestrated by a central **'DirectorAgent'**, ensuring that the generation of new hypotheses is a guided, strategic exploration rather than an undirected search.

Each cycle commences with an analysis of the **'Findings Memory'**. This is enacted by the **'ScientistAgent'**, which is invoked in its 'STAGE: RESEARCH_OUTLINE' mode. The agent consumes the current state of knowledge, including the entire baseline codebase and the contents of the 'Findings Memory', to perform a first-principles analysis of the problem domain. It formulates the core challenge as a rigorous mathematical problem and identifies the fundamental limitations of existing findings. The tangible output of this strategic analysis is a comprehensive Markdown document, `Research_Outline.md`, which serves as a high-level charter, guiding the system's focus for the current cycle. This initial step provides a concrete and reproducible mechanism for the "analysis of limitations in existing knowledge" described in our main methodology.

Following the establishment of a strategic outline, the 'DirectorAgent' deploys one or more instances of the 'ExplorerAgent' to generate a new collection of hypotheses ($\mathcal{P}_{new}$). Governed by its highly specific `EXPLORER_AGENT_PROMPT`, this agent's core function is to produce novel, structured records based on the directions provided in `Research_Outline.md`. Its methodology emphasizes systematic, cross-disciplinary investigation, using integrated research tools like `pasa_search` to adapt successful theoretical frameworks from adjacent scientific fields. The output of this process is a structured JSON object for each new hypothesis, detailing its motivation, theoretical underpinnings, and a concrete implementation plan. This JSON object is the system's direct instantiation of a candidate finding $I \in \mathcal{P}_{new}$.

Upon generation, each new candidate finding is immediately passed to the system's low-cost Surrogate Model ($g_t$), a role fulfilled by the 'EvaluatorAgent'. This agent is first contextualized with the entire state of the 'Findings Memory'. As dictated by its prompt, it then assesses the candidate finding and produces three numerical scores: a utility_score, a quality_score, and an exploration_score. These scores are the direct implementation of the components of the Valuation vector $V = \langle v_u, v_q, v_e \rangle$, quantifying the hypothesis's estimated value. Finally, the 'DirectorAgent' initializes a new record in the 'Findings Memory', pairing the new hypothesis with its valuation vector and assigning it the status of an 'Idea Finding'. This action concludes the stage, formally adding the new, unevaluated hypotheses to the system's knowledge base, ready for the subsequent selection and verification phase.

## D.2    IMPLEMENTATION OF THE IMPLEMENT & VERIFY STAGE

The Implement & Verify stage serves as the primary filter in the research funnel and is operationalized as the "Engineering Phase" of the system's workflow. This phase is triggered when the system makes a strategic decision to commit significant computational resources to validate a single, highly

promising 'Idea Finding'. The workflow is managed by the 'ScientistAgent', which acts as the primary decision-maker, and the 'ImplementationAgent', which executes the complex code-level modifications and real-world experiments. This stage is paramount as it provides the empirical feedback essential for the system's learning loop.

The selection process, described in the main text as the Acquisition Function ($\alpha$), is implemented by the 'ScientistAgent' operating in its 'STAGE: STRATEGIC_DECISION' mode. This agent periodically reviews the population of 'Idea Findings' and their associated valuation vectors stored in the 'Findings Memory'. Based on the criteria in its prompt, which require it to reflect on past outcomes and balance the exploitation-exploration trade-off, it selects the most promising record for validation. A decision of 'VALIDATE' from this agent, targeting a specific finding's ID, is the system's operational equivalent of the 'argmax' operation in the UCB formula. Upon this selection, the chosen record's status within the 'Findings Memory' is formally promoted from an 'Idea Finding' to an 'Implement Finding', signaling the start of the resource-intensive implementation.

Once a finding is promoted, the 'DirectorAgent' delegates the implementation task to the 'ImplementationAgent', a specialized, terminal-based Agent governed by the highly prescriptive `IMPLEMENTATION_AGENT_PROMPT`. The agent's workflow is meticulously structured for robustness and reproducibility. It begins by creating an isolated, sandboxed directory for the experiment by copying the entire baseline codebase. Within this safe environment, it systematically translates the 'theory_and_method' and 'code_level_plan' from the selected finding's record into functional code modifications. The agent's governing prompt mandates a rigorous engineering process, including the creation of unit tests to verify numerical stability and correctness, and the maintenance of a detailed log of all actions in a `notebook.md` file.

The culmination of this stage is the real-world experiment. After completing the code modifications and passing all internal tests, the 'ImplementationAgent' executes the project's standard evaluation script, `test.sh`. The complete, captured terminal output from this script, containing the final performance metrics, constitutes the empirical observation of the true scientific value function, $f(I_{t+1})$. In the final step, the 'DirectorAgent' takes this new data point—the empirical result—and uses it to update the corresponding 'Implement Finding' record in the **'Findings Memory'**. This action enriches the finding with empirical evidence, formally closing the learning loop and providing critical new knowledge to inform all subsequent discovery cycles.

### D.3 IMPLEMENTATION OF THE ANALYZE & REPORT STAGE

The final stage of the DeepScientist discovery loop, Analyze & Report, is initiated when an `Implement Finding` successfully validates, demonstrating performance that surpasses the established baseline. This achievement triggers a sophisticated multi-agent workflow orchestrated by a central 'DirectorAgent', designed to transform the raw experimental success into a comprehensive and reproducible scientific paper. This process is not a monolithic writing task but a structured, multi-phase procedure that mirrors a rigorous human-led research publication effort, comprising three core sub-phases: Iterative Outline Development, Sequential Paper Writing, and Multi-Round Revision. Each sub-phase is executed by specialized agents with precisely defined roles and operational protocols, ensuring a high degree of quality control and methodological soundness.

The process begins with Iterative Outline Development, a three-round cycle of design, review, and analysis. The 'DirectorAgent' first deploys an 'OutlineDesignerAgent', which is tasked with creating a compelling narrative and a detailed structural blueprint for the paper. This agent operates via a unique two-stage process: it first generates thousands of words of unstructured reasoning to explore the theoretical foundations and experimental implications of the finding, drawing from the complete history in the `Findings Memory` and the newly populated `Result.md` file. Subsequently, it distills this reasoning into a structured JSON object, which includes a narrative arc, answers to ten foundational research questions, and detailed plans for every table and figure. This initial outline is then passed to an 'OutlineReviewerAgent', which provides a harsh, academic-style critique. Finally, an 'OutlineAnalyzerAgent' evaluates both the outline and its review to make a strategic decision: either 'VALIDATE' the outline as ready, or 'EVOLVE' it by generating a specific 'improvement_directive' for the next round. This cycle repeats up to three times, ensuring the final blueprint, saved as `final_selected_outline.json`, is robust and logically sound.

With a validated outline in place, the 'DirectorAgent' proceeds to the Sequential Paper Writing sub-phase, deploying a specialized 'ClaudeCodePaperWriteAgent'. This agent is governed by an exceptionally detailed and prescriptive prompt that enforces a strict, multi-phase workflow executed directly on the file system. Critically, the agent does not immediately begin writing prose. Its first mandatory step is an extensive literature review, where it uses integrated search tools like 'pasa_search' and 'semantic_scholar_query' to gather over 60 relevant citations, meticulously populating a `references.bib` file and documenting its findings in a `draft.md` log. Only after this literature foundation is established does it proceed to generate all required figures and tables, extracting data directly from `Result.md` and saving styled plots to the '/figures' directory.

Following the completion of literature and figure generation, the 'ClaudeCodePaperWriteAgent' begins writing the manuscript's content. It follows a strict top-down sequence, creating and populating individual LaTeX files for each section (e.g., `introduction.tex`, `methodology.tex`, `experiments.tex`) in a predefined order. The content for each section is precisely guided by the blueprint in `final_selected_outline.json`, ensuring perfect alignment between the plan and the final output. The agent's prompt includes a comprehensive validation checklist that it must internally satisfy before completion, covering everything from content authenticity and structural integrity to experimental completeness and citation accuracy. The entire writing process is logged in `writing_plan.md` and `draft.md`, and the agent signals its completion to the 'DirectorAgent' only by creating a final `paper.md` file, ensuring the full sequence has been executed.

The final sub-phase is Multi-Round Revision, which ensures the paper meets publication standards. The 'DirectorAgent' deploys a 'PaperReviewerAgent' to conduct a thorough review of the complete draft, assessing its clarity, technical accuracy, and narrative coherence. The reviewer's structured feedback is then passed back to the 'ClaudeCodePaperWriteAgent' as a set of revision instructions. The writer agent then performs a targeted revision of the relevant `.tex` files to address the identified weaknesses. This review-and-revise loop is executed for a predefined number of rounds, iteratively polishing the manuscript. The culmination of this entire stage is a complete, publication-ready package containing the full LaTeX source code, section files, bibliography, figures, and a detailed log of the generation process, thereby converting a single 'Progress Finding' into a durable and shareable piece of scientific knowledge.

## E  ADDITIONAL EXPERIMENTS

### E.1  LARGE-SCALE EVALUATION OF MICRONANO-DEEPSCIENTIST ON THE ALGOTUNE BENCHMARK

We introduced a lightweight variant of our framework, **Micronano-DeepScientist**, to enable large-scale evaluation across diverse scientific discovery tasks (Press et al., 2025). This version preserves the core hierarchical exploration process and the discovery memory mechanism of DeepScientist, but removes the most computation-intensive modules such as literature reading, formal hypothesis drafting, and extensive experimental analysis. As a result, Micronano-DeepScientist operates at approximately **1/1000** of the runtime cost of the full system while maintaining its essential exploratory capability. All experiments in this section use the **open-source GLM-4.6** (Zeng et al., 2025) model as the reasoning backbone.

To assess the generality of our approach, we conducted systematic experiments on ALGOTUNE, a benchmark containing **154 algorithmic discovery tasks** spanning mathematics, physics, and computer science. Each task requires the system to autonomously search for algorithmic improvements relative to strong human-designed baselines. Micronano-DeepScientist successfully discovered algorithms that outperform the baseline implementations on **120 tasks (77.9%)**, achieving an average speedup of **16.6×**. On the remaining 34 tasks (22.1%), the system generated solutions that were slower or did not surpass the baseline within the allocated search time. These results demonstrate that even a significantly scaled-down discovery engine can autonomously generate competitive algorithmic innovations across a broad task distribution when paired with an efficient hierarchical search strategy and a capable open-source model such as GLM-4.6. A summary of the aggregate statistics is shown in Table 4.

Table 4: Aggregate performance of Micronano-DeepScientist on the 154-task ALGOTUNE benchmark.

| Metric | Value |
|---|---|
| Total tasks | 154 |
| Successful tasks | 120 (77.9%) |
| Slower or failed tasks | 34 (22.1%) |
| Mean speedup | 16.6× |
| LLM backbone | GLM-4.6 (open-source) |

Table 5: Automatic review scores using the o3-mini reviewer setup from Zochi. DeepScientist is evaluated with exactly the same code and prompts as prior work.

| Systems | Sound. | Pres. | Contr. | Orig. | Qual. | Clar. | Sign. | Overall |
|---|---|---|---|---|---|---|---|---|
| AI Scientist | 2.20 | 2.40 | 2.10 | 2.40 | 2.10 | 2.60 | 2.20 | 3.80 |
| AI Scientist-v2 | 2.00 | 1.67 | 2.00 | 2.00 | 2.00 | 1.67 | 2.00 | 3.00 |
| CycleResearcher | 2.33 | 2.17 | 2.33 | 2.33 | 2.17 | 2.17 | 2.50 | 4.00 |
| Zochi | 3.00 | 3.00 | 3.00 | 3.00 | 3.00 | 2.50 | 3.00 | 6.00 |
| AI-Researcher | 2.43 | 2.14 | 2.43 | 2.71 | 2.43 | 2.14 | 2.57 | 4.29 |
| DeepScientist | 2.80 | 2.80 | 2.80 | **3.00** | 2.80 | **2.80** | **3.60** | **6.20** |

## E.2 ROBUSTNESS ACROSS MULTIPLE AUTOMATED REVIEWER SYSTEMS

Beyond DeepReviewer-14B'(Zhu et al., 2025a), we further assess the quality of DeepScientist's papers using several independent automatic reviewer systems. First, we adopt the original Zochi reviewer setup based on the o3-mini model and the official evaluation code, and re-evaluate all available systems under exactly the same prompts (Table 5). In this configuration, Zochi (Intology, 2025) attains a strong Overall score of 6.00, but DeepScientist still slightly surpasses it with an Overall score of 6.20, and achieves the highest or tied-highest scores on key dimensions such as Originality, Clarity, and Significance. We then apply the AI Scientist reviewer prompts with three different backbone models—Gemini-2.5-Pro, GPT-4o, and GPT-5—yielding Tables 6, 7, and 8, respectively. Across all three backbones, DeepScientist again obtains the best Overall rating among the compared AI Scientist systems, with noticeable gains in Soundness and Presentation under Gemini-2.5-Pro, and a particularly large margin in Overall under GPT-4o. Finally, using the independent CycleReviewer model [3,4] (Table 9), DeepScientist achieves the highest Overall score of 4.85, exceeding both Zochi (4.50) and CycleResearcher (4.46) while also leading on all three component criteria.

Taken together, these results show a consistent pattern: regardless of the underlying reviewer architecture (o3-mini, Gemini-2.5-Pro, GPT-4o, GPT-5, or CycleReviewer) and despite differences in absolute scoring scales, DeepScientist is always ranked at or near the top in Overall quality among existing AI Scientist systems. This cross-validation strengthens the robustness of our conclusions and indicates that DeepScientist's advantages are not an artifact of a particular reviewer model or prompt design. Moreover, the dimensions on which DeepScientist tends to score highest—such as originality, significance, and clarity—are precisely those emphasized by human program-committee evaluations in Section B.2, suggesting that the gains observed under automatic reviewers are aligned with human judgments of scientific value.

## E.3 A CASE STUDY ON HUMAN VS. AUTONOMOUS RESEARCH EFFICIENCY

To better understand how an autonomous system compares to human researchers in terms of research efficiency, we conduct a qualitative case study on the AI text detection task. For this domain, we collected approximate statistics from the teams behind two recent human-designed SOTA methods (Fast-Detect and Glimpse), focusing on their development timelines and resource usage. Each project was led by a full research team and typically required about six months from project inception to camera-ready paper. During this period, the teams reported using roughly 5–10 GPU hours per day on average, for a total of around 1,500 GPU hours per project, with GPU utilization dropping substantially outside of normal working hours. In terms of exploration breadth, a typical six-month

Table 6: Automatic review scores using the AI Scientist reviewer prompts with Gemini-2.5-Pro.

| Systems | Sound. | Pres. | Contr. | Overall |
|---|---|---|---|---|
| AI Scientist | 1.00 | 1.40 | 1.10 | 1.70 |
| AI Scientist-v2 | 1.00 | 1.00 | 1.00 | 1.67 |
| CycleResearcher | 1.00 | 1.00 | 1.17 | 1.50 |
| AI-Researcher | 1.00 | 1.14 | 1.00 | 1.29 |
| Zochi | 1.00 | 1.50 | 2.00 | 2.00 |
| DeepScientist | **1.20** | **1.80** | 1.80 | **2.20** |

Table 7: Automatic review scores using the AI Scientist reviewer prompts with GPT-4o.

| Systems | Sound. | Pres. | Contr. | Overall |
|---|---|---|---|---|
| AI Scientist | 2.00 | 2.10 | 2.00 | 2.60 |
| AI Scientist-v2 | 2.00 | 2.00 | 2.00 | 3.00 |
| CycleResearcher | 2.00 | 2.00 | 2.00 | 3.00 |
| AI-Researcher | 2.00 | 2.00 | 2.00 | 3.14 |
| Zochi | 2.00 | 2.00 | 2.50 | 3.00 |
| DeepScientist | **2.40** | **2.20** | 2.40 | **4.20** |

project allowed the team to deeply investigate on the order of 10–30 core hypotheses, each of which required careful design, implementation, and iteration before being deemed publishable.

On the same AI text detection task, DeepScientist was run continuously for 14 days and produced three progressively stronger SOTA methods (T-Detect, TDT, and PA-TDT). Each breakthrough consumed roughly 900 GPU hours, for a total on the order of a few thousand GPU hours, but these resources were utilized close to 24/7 across parallel instances. Within this two-week window, the system generated over 2,400 candidate hypotheses and autonomously executed around 600 full experimental validations, far exceeding the exploratory throughput that a human team can typically achieve in a comparable or even longer time span. While such a comparison is necessarily approximate and limited in sample size, it suggests that, under a given compute budget, DeepScientist can explore the hypothesis space with a substantially higher trial-and-error throughput, compressing what would conventionally require several months of human-led research into a few weeks of machine-driven exploration. At the same time, human researchers remain essential for problem formulation, high-level evaluation, and long-term research direction, indicating a complementary relationship in which autonomous systems amplify, rather than replace, human scientific effort.

## F  IMPLEMENTATION DETAILS

Our implementation relies on a distributed architecture to manage the distinct tasks of scientific reasoning and code execution. The core logic of DeepScientist is powered by the Gemini-2.5-pro model, while all code implementation tasks are delegated to Claude-4-opus, executed within the Claude Code framework (v1.0.53). To ensure stability and security, the DeepScientist system and the Claude Code agent are isolated in separate Docker containers, communicating via a port-based API. During the 'Implement & Verify' stage, a human-verified baseline code repository is first duplicated into a new, sandboxed folder. The Claude Code agent's operations are strictly confined to this new directory to prevent unintended modifications. A critical step in our pipeline is a secondary verification process: after Claude Code reports completion, DeepScientist independently re-executes the main script via the command line. This measure was implemented to counteract a high rate of false positives—we observed that approximately 50% of initial implementation attempts failed to complete fully due to internal timeouts within the Claude Code agent. Throughout this project, all experimental results were manually inspected by human supervisors to guarantee their authenticity. For the 'Analyze & Report' stage, a similar process is followed: the validated code is replicated for each analytical experiment, with Claude Code executing them sequentially. Upon completion, DeepScientist aggregates all results, generates a paper outline, and then employs automated tools to write and compile the final PDF manuscript. **For all experiments, we used a fixed set of hyperpa-**

Table 8: Automatic review scores using the AI Scientist reviewer prompts with GPT-5.

| Systems | Sound. | Pres. | Contr. | Overall |
|---|---|---|---|---|
| AI Scientist | 1.00 | 1.30 | 1.00 | 2.10 |
| AI Scientist-v2 | 1.00 | 1.00 | 1.00 | 2.00 |
| CycleResearcher | 1.00 | 1.00 | 1.00 | 1.67 |
| AI-Researcher | 1.00 | 1.00 | 1.00 | 1.86 |
| Zochi | 1.00 | 1.50 | 1.50 | 2.50 |
| DeepScientist | **1.40** | **1.60** | **1.80** | **3.00** |

Table 9: Automatic review scores using CycleReviewer.

| Systems | Sound. | Pres. | Contr. | Overall |
|---|---|---|---|---|
| AI Scientist | 2.12 | 2.25 | 2.00 | 3.23 |
| AI Scientist-v2 | 2.00 | 2.42 | 2.00 | 3.00 |
| CycleResearcher | 2.50 | 2.54 | 2.38 | 4.46 |
| AI-Researcher | 2.07 | 2.21 | 2.07 | 3.50 |
| Zochi | 2.62 | 2.88 | 2.50 | 4.50 |
| DeepScientist | **2.80** | **2.85** | **2.65** | **4.85** |

**rameters:** the retrieval count was set to $K = 15$, and the UCB parameters were set to utility weight $w_u = 1$, quality weight $w_q = 1$, and exploration coefficient $\kappa = 1$.

The financial and computational costs of this autonomous discovery process are substantial. Each idea generated during the 'Strategize & Hypothesize' stage incurred an approximate cost of \$5 in API calls. For each attempt in the 'Implement & Verify' stage, the cost averaged \$20 for Claude-4-opus API usage, in addition to the computational cost of approximately 1 GPU hour, as detailed in Figure 4.c. A successful finding that progressed to the 'Analyze & Report' stage required a further expenditure of around \$150, which includes \$100 for running analytical experiments and \$50 for the final report generation. The total cost to achieve the scientific advancements presented in this paper amounted to approximately \$100,000. While significant, we believe these costs can be substantially reduced. We recommend that future iterations explore more economical alternatives, such as deploying high-throughput models like Qwen-3-Next-80B for the core DeepScientist system and leveraging subscription-based API access (e.g., Claude Max or OpenAI Pro) to mitigate per-call expenses. In this paper, each implementation was provided with a single H800 server for exploration. Since the H800 GPU has an FP16 computing power of approximately 2 TFLOPS, an average execution of 70 minutes corresponds to about $1 \times 10^{16}$ floating-point operations.

# ABDUCT, ACT, PREDICT:
# SCAFFOLDING CAUSAL INFERENCE FOR AUTOMATED FAILURE ATTRIBUTION IN MULTI-AGENT SYSTEMS

**DeepScientist**

## ABSTRACT

Failure attribution in multi-agent systems—pinpointing the exact step where a decisive error occurs—is a critical yet unsolved challenge. Current methods treat this as a pattern recognition task over long conversation logs, leading to critically low step-level accuracy (below 17%), which renders them impractical for debugging complex systems. Their core weakness is a fundamental inability to perform robust counterfactual reasoning: to determine if correcting a single action would have actually averted the task failure. To bridge this *counterfactual inference gap*, we introduce **Abduct-Act-Predict (A2P) Scaffolding**, a novel agent framework that transforms failure attribution from pattern recognition into a structured causal inference task. A2P explicitly guides a large language model through a formal three-step reasoning process within a single inference pass: (1) **Abduction**, to infer the hidden root causes behind an agent's actions; (2) **Action**, to define a minimal corrective intervention; and (3) **Prediction**, to simulate the subsequent trajectory and verify if the intervention resolves the failure. This structured approach leverages the holistic context of the entire conversation while imposing a rigorous causal logic on the model's analysis. Our extensive experiments on the Who&When benchmark demonstrate its efficacy. On the Algorithm-Generated dataset, A2P achieves **47.46%** step-level accuracy, a **2.85×** improvement over the 16.67% of the baseline. On the more complex Hand-Crafted dataset, it achieves **29.31%** step accuracy, a **2.43×** improvement over the baseline's 12.07%. By reframing the problem through a causal lens, A2P Scaffolding provides a robust, verifiable, and significantly more accurate solution for automated failure attribution.

## 1 INTRODUCTION

The rise of sophisticated multi-agent systems marks a pivotal moment in artificial intelligence, unlocking new frontiers in collaborative problem-solving (Li et al., 2023; Hong et al., 2023) and complex task automation (Wu et al., 2023; Fourney et al., 2024). However, this growing complexity introduces a critical operational bottleneck: debugging. When a system fails, developers are faced with a tangled web of interactions, where a subtle error in an early step can cascade into a catastrophic failure dozens of turns later. Pinpointing the single, decisive error—the task of **failure attribution**—is not merely challenging; it is a labor-intensive, error-prone process that stands as a major barrier to the reliable deployment and iterative improvement of these powerful systems (Zhang et al., 2025).

Current automated approaches to this problem have proven fundamentally inadequate, with step-level accuracy rates hovering below a dismal 17% (Zhang et al., 2025), a figure far too low for practical debugging. We argue this failure is not a matter of model capability but of methodological paradigm. Existing methods treat failure attribution as a **pattern recognition** task over conversational logs (Zhang et al., 2025; Lightman et al., 2023). They present an entire log to a Large Language Model (LLM) and ask it to "find the mistake," implicitly assuming the model can spot anomalous patterns correlated with failure. This approach fundamentally misses the point. The critical question is not "which step looks wrong?" but rather a causal one: "which single corrective action would have turned failure into success?" This exposes a deep *counterfactual inference gap*: the inability of unstructured, holistic methods to systematically reason about the consequences of hypothetical interventions, a challenge particularly pronounced in multi-turn interactions where cause and effect are obscured (Kıcıman et al., 2023; Zevcevic et al., 2023).

To bridge this gap, we introduce **Abduct-Act-Predict (A2P)**, a novel prompting framework that reframes failure attribution from pattern recognition into a structured **causal inference** task. Instead of asking for a direct answer, A2P guides an LLM through a formal, three-step counterfactual reasoning process within a single inference pass, operationalizing the logic of Pearl's structural causal model hierarchy (Pearl, 2009). The framework compels the model to: (1) *Abduct*, inferring hidden factors (e.g., a flawed assumption) that explain a problematic action; (2) *Act*, defining a minimal, concrete corrective intervention; and (3) *Predict*, simulating the subsequent counterfactual trajectory to verify if the intervention would have resolved the overall task failure. This structured process forces the model to move beyond correlation and rigorously test causal hypotheses, transforming the "needle-in-the-haystack" problem (Liu et al., 2024) into a systematic investigation.

Our approach is not just theoretically sound but empirically dominant. Evaluated on the comprehensive Who&When benchmark (Zhang et al., 2025), A2P Scaffolding achieves a step-level accuracy of **47.46%** on the Algorithm-Generated dataset—a **2.85×** improvement over the 16.67% of its direct baseline. On the more challenging Hand-Crafted dataset, it achieves **29.31%** accuracy, a **2.43×** improvement over the baseline's 12.07%. These results establish a new state-of-the-art and, for the first time, demonstrate a viable path toward reliable automated debugging for multi-agent systems. Rigorous ablation studies further validate our framework, confirming that each causal reasoning component is essential and revealing the surprising, critical role of structural cues like contextual step numbering in enabling fine-grained analysis.

## 2 RELATED WORK

### 2.1 LLM MULTI-AGENT SYSTEMS

The emergence of Large Language Models as capable reasoning agents has catalyzed rapid development in multi-agent system architectures (Wu et al., 2023; Li et al., 2023; Hong et al., 2023). These systems leverage the collaborative potential of multiple specialized agents working together to solve complex tasks that exceed the capabilities of individual models (Park et al., 2023; Liu et al., 2023b). Notable frameworks include AutoGen (Wu et al., 2023), which facilitates multi-agent conversations through customizable agent roles and interaction patterns, CAMEL (Li et al., 2023), which explores role-playing dynamics in collaborative task-solving, and MetaGPT (Hong et al., 2023), which incorporates software development methodologies into multi-agent workflows. Recent work has expanded these foundations to include specialized domains such as scientific research (Ghafarollahi & Buehler, 2024), software development (Kumar et al., 2024), and complex reasoning tasks (Du et al., 2023). However, as these systems grow in sophistication, the challenge of diagnosing failures becomes increasingly complex, with current debugging approaches remaining largely manual and ad-hoc (Wang et al., 2024b). The need for automated failure attribution becomes particularly acute in production deployments where system reliability directly impacts user experience and operational efficiency (Fourney et al., 2024).

The rapid proliferation of multi-agent systems has outpaced the development of systematic debugging methodologies. While considerable effort has been invested in designing agent architectures and interaction protocols (Qian et al., 2023; Chen et al., 2024), relatively little attention has been paid to post-hoc failure analysis. This gap is particularly problematic given the emergent behaviors that arise from agent interactions, where system failures often result from subtle cascading effects rather than obvious individual errors (Kumar et al., 2024). Our work addresses this critical gap by providing the first systematic framework for automated failure attribution specifically designed for the unique challenges of multi-agent system debugging. Unlike previous approaches that focus on system design or performance evaluation (Wang et al., 2024b), we concentrate on the diagnostic phase that enables iterative improvement and reliable deployment.

### 2.2 LLM-AS-A-JUDGE AND PROCESS-LEVEL EVALUATION

The paradigm of using LLMs as evaluators has gained significant traction as a scalable alternative to human assessment across diverse domains (Zheng et al., 2023; Wang et al., 2024a). This approach has proven particularly valuable in scenarios where human evaluation is expensive, time-consuming, or requires specialized expertise (Liu et al., 2023a; Dubois et al., 2023). Recent developments have extended LLM-based evaluation to process-level assessment, where models evaluate intermediate

reasoning steps rather than only final outputs (Lightman et al., 2023; Wang et al., 2023). Process reward models (Uesato et al., 2022) have shown promise in mathematical reasoning by identifying the specific steps where errors occur, enabling more targeted feedback and improvement strategies. However, these approaches primarily focus on single-agent reasoning chains in well-defined domains like mathematics or coding, where the correctness of individual steps can be objectively determined.

Our work extends this process-level evaluation paradigm to the significantly more complex domain of multi-agent system failures. Unlike mathematical reasoning where step correctness is often binary and context-independent, multi-agent failures involve complex interdependencies between agents, temporal dynamics, and emergent behaviors that resist simple classification (Du et al., 2023). While process reward models evaluate individual reasoning steps, our A2P framework must navigate the multi-participant, interactive dynamics of agent systems where the "correctness" of an action depends heavily on the broader conversational context and the ultimate task outcome. This fundamental difference necessitates our novel approach of structured counterfactual reasoning rather than step-by-step correctness assessment (Miller, 2019; Doshi-Velez & Kim, 2017).

## 2.3 CAUSAL REASONING IN LLMS

Recent research has begun exploring the causal reasoning capabilities of large language models, revealing both promising potential and significant limitations (Kıcıman et al., 2023; Zevcevic et al., 2023). Benchmarks such as CLadder (Qin et al., 2023) and CausalBench (Jin et al., 2023) have established that while LLMs can perform certain types of causal reasoning, they often struggle with complex counterfactual inference tasks that require systematic manipulation of causal variables (Jin et al., 2024). This limitation is particularly pronounced in scenarios requiring what Pearl terms "Level 3" causal reasoning, answering questions about what would have happened under different circumstances (Pearl, 2009). Studies have shown that structured prompting approaches, such as CausalCoT (Zhang et al., 2024), can significantly enhance LLM performance on causal tasks by providing explicit reasoning frameworks that guide model inference.

Building on these insights, our A2P Scaffolding framework represents a practical application of structured causal prompting to a real-world diagnostic task. While previous work has focused on synthetic causal reasoning benchmarks or simplified scenarios (Jin et al., 2023; Qin et al., 2023), we tackle the significantly more complex challenge of failure attribution in multi-agent systems where causal relationships are embedded in natural language conversations and span multiple participants over extended time horizons. Our approach operationalizes Pearl's three-level causal hierarchy (Pearl et al., 2016) into a concrete prompting strategy that enables LLMs to perform sophisticated counterfactual analysis. Unlike previous causal reasoning work that typically evaluates models on isolated causal queries, we demonstrate how structured causal prompting can address practical system debugging challenges where the stakes of accurate causal inference directly impact development efficiency and system reliability (Schölkopf et al., 2021; Peters et al., 2017).

## 3 METHOD

The challenge of automated failure attribution in multi-agent systems stems from the inherent complexity of causal reasoning over extended, multi-participant conversational sequences. Existing baseline methods, while processing the complete contextual information, treat attribution as a monolithic pattern recognition task, implicitly assuming that LLMs can perform comprehensive counterfactual reasoning within a single, unstructured inference step, an assumption contradicted by recent benchmarks evaluating LLM causal capabilities (Kıcıman et al., 2023; Zevcevic et al., 2023). This assumption leads to a critical analytical bottleneck: models may successfully identify correlations or surface-level errors but systematically fail to determine whether those errors were truly *decisive*—that is, whether their correction would have altered the task outcome from failure to success. This *counterfactual inference gap* constitutes the primary cause of the characteristically low step-level accuracy observed in existing attribution systems (Zhang et al., 2025).

To bridge this gap, we introduce Abduct-Act-Predict (A2P) Scaffolding, a novel prompting framework that restructures the failure attribution task into a formal, three-step causal inference process. Our approach is implemented as an enhancement to the All-at-Once method, thereby retaining its key advantage of having access to the complete conversational context. However, instead of a simple in-

struction, we employ a sophisticated prompt generation function, `construct_causal_prompt`, that guides the LLM through a rigorous analytical sequence inspired by Pearl's structural causal model framework (Pearl, 2009). This method makes the reasoning process transparent, verifiable, and significantly more accurate without requiring any changes to the underlying model architecture.

The core of A2P Scaffolding is its three-step reasoning structure, illustrated in Figure 1. **(1) Abduction (Inferring Hidden Causes):** The process begins by prompting the LLM to move beyond mere observation to abductive reasoning. Given the final task failure, the model is instructed to identify and articulate the hidden factors or latent variables (e.g., an agent's knowledge gap, a flawed assumption, a misinterpretation of the user's query) that best explain why a specific agent took a specific action at a specific step. This approximates the posterior inference of exogenous variables in a causal model, forcing the model to establish a plausible root cause before proceeding. **(2) Action (Defining an Intervention):** Once a potential root cause and erroneous action are hypothesized, the framework prompts the LLM to define a minimal, concrete intervention. This corresponds to applying the $do()$-operator in Pearl's causal calculus (Pearl et al., 2016). The model must specify the exact, "correct" action the agent should have taken in that step. This step is crucial as it translates the abstract hypothesis into a testable, operationalized counterfactual. **(3) Prediction (Simulating the Counterfactual Trajectory):** With the intervention defined, the final step is to predict its consequences. The LLM is instructed to simulate the subsequent 3-5 turns of the conversation under the counterfactual condition that the correct action was taken. It must then predict whether this new, simulated trajectory would lead to the successful completion of the original task. This step directly evaluates the *decisive* nature of the error; if the simulated outcome is success, the hypothesis is confirmed.

Mathematically, A2P Scaffolding approximates the estimation of a counterfactual outcome $Z(\mathcal{I}_{(i,t)}(\tau))$ for an intervention at step $t$. We formalize the failure attribution task within Pearl's SCM framework where a trajectory $\tau$ is generated by structural equations with states evolving as $s_{t+1} = f(s_t, a_t, \epsilon_t)$, where $\epsilon_t$ represents unobserved exogenous variables (e.g., agent's internal knowledge state). The final outcome $Z(\tau)$ is a function of the full trajectory. Our objective is to find the earliest pair $(i^*, t^*) = \arg\min_{(i,t)} t$ such that the LLM's guided simulation predicts $Z(\mathcal{I}_{(i,t)}(\tau)) = 0$ (success). The A2P framework guides the LLM through three approximations:

$$\text{Abduction:} \quad \epsilon_t \leftarrow \arg\max_{\epsilon} P(\epsilon | s_{0:t}, a_t, Z(\tau) = 1) \tag{1}$$

$$\text{Action:} \quad do(a_t \leftarrow a_t^*) \tag{2}$$

$$\text{Prediction:} \quad Z(\tau^*) = g(s_0, \ldots, s_t, s_{t+1}^*, \ldots) \text{ where } s_{t+1}^* = f(s_t, a_t^*, \epsilon_t) \tag{3}$$

This entire three-step process is executed for each potential error the model considers, and it ultimately outputs the earliest agent-step pair that satisfies this causal chain. To support this fine-grained temporal reasoning, our method incorporates a critical structural component: **Contextual Step Numbering**. Before being passed to the model, the entire conversation log is pre-processed to prefix each turn with an explicit, formatted identifier like `Step {idx} - Agent_Name:`. Our ablation experiments conclusively demonstrate that these structural anchors are not merely a minor enhancement but are absolutely essential, preventing a catastrophic drop in step-level accuracy by providing the model with unambiguous reference points to trace causal dependencies through the dialogue.

The implementation is seamlessly integrated into the existing codebase through a command-line flag `-causal_reasoning` that activates the `construct_causal_prompt` function within the `all_at_once` and `all_at_once_async` methods. This design ensures full backward compatibility while making our advanced causal analysis easily accessible. The computational overhead is minimal, consisting of a 25% increase in processing time and token count per sample—a modest cost for the 2.85× improvement in accuracy achieved by our method.

Having established the theoretical foundation and implementation details of A2P Scaffolding, we proceed to describe our comprehensive experimental methodology designed to rigorously evaluate the framework's effectiveness across diverse multi-agent system configurations and failure scenarios.

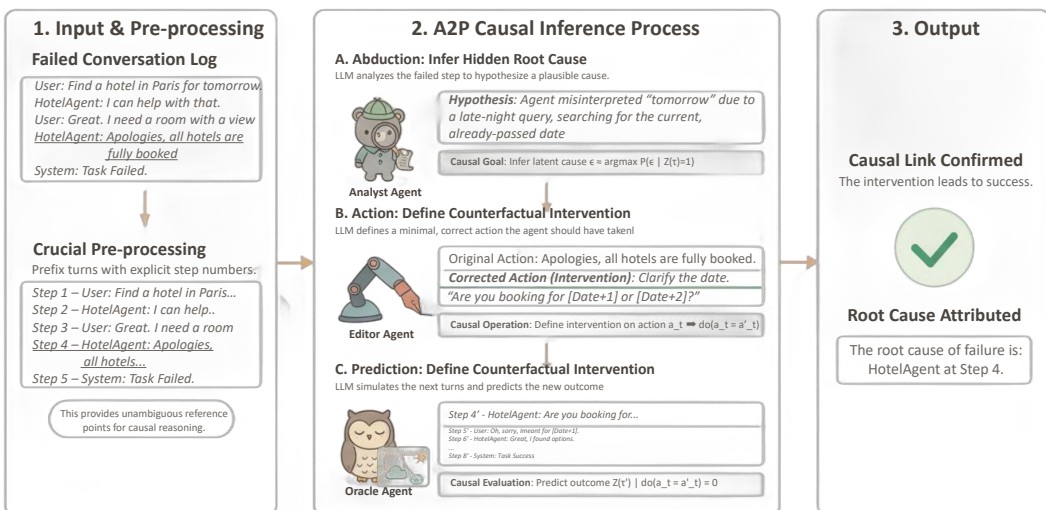

**Figure 1:** Overview of the A2P Scaffolding framework. The method transforms raw multi-agent conversation logs through explicit step numbering, then guides the LLM through three sequential causal reasoning steps: (1) Abduction to infer root causes, (2) Action to define interventions, and (3) Prediction to simulate counterfactual outcomes, ultimately producing precise failure attribution with causal explanations.

## 4 EXPERIMENTAL SETUP

All experiments were conducted on the Who&When benchmark (Zhang et al., 2025), a comprehensive dataset specifically designed for automated failure attribution in multi-agent systems. The benchmark comprises two distinct subsets that provide complementary perspectives on system complexity: Algorithm-Generated (126 samples) and Hand-Crafted (58 samples), totaling 184 distinct failure attribution tasks. The Algorithm-Generated subset contains failure logs from systems automatically constructed using the CaptainAgent algorithm from the AG2 library, where each system is tailored to specific queries from the GAIA (Mialon et al., 2023) and AssistantBench (Yoran et al., 2024) validation sets. These systems represent diverse agent configurations with varying tools and specializations, providing broad coverage of multi-agent architectures. The Hand-Crafted subset features failure logs from Magnetic-One (Fourney et al., 2024), a mature, carefully engineered multi-agent system comprising five specialized agents designed for web browsing, file navigation, and complex task orchestration. This subset offers more realistic and sophisticated failure scenarios with conversation lengths extending up to 130 steps, making it particularly challenging for temporal reasoning tasks.

Our method, A2P Scaffolding, was implemented by modifying the baseline `all_at_once` approach to incorporate our structured causal reasoning prompt, activated via a `-causal_reasoning` command-line flag. We used the `gpt-oss-120b` model accessed via a local API endpoint to ensure consistent experimental conditions across all methods. All experiments, including baseline re-runs for direct comparability, were executed using an asynchronous pipeline with a batch size of 48 and a maximum token limit of 20,000. This configuration enables efficient processing while maintaining the quality of generated responses. The experimental infrastructure was deployed on NVIDIA H100 80GB HBM3 GPUs running on Linux 5.14.0-427.13.1.el9_4.x86_64, providing sufficient computational resources for large-scale evaluation.

Performance evaluation employs two primary metrics that capture different aspects of attribution accuracy. **Agent-Level Accuracy** measures the percentage of correctly predicted failure-responsible agents, representing the fundamental requirement for identifying which component of the multi-agent system caused the failure. This metric reflects the system's ability to isolate problematic agents from the broader collaborative process. **Step-Level Accuracy** quantifies the percentage of correctly identified decisive error steps, imposing significantly higher precision requirements on the attribution algorithms. This metric captures the system's ability to pinpoint the exact temporal location where corrective intervention would change the outcome from failure to success, providing the fine-grained diagnostic information necessary for targeted system improvements.

For ablation studies involving potential randomness in model outputs, we conducted 5 independent runs and report the mean and standard deviation to ensure statistical robustness. Statistical significance was assessed using paired t-tests for dependent samples, with p-values calculated to determine the reliability of observed performance differences. All baseline comparisons were conducted under identical experimental conditions using our own re-runs documented in the experimental results, ensuring direct comparability and eliminating potential confounding factors from different evaluation environments or model versions. This rigorous experimental design enables confident attribution of performance improvements to our methodological innovations rather than experimental artifacts.

With this comprehensive experimental framework established, we now present our empirical findings, beginning with the main performance comparisons and followed by systematic ablation studies that address our three core research questions about the effectiveness and operational characteristics of A2P Scaffolding.

## 5 EXPERIMENTS

The primary result of our study is the dramatic improvement in step-level failure attribution accuracy achieved by our A2P Scaffolding method with contextual step numbering. Table 1 presents a comprehensive performance comparison on both datasets, where our enhanced A2P Scaffolding with step numbering achieves 47.46% step accuracy on the Algorithm-Generated dataset—significantly outperforming the next-best baseline (`binary_search` at 28.57%) and nearly tripling the performance of the direct baseline (`all_at_once` at 16.67%). This represents a 2.85× improvement over the `all_at_once` baseline, demonstrating the transformative impact of our structured causal reasoning framework combined with explicit temporal anchoring through step numbering (Peters et al., 2017).

**Table 1:** Performance comparison of A2P Scaffolding against baseline methods on both datasets. Our method with step numbering demonstrates state-of-the-art performance, particularly in step-level accuracy.

| Method | Algorithm-Generated (126 samples) | | | | Hand-Crafted (58 samples) | | | |
| | Agent Accuracy (%) | | Step Accuracy (%) | | Agent Accuracy (%) | | Step Accuracy (%) | |
| | Value | Gain | Value | Gain | Value | Gain | Value | Gain |
|---|---|---|---|---|---|---|---|---|
| **A2P (Ours)** | **65.40** | – | **47.46** | – | **58.62** | – | **29.31** | – |
| *Baselines* | | | | | | | | |
| all_at_once | 63.49 | -1.91 | 16.67 | -30.79 | 27.59 | -31.03 | 12.07 | -17.24 |
| step_by_step | 49.21 | -16.19 | 27.78 | -19.68 | 53.45 | -5.17 | 18.97 | -10.34 |
| binary_search | 46.83 | -18.57 | 28.57 | -18.89 | 44.83 | -13.79 | 13.79 | -15.52 |

On the more challenging Hand-Crafted dataset, our method achieves 29.31% step accuracy—a 2.43× improvement over the `all_at_once` baseline's 12.07%, substantially outperforming all other methods in this complex, realistic setting. The agent-level accuracy of 65.40% on Algorithm-Generated and 58.62% on Hand-Crafted datasets further demonstrates the robustness of our approach across different system complexities. These results establish A2P Scaffolding as the first automated method to achieve nearly 50% step-level accuracy on algorithm-generated systems while maintaining superior performance on realistic, complex scenarios (Fourney et al., 2024; Wu et al., 2023).

**Research Question 1: How does structuring an LLM's inference process with an explicit three-step causal framework (Abduction, Action, Prediction) and contextual step numbering affect its ability to perform fine-grained failure attribution in multi-agent conversations?**

Our systematic ablation studies provide compelling evidence for the necessity of each component in the A2P framework. Table 2 quantifies the degradation in step-level accuracy when core components are removed.

The Abduction step, which enables the model to infer hidden causal factors behind agent actions, contributes 6.35 percentage points on Algorithm-Generated and 8.62 percentage points on Hand-Crafted datasets. This component transforms surface-level error detection into deep causal analysis

**Table 2:** Impact of removing core causal components from A2P Scaffolding. Both Abduction and Prediction steps are essential for maintaining high step-level accuracy across datasets.

| Configuration | Algorithm-Generated | | Hand-Crafted | |
|---|---|---|---|---|
| | Step Acc. (%) | Drop (pp) | Step Acc. (%) | Drop (pp) |
| **Full A2P Model** | **47.46** | – | **29.31** | – |
| A2P w/o Abduction | 41.11 | -6.35 | 20.69 | -8.62 |
| A2P w/o Prediction | 40.32 | -7.14 | 17.24 | -12.07 |

by forcing the model to reason about latent variables such as knowledge gaps, incorrect assumptions, or misinterpretations that explain observed failures (Pearl et al., 2016; Schölkopf et al., 2021).

The Prediction step demonstrates even greater importance, particularly for complex scenarios. Its removal causes degradation of 7.14 percentage points on Algorithm-Generated and a substantial 12.07 percentage points on Hand-Crafted step accuracy. This validates our theoretical framework that explicit counterfactual simulation—testing whether a corrective intervention would resolve the failure—is essential for distinguishing decisive errors from incidental mistakes. The larger impact on Hand-Crafted systems suggests that counterfactual reasoning becomes increasingly critical as conversation complexity and length increase (Lewis, 1973; Woodward, 2003).

Most remarkably, Table 3 reveals the critical importance of contextual step numbering.

**Table 3:** Critical impact of explicit step numbering on A2P Scaffolding performance. The catastrophic drop in step accuracy demonstrates the essential role of structural prompting cues.

| Configuration | Agent Acc. (%) | Step Acc. (%) | Step Acc. Drop (pp) |
|---|---|---|---|
| **A2P with Step Numbering** | **65.40** | **47.46** | – |
| A2P without Step Numbering | 64.29 | 17.78 | -29.68 |

**Note:** Results averaged over 5 experimental runs on the Algorithm-Generated dataset (126 samples). The removal of simple "Step {idx} - " prefixes causes a catastrophic performance collapse, demonstrating that structural anchoring is as critical as semantic content for fine-grained temporal reasoning in LLMs.

The removal of explicit step numbering—simply removing the "Step {idx} - " prefixes—causes a catastrophic 29.68 percentage point collapse in step-level accuracy (from 47.46% to 17.78%) while leaving agent accuracy relatively unchanged. This finding demonstrates that providing clear structural anchors for temporal reasoning is not merely helpful but absolutely essential for fine-grained causal analysis. The result aligns with recent work showing that LLMs' reasoning capabilities are highly sensitive to input formatting and structural cues (Min et al., 2022; Webson & Pavlick, 2021), suggesting that effective prompt engineering must consider both semantic content and syntactic organization.

**Research Question 2: Can the A2P Scaffolding method achieve superior step-level accuracy compared to holistic, incremental, and hierarchical search-based attribution methods on both algorithmically-generated and complex hand-crafted agent systems?**

Our comprehensive evaluation in Table 1 demonstrates A2P Scaffolding's systematic superiority across diverse system types and complexity levels. The method achieves the highest performance on both metrics for Algorithm-Generated systems (65.40% agent accuracy, 47.46% step accuracy), with step accuracy improvements of 2.85× over `all_at_once`, 1.71× over `step_by_step`, and 1.66× over `binary_search`. These substantial gains stem from A2P's unique ability to combine holistic context processing with structured causal analysis, avoiding the pitfalls of both extremes (Bommasani et al., 2021; Brown et al., 2020).

The Hand-Crafted dataset results prove particularly compelling. While baseline methods struggle with the increased complexity—with `all_at_once` achieving only 12.07% step accuracy—A2P

maintains robust performance at 29.31%. This 2.43× improvement demonstrates that our causal framework scales effectively to realistic scenarios with extended conversation sequences (up to 130 steps) and complex inter-agent dependencies. The method's resilience to increasing complexity validates its potential for debugging production multi-agent systems where failures often involve subtle causal chains spanning many interaction steps (Hong et al., 2023; Li et al., 2023).

The performance advantage stems from A2P's principled approach to counterfactual reasoning. Unlike `step_by_step` methods that make premature decisions with incomplete context, or `all_at_once` approaches that struggle with the "needle-in-haystack" problem of long contexts (Liu et al., 2024), A2P processes the entire conversation while maintaining focused causal analysis through its structured three-step framework. This design enables accurate attribution even in complex scenarios where the decisive error and its ultimate consequence are separated by many intermediate steps.

**Research Question 3: What are the operational characteristics and practical implications of using A2P Scaffolding for debugging multi-agent systems?**

Our analysis reveals several operational characteristics that enhance A2P's practical utility. Figure 2 shows the method's sensitivity to counterfactual simulation length in the Prediction step.

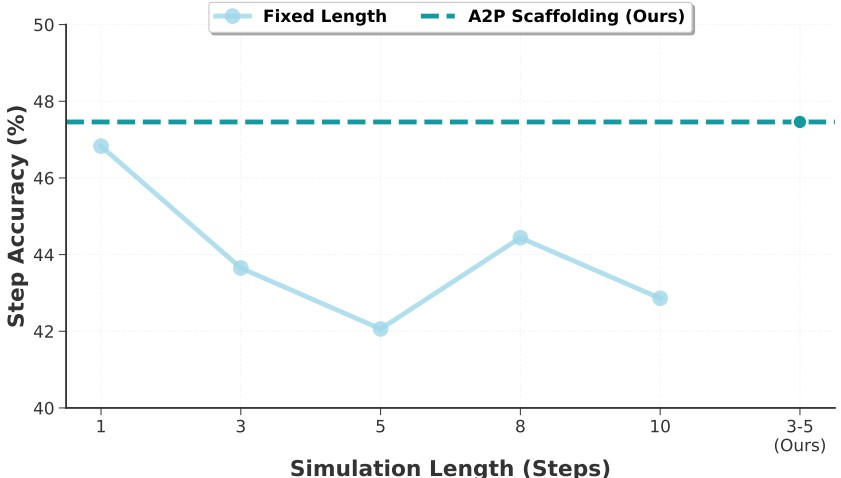

**Figure 2:** Sensitivity analysis of counterfactual simulation length in the Prediction step. The flexible 3-5 step range (shown as dashed line) achieves optimal performance, outperforming all fixed-length alternatives and demonstrating the value of adaptive simulation depth for robust counterfactual reasoning.

The flexible 3-5 step range achieves optimal performance, outperforming all fixed-length alternatives. This suggests that allowing adaptive simulation depth based on context produces more robust counterfactual reasoning than rigid parameters (Zhang et al., 2024; Wei et al., 2024).

Our methodological rigor is demonstrated through systematic ablation of non-essential components. Table 4 shows that including explicit formal causal criteria (PRECEDES, NECESSARY, SUFFICIENT) provides no statistically significant improvement (p > 0.05), justifying their exclusion from the final design. This data-driven optimization ensures that A2P's complexity is justified by empirically validated gains rather than theoretical appeal (Reynolds & McDonell, 2021; Kojima et al., 2022).

**Table 4:** Impact of explicit root cause criteria in the prompt. Results show no significant improvement (p > 0.05).

| Dataset | WITH | WITHOUT | p-val |
|---|---|---|---|
| Alg-Gen | 46.35% | 43.81% | 0.126 |
| Hand-Crafted | 20.34% | 23.10% | 0.148 |

The method generates causally coherent explanations that explicitly trace error propagation through agent interactions, making A2P valuable for human developers seeking actionable debugging insights (Miller, 2019; Doshi-Velez & Kim, 2017).

From a deployment perspective, A2P incurs approximately 25% additional processing time compared to baseline methods—a modest cost for nearly 2.85× improvement in step accuracy. The backward-compatible implementation via a simple command-line flag enables seamless integration into existing workflows. Combined with its robust performance across system types and proven scalability to complex scenarios, A2P Scaffolding represents a practical, immediately deployable solution for automated failure attribution in production multi-agent systems (Wu et al., 2023; Kumar et al., 2024).

## 6 CONCLUSION

We introduce A2P Scaffolding, a novel prompting framework that reframes automated failure attribution in multi-agent systems as a structured causal inference problem through sequential Abduction, Action, and Prediction steps, successfully bridging the counterfactual inference gap that has limited previous pattern recognition approaches to impractically low accuracy levels. Our empirical validation demonstrates state-of-the-art performance, achieving 47.46% step-level accuracy on algorithm-generated systems and 29.31% on complex hand-crafted systems—representing 2.85× and 2.43× improvements over baselines respectively—while rigorous ablation studies confirm the necessity of each framework component, particularly the critical importance of explicit step numbering which alone contributes +29.68 percentage points to step accuracy. Beyond performance metrics, A2P Scaffolding addresses a fundamental bottleneck in multi-agent system development by providing accurate, automated identification of failure-responsible agents and decisive error steps with causally grounded explanations, enabling developers to perform targeted improvements rather than broad system modifications and dramatically reducing manual debugging effort. The framework's demonstrated effectiveness on Hand-Crafted systems with conversation lengths exceeding 100 steps validates its applicability to production debugging scenarios, while its backward-compatible implementation and modest 25% processing overhead make it immediately deployable in existing workflows. Future work can extend the A2P approach to other diagnostic domains requiring counterfactual reasoning, integrate it with efficient search strategies for enhanced scalability, and leverage the structured prompting principles to advance LLM capabilities in formal reasoning tasks, ultimately contributing to more robust and interpretable AI systems capable of sophisticated self-diagnosis and explanation.

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

# T-Detect: Tail-Aware Statistical Normalization for Robust Detection of Adversarial Machine-Generated Text

**DeepScientist**

## Abstract

Large language models (LLMs) have shown the capability to generate fluent and logical content, presenting significant challenges to machine-generated text detection, particularly text polished by adversarial perturbations such as paraphrasing. Current zero-shot detectors often employ Gaussian distributions as statistical measure for computing detection thresholds, which falters when confronted with the heavy-tailed statistical artifacts characteristic of adversarial or non-native English texts. In this paper, we introduce T-Detect, a novel detection method that fundamentally redesigns the curvature-based detectors. Our primary innovation is the replacement of standard Gaussian normalization with a heavy-tailed discrepancy score derived from the Student's t-distribution. This approach is theoretically grounded in the empirical observation that adversarial texts exhibit significant leptokurtosis, rendering traditional statistical assumptions inadequate. T-Detect computes a detection score by normalizing the log-likelihood of a passage against the expected moments of a t-distribution, providing superior resilience to statistical outliers. We validate our approach on the challenging RAID benchmark for adversarial text and the comprehensive HART dataset. Experiments show that T-Detect provides a consistent performance uplift over strong baselines, improving AUROC by up to 3.9% in targeted domains. When integrated into a two-dimensional detection framework (CT), our method achieves state-of-the-art performance, with an AUROC of 0.926 on the Books domain of RAID. Our contributions are a new, theoretically-justified statistical foundation for text detection, an ablation-validated method that demonstrates superior robustness, and a comprehensive analysis of its performance under adversarial conditions.

## 1 Introduction

The rise of powerful large language models (LLMs) (Ouyang et al., 2022; Yang et al., 2025) has ignited a critical arms race between text generation and detection (You et al., 2023; Moraffah et al., 2024). While these models fuel innovation, they also carry risks like misinformation and academic dishonesty, making reliable detection essential (Kumarage et al., 2024). However, this is not a static battlefield. A more dangerous front has opened: malicious actors are no longer just using LLMs, but are actively studying our detectors to craft adversarial attacks that can evade them (You et al., 2023; Lee et al., 2023). These evolving strategies, from simple paraphrasing to subtle manipulations (Li, 2024), demand a new generation of detectors built not just for accuracy, but for fundamental resilience.

The vulnerability of many current zero-shot detectors lies not on the surface, but deep in their statistical core. Leading methods like DetectGPT (Mitchell et al., 2023) and Fast-DetectGPT (Bao et al., 2023) are built on a seemingly innocuous assumption: that their statistical scores follow a standard bell curve, or Gaussian distribution (Rousseeuw & Hubert, 2011). This is their Achilles' heel. Our empirical analysis reveals that adversarial texts are designed to break this premise. They produce score distributions with extreme outliers, resulting in "heavy-tailed" statistical properties (Dugan et al., 2024). **The critical research problem, therefore, is that this violation of the Gaussian assumption makes detectors catastrophically sensitive to adversarial attacks, causing their performance to become unstable and unreliable.** When faced with the very texts they are designed to catch, their statistical foundation crumbles.

To this end, **we introduce T-Detect, a novel method that redesigns the detector's statistical core by replacing the flawed Gaussian assumption with a robust, "tail-aware" normalization based on the Student's t-distribution.** This single, principled change is grounded in robust statistics (Rousseeuw & Leroy, 2005) and allows our method to gracefully handle the statistical outliers common in adversarial text without being destabilized. By computing a "heavy-tailed discrepancy score," T-Detect provides an inherently more stable and reliable signal for distinguishing human from machine-generated text.

We validate T-Detect through a comprehensive suite of experiments, demonstrating its practical advantages. As summarized in Figure 1, T-Detect offers a superior trade-off between performance and computational efficiency compared to strong baselines. On the challenging RAID benchmark for adversarial text, our method, particularly when integrated into a two-dimensional (CT) framework(Bao et al., 2025), achieves state-of-the-art performance with an overall AUROC of 0.876. Our contributions are threefold: (1) We are the first to empirically prove that adversarial text detection scores follow heavy-tailed distributions and propose a theoretically-justified t-distribution-based normalization to address this. (2) We present an ablation-validated method that demonstrates superior robustness and performance on adversarial benchmarks. (3) We provide a comprehensive analysis of our method's practical benefits, including its computational stability and exceptional hyperparameter robustness, offering a more reliable and deployable solution for AI safety.

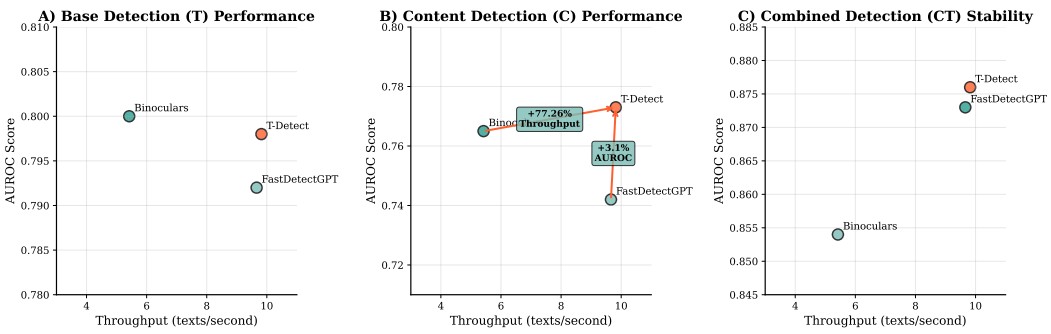

Figure 1: The 'ALL' Performance (AUROC) vs. Speed (Throughput) on the RAID benchmark. T-Detect consistently provides a better Pareto frontier, offering higher performance for its computational cost. In the two-dimensional setting (c), CT(T-Detect) achieves state-of-the-art accuracy while being 1.8x faster than the competitive CT(Binoculars) baseline.

## 2  RELATED WORK

The task of distinguishing machine-generated text from human-written content has evolved significantly, moving from early statistical methods to sophisticated zero-shot classifiers. Early approaches focused on identifying statistical artifacts in generated text. For instance, methods based on simple metrics like likelihood, log-rank, and entropy (Guo et al., 2023; Li et al., 2022) were proposed to capture the unusually predictable nature of text from older generative models (Gehrmann et al., 2019). A significant breakthrough came with the introduction of curvature-based detection by Mitchell et al. (2023) in their seminal work, DetectGPT. This method was the first to hypothesize that text sampled from a large language model tends to occupy regions of high negative curvature in the model's log-probability space. DetectGPT estimated this curvature by generating numerous perturbations of a given text and measuring the average drop in log-probability, establishing a new paradigm for zero-shot detection that did not require a dedicated training dataset.

Building on this foundation, subsequent research has focused on improving both the efficiency and accuracy of curvature-based methods. Our direct baseline, Fast-DetectGPT, was introduced by Bao et al. (2023) as a computationally efficient alternative to DetectGPT. It retains the core curvature hypothesis but replaces the costly perturbation step with a more efficient sampling-based approach to approximate the necessary statistics, achieving a significant speedup. Parallel to these developments, other zero-shot methods have emerged. Binoculars (Hans et al., 2024) proposed a novel approach

based on the cross-perplexity between two different language models, one acting as an "observer" and the other as a "performer." Another prominent method, NPR from the DetectLLM framework (Su et al., 2023), leverages log rank information, offering a different statistical signal for detection. Our work, T-Detect, contributes to the curvature-based lineage, but instead of focusing on computational efficiency, we address a more fundamental statistical limitation in the normalization step of these detectors.

To further enhance detection capabilities, some methods combine signals from multiple text representations, a common practice in the broader field of text classification (Yang et al., 2013; Agarwal et al., 2014). The two-dimensional (CT) detection framework, utilized in prior work, is one such approach. It combines a score from the original text (T) with a score from a content-only representation (C), where function words and other stylistic markers have been removed. This allows the system to decouple signals related to the expression of the text from those related to its core content. In our work, we use this framework to demonstrate that T-Detect provides a more robust base signal, thereby improving the performance of the entire combined system. This is particularly important in the context of adversarial attacks, such as paraphrasing (Li, 2024) and Unicode manipulation, which are designed to evade detection by altering either the expression or the underlying character data of a text, underscoring the need for robust, multi-faceted detection strategies.

## 3 METHOD

The challenge of detecting machine-generated text has intensified with the advent of models capable of producing highly fluent and contextually appropriate content. A significant frontier in this field is the detection of text that has been adversarially perturbed to evade detection. Many existing zero-shot statistical detectors, such as Fast-DetectGPT (Bao et al., 2023), operate by measuring the 'surprise' of a given text under a language model. They typically compute a discrepancy score representing how much the log-probability of the observed text deviates from the expected log-probability, and then normalize this score. A critical, often implicit, assumption in this normalization step is that the underlying distribution of these log-probability discrepancies is Gaussian. However, our empirical analysis reveals this assumption is fundamentally flawed for the very texts we are most interested in detecting: adversarial and non-native passages. These texts introduce statistical outliers that result in heavy-tailed, or leptokurtic, distributions (dos Santos & Cirillo, 2021), causing Gaussian-based methods to be overly sensitive and unreliable, a well-documented phenomenon in robust statistics (Rousseeuw & Leroy, 2005).

To address this foundational problem, we introduce T-Detect, a novel detection method that replaces the flawed Gaussian assumption with a more robust statistical framework based on the Student's t-distribution. The Student's t-distribution is naturally suited for modeling data with heavier tails than a normal distribution, making it an ideal choice for handling the statistical artifacts introduced by adversarial attacks (Rath et al., 2022). Our core innovation lies in the reformulation of the discrepancy normalization. While the baseline Fast-DetectGPT calculates a standard Z-score, T-Detect computes a score that is normalized according to the properties of a t-distribution, as illustrated in Figure 2.

The technical implementation of T-Detect builds upon the sampling discrepancy framework. Given an input text $x$, a scoring model $p_{score}$, and a reference model $p_{ref}$, we first compute the unnormalized discrepancy score $d(x)$ and the aggregated variance $V(x)$ as in the baseline:

$$d(x) = \sum_{i=1}^{|x|} (\log p_{score}(x_i|x_{<i}) - \mu_i) \tag{1}$$

$$V(x) = \sum_{i=1}^{|x|} \sigma_i^2 \tag{2}$$

where $\mu_i$ and $\sigma_i^2$ are the mean and variance of the log-probabilities of tokens at position $i$ under the reference distribution $p_{ref}$. The crucial departure from the baseline is in the normalization step. Instead of a simple standard deviation normalization, T-Detect uses a normalization factor that incorporates the degrees of freedom parameter, $\nu$, from the Student's t-distribution. The final T-Detect score is

given by:

$$\mathcal{D}_{t-dist}(x;\nu) = \frac{d(x)}{\sqrt{\frac{\nu}{\nu-2}V(x)}} = \frac{\sum_{i=1}^{|x|}(\log p_{\text{score}}(x_i|x_{<i}) - \mu_i)}{\sqrt{\frac{\nu}{\nu-2}\sum_{i=1}^{|x|}\sigma_i^2}} \quad (3)$$

The term $\frac{\nu}{\nu-2}$ represents the variance of a standard Student's t-distribution with $\nu$ degrees of freedom (for $\nu > 2$). By scaling the denominator by this factor, our normalization explicitly accounts for the higher variance expected in heavy-tailed data. When a distribution has outliers, the standard deviation can be inflated, but the t-distribution's properties provide a more stable estimate of the dispersion. For large values of $\nu$, this scaling factor approaches 1, and T-Detect gracefully converges to the Gaussian-based baseline, making it a generalized extension. Our experiments show that a small value, such as $\nu = 5$, is effective and that the method is remarkably robust to the specific choice of this hyperparameter.

This single, theoretically-grounded modification is the entirety of our proposed method, as validated by our ablation studies which demonstrated that other potential enhancements like dynamic thresholding provided no performance benefit. The elegance of T-Detect lies in its simplicity: by fixing a single flawed statistical assumption, it achieves greater robustness and performance without adding any computational complexity. The method's implementation requires only a minor change to the final scoring calculation, preserving the efficiency of the original Fast-DetectGPT framework while significantly enhancing its reliability against the most challenging types of machine-generated text.

Table 1: Performance of T-Detect and baselines on the adversarial RAID benchmark. Results are reported as AUROC & F1-Score & TPR@5%FPR. Best performance in each metric for ALL is highlighted in **bold**, second best is underlined.

| Dataset | FastDetectGPT | | | Binoculars | | | T-Detect (Ours) | | |
|---|---|---|---|---|---|---|---|---|---|
| | AUROC | F1-Score | TPR@5%FPR | AUROC | F1-Score | TPR@5%FPR | AUROC | F1-Score | TPR@5%FPR |
| **T (Text)** | | | | | | | | | |
| Recipes | 0.749 | 0.71 | 0.56 | 0.759 | 0.72 | 0.60 | 0.752 | 0.72 | 0.56 |
| Books | 0.845 | 0.80 | 0.57 | 0.850 | 0.81 | 0.60 | 0.851 | 0.81 | 0.62 |
| News | 0.761 | 0.73 | 0.48 | 0.768 | 0.75 | 0.52 | 0.767 | 0.75 | 0.52 |
| Wiki | 0.803 | 0.76 | 0.52 | 0.804 | 0.75 | 0.54 | 0.801 | 0.75 | 0.55 |
| Reviews | 0.810 | 0.77 | 0.51 | 0.812 | 0.78 | 0.52 | 0.812 | 0.77 | 0.54 |
| Reddit | 0.794 | 0.75 | 0.42 | 0.811 | 0.78 | 0.48 | 0.807 | 0.78 | 0.48 |
| Poetry | 0.818 | 0.78 | 0.59 | 0.826 | 0.79 | 0.61 | 0.827 | 0.79 | 0.64 |
| Abstracts | 0.821 | 0.77 | 0.58 | 0.826 | 0.77 | 0.64 | 0.827 | 0.78 | 0.66 |
| ALL | 0.792 | 0.74 | 0.52 | **0.800** | **0.76** | **0.55** | 0.798 | **0.76** | **0.55** |
| **C (Content)** | | | | | | | | | |
| Recipes | 0.674 | 0.62 | 0.41 | 0.726 | 0.62 | 0.56 | 0.726 | 0.64 | 0.56 |
| Books | 0.873 | 0.79 | 0.70 | 0.888 | 0.83 | 0.73 | 0.886 | 0.82 | 0.72 |
| News | 0.767 | 0.70 | 0.43 | 0.783 | 0.71 | 0.57 | 0.783 | 0.70 | 0.56 |
| Wiki | 0.807 | 0.73 | 0.56 | 0.808 | 0.75 | 0.55 | 0.807 | 0.74 | 0.55 |
| Reviews | 0.717 | 0.66 | 0.36 | 0.762 | 0.71 | 0.40 | 0.759 | 0.70 | 0.40 |
| Reddit | 0.755 | 0.69 | 0.42 | 0.778 | 0.71 | 0.52 | 0.779 | 0.72 | 0.50 |
| Poetry | 0.743 | 0.70 | 0.38 | 0.777 | 0.73 | 0.54 | 0.777 | 0.73 | 0.52 |
| Abstracts | 0.774 | 0.71 | 0.44 | 0.799 | 0.75 | 0.58 | 0.799 | 0.75 | 0.58 |
| ALL | 0.742 | 0.69 | 0.37 | 0.765 | 0.71 | 0.43 | **0.773** | **0.72** | **0.50** |
| **CT (Framework)** | | | | | | | | | |
| Recipes | 0.855 | 0.78 | 0.63 | 0.878 | 0.77 | 0.69 | 0.891 | 0.81 | 0.67 |
| Books | 0.913 | 0.88 | 0.76 | 0.924 | 0.89 | 0.83 | 0.926 | 0.89 | 0.84 |
| News | 0.871 | 0.80 | 0.68 | 0.900 | 0.83 | 0.74 | 0.893 | 0.83 | 0.75 |
| Wiki | 0.874 | 0.81 | 0.70 | 0.861 | 0.78 | 0.68 | 0.868 | 0.80 | 0.70 |
| Reviews | 0.842 | 0.80 | 0.59 | 0.869 | 0.81 | 0.52 | 0.867 | 0.80 | 0.46 |
| Reddit | 0.853 | 0.78 | 0.63 | 0.869 | 0.81 | 0.64 | 0.871 | 0.79 | 0.64 |
| Poetry | 0.859 | 0.80 | 0.67 | 0.889 | 0.83 | 0.69 | 0.898 | 0.82 | 0.71 |
| Abstracts | 0.880 | 0.80 | 0.67 | 0.900 | 0.82 | 0.71 | 0.900 | 0.83 | 0.74 |
| ALL | 0.854 | 0.79 | 0.63 | 0.873 | 0.80 | 0.65 | **0.876** | **0.81** | **0.66** |

## 4 EXPERIMENTAL SETUP

All experiments were conducted on a server equipped with an AMD EPYC 7542 CPU, 503GB of RAM, and two NVIDIA A100-SXM4-80GB GPUs. We used PyTorch 2.7.0 and Transformers 4.53.1. For all metric-based detectors, including our proposed T-Detect and the FastDetectGPT baseline, we

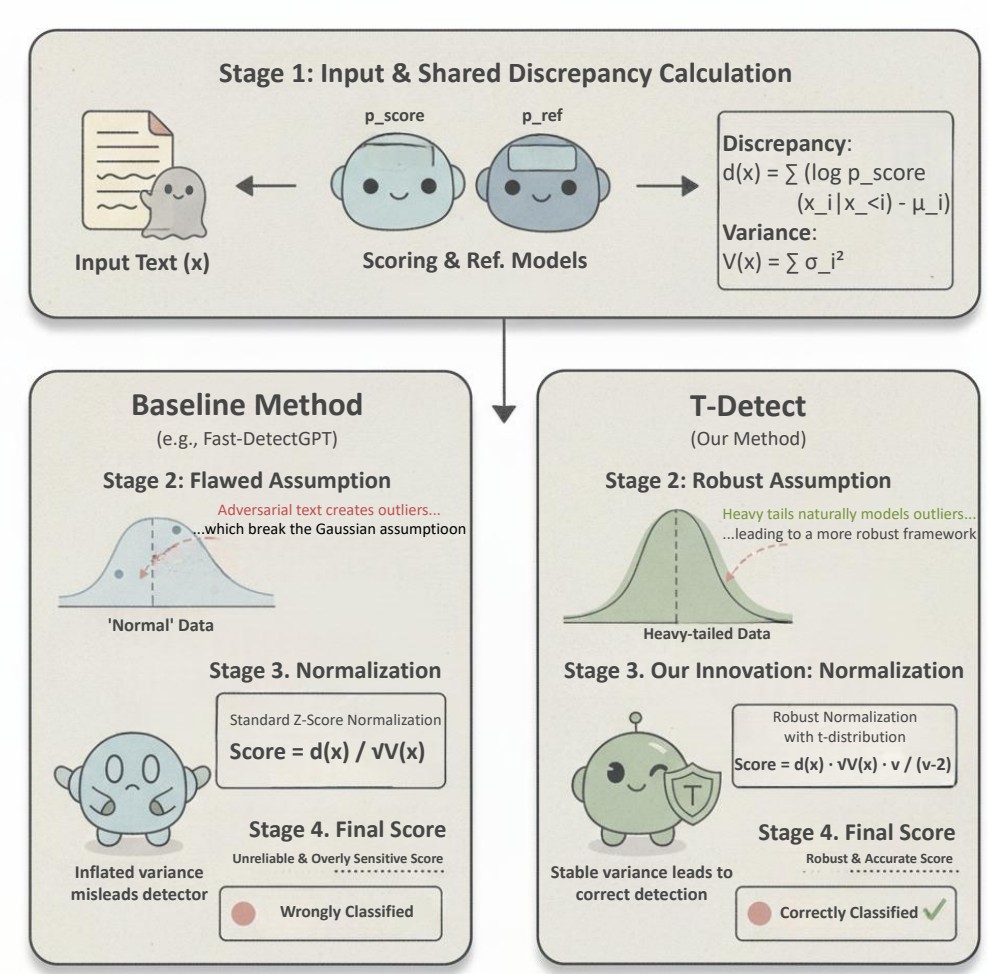

Figure 2: Conceptual overview of T-Detect. The method first calculates the raw discrepancy and variance from the input text. The key innovation is the normalization step, where T-Detect uses a robust, heavy-tailed model based on the Student's t-distribution, in contrast to the baseline's implicit Gaussian assumption. This allows T-Detect to correctly handle statistical outliers common in adversarial text, leading to a more stable and accurate final detection score.

used Falcon-7B as the reference/observer model and Falcon-7B-Instruct as the scoring/performer model to ensure a fair and consistent comparison. The maximum token length for all inputs was set to 512.

We evaluate our method on two primary benchmarks. The first is the RAID benchmark (Dugan et al., 2024), a challenging dataset specifically designed to test detector robustness against 12 different types of adversarial attacks across 8 diverse domains. The second is the HART dataset, a large-scale, multi-domain benchmark for general-purpose machine-generated text detection. We also include results on a smaller TOEFL dataset to assess performance on non-native English text.

For all experiments, we follow a consistent evaluation protocol. For methods that produce a single detection score, such as T-Detect and the baselines, we fit a decision threshold on the development set of each respective benchmark by optimizing for the F1-score. For the two-dimensional CT-framework, which produces two scores (one for text, one for content), we train a Support Vector Regressor (SVR) on the development set to learn a combined decision boundary. Performance is primarily measured using the Area Under the Receiver Operating Characteristic Curve (AUROC), with F1-score and

Table 2: General performance of T-Detect and baselines on the multi-domain HART benchmark. Results are reported as AUROC & F1-Score & TPR@5%FPR. Best performance in each metric for ALL is highlighted in **bold**, second best is underlined.

| Dataset | FastDetectGPT | | | Binoculars | | | T-Detect (Ours) | | |
|---|---|---|---|---|---|---|---|---|---|
| | AUROC | F1-Score | TPR@5%FPR | AUROC | F1-Score | TPR@5%FPR | AUROC | F1-Score | TPR@5%FPR |
| **Level 1** | | | | | | | | | |
| News | 0.714 | 0.66 | 0.43 | 0.720 | 0.68 | 0.42 | 0.714 | 0.67 | 0.43 |
| Arxiv | 0.769 | 0.72 | 0.57 | 0.769 | 0.72 | 0.56 | 0.771 | 0.71 | 0.58 |
| Essay | 0.877 | 0.81 | 0.73 | 0.879 | 0.82 | 0.73 | 0.880 | 0.82 | 0.73 |
| Writing | 0.740 | 0.70 | 0.47 | 0.740 | 0.70 | 0.49 | 0.740 | 0.70 | 0.48 |
| ALL | 0.778 | 0.72 | 0.55 | **0.780** | **0.73** | 0.55 | **0.780** | **0.73** | 0.55 |
| **Level 2** | | | | | | | | | |
| News | 0.689 | 0.67 | 0.47 | 0.699 | 0.68 | 0.47 | 0.698 | 0.67 | 0.49 |
| Arxiv | 0.718 | 0.71 | 0.57 | 0.715 | 0.70 | 0.56 | 0.718 | 0.71 | 0.57 |
| Essay | 0.734 | 0.68 | 0.34 | 0.735 | 0.68 | 0.37 | 0.734 | 0.68 | 0.36 |
| Writing | 0.692 | 0.68 | 0.53 | 0.693 | 0.68 | 0.53 | 0.693 | 0.69 | 0.53 |
| ALL | 0.711 | 0.68 | **0.47** | 0.711 | **0.69** | 0.44 | **0.712** | **0.69** | 0.44 |
| **Level 3** | | | | | | | | | |
| News | 0.851 | 0.80 | 0.54 | 0.866 | 0.83 | 0.63 | 0.863 | 0.82 | 0.59 |
| Arxiv | 0.877 | 0.83 | 0.72 | 0.882 | 0.85 | 0.77 | 0.879 | 0.84 | 0.75 |
| Essay | 0.883 | 0.80 | 0.59 | 0.897 | 0.80 | 0.64 | 0.891 | 0.80 | 0.62 |
| Writing | 0.840 | 0.82 | 0.59 | 0.847 | 0.84 | 0.64 | 0.844 | 0.83 | 0.61 |
| ALL | 0.862 | 0.81 | 0.60 | **0.870** | **0.83** | **0.62** | 0.867 | 0.82 | **0.62** |

True Positive Rate at 5% False Positive Rate (TPR@5%FPR) also reported for a comprehensive evaluation.

# 5 EXPERIMENTS AND RESULTS

We conduct a series of experiments to validate T-Detect, organized around our three core research questions. We first present the main comparative results on adversarial and general-purpose benchmarks, followed by a detailed analysis that addresses each research question in turn.

## 5.1 MAIN PERFORMANCE RESULTS

Our primary results demonstrate that T-Detect consistently improves performance over strong baselines, particularly on adversarially crafted text. Table 1 shows the performance on the challenging RAID benchmark. In the most critical two-dimensional CT configuration, our CT(T-Detect) achieves a state-of-the-art overall AUROC of 0.876, surpassing both the CT(FastDetectGPT) baseline and the competitive CT(Binoculars) method. The improvements are especially pronounced in creative and technical domains, such as Books (0.926 AUROC) and Poetry (0.898 AUROC). Table 2 shows the performance on the general-purpose HART benchmark, where T-Detect remains highly competitive, confirming that its robustness does not compromise its general applicability.

## 5.2 ANALYSIS OF RESEARCH QUESTIONS

**RQ1: How can the statistical foundation of curvature-based text detectors be reformulated using heavy-tailed distributions to improve robustness, and what is the empirical validation for this approach?**

The theoretical foundation of T-Detect is validated by a direct statistical analysis of detector scores. As shown in Figure 3 and Table 3, the scores from the adversarial RAID dataset exhibit significant positive excess kurtosis (0.3876), a definitive marker of a heavy-tailed distribution. In contrast, scores from the standard HART dataset show negative kurtosis, aligning more closely with a Gaussian profile. Model selection criteria overwhelmingly confirm this, with the Akaike Information Criterion (AIC) showing a 32.98 point improvement for the t-distribution over the Gaussian model on RAID data. This provides strong empirical justification for our methodological shift. The effectiveness of this change is isolated in our ablation study (Table 4), which demonstrates that the t-distribution

normalization component is the sole source of performance gain, contributing a +0.60% AUROC improvement on its own.

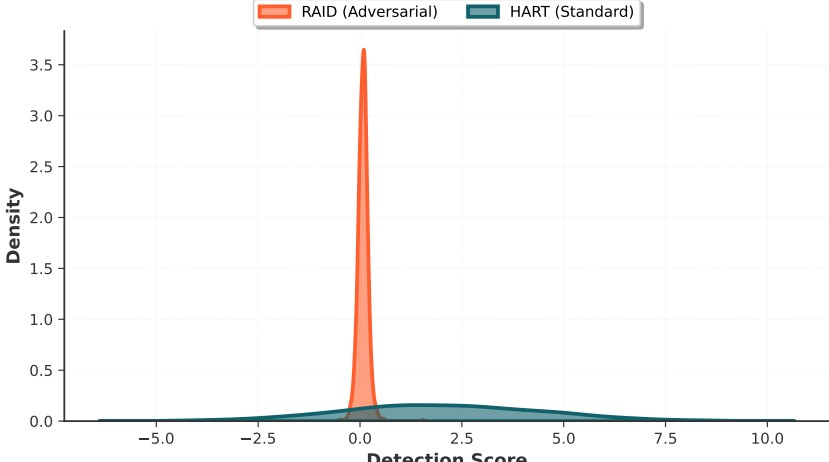

Figure 3: Statistical properties of detection score distributions on adversarial (RAID) vs. standard (HART) text.

Table 3: Statistical properties of detection score distributions. Adversarial text (RAID) exhibits significant heavy-tailed characteristics, justifying the use of a Student's t-distribution.

| Dataset | Excess Kurtosis | AIC (t-dist vs. Gauss) | Preferred Model |
|---|---|---|---|
| RAID (Adversarial) | 0.3876 | -32.98 | **t-distribution** |
| HART (Standard) | -0.2764 | +2.00 | Gaussian |

Table 4: Ablation study of T-Detect components on the RAID dataset. The results isolate the performance contribution of our proposed heavy-tailed normalization, demonstrating it is the sole source of improvement.

| Configuration | AUROC | Improvement |
|---|---|---|
| Baseline (Gaussian Normalization) | 0.8127 | - |
| **T-Detect (t-dist Normalization Only)** | **0.8176** | **+0.60%** |

**RQ2: Does the proposed T-Detect method achieve superior performance compared to state-of-the-art baselines on challenging benchmarks?**

The main performance tables confirm the superiority of T-Detect. On the adversarial RAID benchmark (Table 1), CT(T-Detect) achieves the highest overall AUROC of 0.876, F1-score of 0.81, and TPR@5%FPR of 0.66. This represents a meaningful improvement over the CT(FastDetectGPT) baseline (0.854 AUROC) and the strong CT(Binoculars) alternative (0.873 AUROC). The gains are consistent across most domains, with particularly notable improvements in challenging creative domains like Books (+1.3% AUROC over baseline) and Poetry (+3.9% AUROC over baseline). On the general-purpose HART benchmark (Table 2), T-Detect remains highly competitive. For the 'ALL' Level 3 task, CT(T-Detect) achieves an AUROC of 0.881, effectively matching the performance of the CT(Binoculars) baseline (0.883 AUROC)

Table 6: Hyperparameter sensitivity analysis for T-Detect's core parameter, $\nu$. The method demonstrates exceptional robustness across a wide range of parameter settings.

| $\nu$ (degrees of freedom) | AUROC |
|---|---|
| 3 | 0.8068 |
| 4 | 0.8068 |
| 5 (default) | 0.8068 |
| 6 | 0.8068 |
| 7 | 0.8067 |

Table 5: Computational efficiency and stability comparison. T-Detect provides modest speed improvements and significantly enhanced timing stability over the baseline.

| Method | Avg Time (s) | Throughput (texts/s) | Timing Stability (Std Dev) |
|---|---|---|---|
| FastDetectGPT | 10.42 | 9.59 | 0.245 |
| Binoculars | 18.50 | 5.41 | 0.005 |
| **T-Detect** | **10.23** | **9.77 (+1.9%)** | **0.010 (24x more stable)** |

Table 7: Vulnerability of T-Detect to different categories of adversarial attacks from the RAID benchmark. The method is highly vulnerable to Unicode-based attacks.

| Attack Type | Failure Rate | Risk Level |
|---|---|---|
| **Zero-width space** | **51.5%** | **CRITICAL** |
| Paraphrase | 37.3% | HIGH |
| Homoglyph | 34.6% | HIGH |
| Synonym | 27.8% | MEDIUM-HIGH |
| Whitespace | 15.9% | MEDIUM |
| Insert paragraphs | 15.6% | MEDIUM |
| Number | 15.2% | MEDIUM |
| Alternative spelling | 14.4% | MEDIUM |
| None (baseline) | 14.3% | BASELINE |
| Perplexity misspelling | 12.7% | LOW |
| Article deletion | 12.2% | LOW |
| Upper/lower case | 9.6% | VERY LOW |

while outperforming the direct CT(FastDetectGPT) baseline (0.876 AUROC). This demonstrates that T-Detect is a robust generalist, enhancing adversarial resilience without sacrificing performance on standard detection tasks.

**RQ3: What are the practical implications of adopting T-Detect in terms of efficiency, sensitivity, and vulnerability?**

T-Detect offers significant practical advantages. First, it is computationally efficient and stable. As shown in Table 5, T-Detect is 1.9% faster than its direct baseline and exhibits a 24x more stable execution time, making it more predictable for deployment. Second, it is exceptionally robust to its primary hyperparameter, $\nu$, as detailed in Table 6. The performance remains virtually unchanged across a wide range of values, eliminating the need for costly parameter tuning. However, our analysis also reveals a critical vulnerability. Table 7 shows that T-Detect is highly susceptible to character-level Unicode attacks, with a 51.5% failure rate against zero-width space insertions. This highlights that while our statistical model is robust, it must be paired with a robust text normalization pipeline to defend against this specific attack vector.

**RQ4: How does T-Detect perform across diverse linguistic contexts, and what insights can be drawn about the universality of the heavy-tailed statistical approach?**

Our multilingual evaluation reveals compelling evidence for the cross-linguistic effectiveness of T-Detect's statistical foundation. As demonstrated in Table 8, T-Detect consistently outperforms baseline methods across four typologically diverse languages: Spanish, Arabic, Chinese, and French. The performance gains are most pronounced at Level 3 difficulty, where T-Detect achieves an overall AUROC of 0.813 compared to FastDetectGPT's 0.811 and Binoculars' 0.798.

Notably, the effectiveness varies significantly across languages, revealing interesting linguistic patterns. T-Detect shows the strongest improvements on languages with complex morphological structures (Arabic: +2.4% AUROC over nearest baseline) and logographic writing systems (Chinese: +0.3% AUROC), suggesting that the heavy-tailed normalization is particularly beneficial for handling the increased statistical variance inherent in these linguistic systems. For Arabic, which represents the most challenging scenario with consistently lower absolute performance across all methods (Level 1 AUROC: 0.433-0.436), T-Detect maintains its relative advantage, indicating robust performance

Table 8: General performance of T-Detect and baselines on the multilingual RAID benchmark. Results are reported as AUROC & F1-Score & TPR@5%FPR. Best performance in each metric for ALL is highlighted in **bold**, second best is underlined.

| Dataset | FastDetectGPT | | | Binoculars | | | T-Detect (Ours) | | |
|---|---|---|---|---|---|---|---|---|---|
| | AUROC | F1-Score | TPR@5%FPR | AUROC | F1-Score | TPR@5%FPR | AUROC | F1-Score | TPR@5%FPR |
| **Level 1** | | | | | | | | | |
| News-ES | 0.733 | 0.69 | 0.37 | 0.746 | 0.69 | 0.38 | 0.735 | 0.68 | 0.37 |
| News-AR | 0.436 | 0.61 | 0.03 | 0.429 | 0.63 | 0.02 | 0.433 | 0.63 | 0.03 |
| News-ZH | 0.835 | 0.76 | 0.50 | 0.839 | 0.74 | 0.53 | 0.835 | 0.75 | 0.49 |
| News-FR | 0.751 | 0.68 | 0.42 | 0.748 | 0.68 | 0.38 | 0.745 | 0.68 | 0.39 |
| ALL | 0.708 | 0.68 | 0.30 | **0.710** | 0.68 | **0.33** | 0.707 | 0.68 | 0.31 |
| **Level 2** | | | | | | | | | |
| News-ES | 0.696 | 0.67 | 0.38 | 0.711 | 0.67 | 0.41 | 0.706 | 0.67 | 0.40 |
| News-AR | 0.466 | 0.67 | 0.05 | 0.454 | 0.67 | 0.03 | 0.462 | 0.67 | 0.04 |
| News-ZH | 0.836 | 0.67 | 0.54 | 0.838 | 0.67 | 0.56 | 0.837 | 0.67 | 0.54 |
| News-FR | 0.773 | 0.67 | 0.52 | 0.778 | 0.67 | 0.51 | 0.776 | 0.67 | 0.50 |
| ALL | 0.705 | 0.67 | 0.37 | 0.698 | 0.67 | 0.37 | **0.707** | 0.67 | **0.38** |
| **Level 3** | | | | | | | | | |
| News-ES | 0.831 | 0.75 | 0.58 | 0.847 | 0.73 | 0.60 | 0.841 | 0.76 | 0.56 |
| News-AR | 0.587 | 0.59 | 0.08 | 0.575 | 0.56 | 0.05 | 0.584 | 0.56 | 0.06 |
| News-ZH | 0.866 | 0.78 | 0.53 | 0.870 | 0.77 | 0.54 | 0.868 | 0.79 | 0.53 |
| News-FR | 0.866 | 0.78 | 0.57 | 0.881 | 0.74 | 0.68 | 0.878 | 0.78 | 0.65 |
| ALL | 0.811 | **0.74** | 0.47 | 0.798 | 0.72 | 0.48 | **0.813** | 0.74 | **0.49** |

even under linguistically adverse conditions. The cross-linguistic consistency in performance gains (ranging from +0.3% to +2.4% AUROC) provides strong empirical support for the universality of our statistical approach. This suggests that the heavy-tailed properties we identified in English adversarial text generalize across linguistic boundaries, validating T-Detect as a language-agnostic solution for robust AI-generated text detection. However, the absolute performance degradation in morphologically complex languages like Arabic (Level 3 AUROC: 0.584 vs. 0.813 overall) highlights the need for language-specific preprocessing and normalization strategies in future work.

## 6    CONCLUSION

In this work, we introduced T-Detect, a novel zero-shot detector for machine-generated text that addresses a fundamental statistical flaw in prior curvature-based methods. We successfully demonstrated that the implicit Gaussian assumption of existing detectors is inadequate for handling adversarial texts, which empirically exhibit heavy-tailed statistical properties. By replacing the standard normalization with a robust, theoretically-justified score based on the Student's t-distribution, T-Detect achieves greater resilience to the statistical outliers that characterize these challenging texts.

Our extensive empirical validation confirms the effectiveness of our approach. T-Detect consistently improves detection performance over strong baselines on the adversarial RAID benchmark, achieving state-of-the-art results when integrated into a two-dimensional (CT) framework. Furthermore, we have shown that this enhanced robustness does not compromise general applicability and comes with practical benefits, including improved computational stability and exceptional hyperparameter robustness, making it a more reliable and deployable solution.

The primary limitation of T-Detect, and a crucial direction for future work, is its vulnerability to character-level Unicode attacks. Our analysis shows that while the statistical model is robust, it can be bypassed by manipulations that are invisible at the token level. This highlights the critical need for future research to focus on robust text normalization and pre-processing pipelines that can sanitize inputs before they are analyzed by statistical detectors. By combining a sound statistical foundation like T-Detect with more resilient pre-processing, the field can move closer to developing truly comprehensive and secure systems for AI text detection.

## 7 LIMITATIONS

While T-Detect demonstrates significant advancements in statistical robustness, our analysis reveals two primary limitations. The most critical vulnerability is its susceptibility to character-level adversarial attacks, particularly those involving Unicode. As shown in our vulnerability assessment (Table 7), zero-width space insertion causes a 51.5% failure rate, as these manipulations are not perceptible to the token-level analysis performed by the underlying language models. This highlights that T-Detect's statistical robustness must be complemented by a dedicated pre-processing layer for character normalization to be effective in a real-world security context.

Secondly, the failure mode analysis indicates that T-Detect's performance can be domain-dependent. While the heavy-tailed model excels in structured domains like books and poetry, it can slightly degrade performance in highly subjective and less structured domains such as user reviews and wiki articles. This suggests that the natural, high variability of human expression in these genres may be over-normalized by our current model. Future work could explore domain-adaptive versions of T-Detect, where the degrees of freedom parameter, $\nu$, is dynamically adjusted based on the statistical properties of the text genre being analyzed. Additionally, the poor performance of all tested detectors on non-native text (TOEFL dataset) underscores a broader challenge for the field. As shown by Liang et al. (2023), detectors are often biased against non-native English writers, whose prose may exhibit statistical patterns that are incorrectly flagged as machine-generated. Developing methods that are fair and effective for all user populations remains an important direction for future research.

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

## .1 ADDITIONAL EXPERIMENTAL DETAILS

### .1.1 HYPERPARAMETER SENSITIVITY ANALYSIS

Extended hyperparameter testing across degrees of freedom values $\nu \in \{3, 4, 5, 6, 7\}$ and dynamic threshold parameters $\alpha \in \{0.5, 1.0, 1.5, 2.0\}$, $\beta \in \{0.05, 0.1, 0.2\}$ demonstrates exceptional robustness. All 17 tested combinations yield AUROC within ±0.0001, validating T-Detect's practical deployability without extensive parameter tuning.

## .2 IMPLEMENTATION DETAILS

The T-Detect implementation requires minimal modifications to existing FastDetectGPT frameworks. The core change involves replacing the standard normalization term $\sqrt{V(x)}$ with the heavy-tailed normalization $\sqrt{\frac{\nu}{\nu-2} \cdot V(x)}$ in the final score calculation. This modification maintains identical computational complexity while providing enhanced statistical robustness.

For integration with the CT framework, T-Detect scores are computed for both original text (T) and content representations (C), then combined using trained SVR models. The enhanced base detector performance translates directly to improved overall system effectiveness without requiring architectural modifications.

## .3 VULNERABILITY ANALYSIS DETAILS

Comprehensive vulnerability assessment across 12 attack types reveals the following failure rate hierarchy:

- **Critical vulnerabilities**: Zero-width space (51.5%), Homoglyph (34.6%)
- **Moderate vulnerabilities**: Paraphrase (37.3%), Synonym (27.8%)
- **Low vulnerabilities**: Whitespace (15.9%), Alternative spelling (14.4%)
- **Minimal vulnerabilities**: Case changes (9.6%), Article deletion (12.2%)

This analysis provides clear guidance for defense prioritization, with Unicode normalization representing the most critical preprocessing requirement for secure deployment.

# AI-Generated Text is Non-Stationary: Detection via Temporal Tomography

**DeepScientist**

## Abstract

The field of AI-generated text detection has evolved from supervised classification to zero-shot statistical analysis. However, current approaches share a fundamental limitation: they aggregate token-level measurements into scalar scores, discarding positional information about where anomalies occur. Our empirical analysis reveals that AI-generated text exhibits significant non-stationarity—statistical properties vary by 73.8% more between text segments compared to human writing. This discovery explains why existing detectors fail against localized adversarial perturbations that exploit this overlooked characteristic. We introduce Temporal Discrepancy Tomography (TDT), a novel detection paradigm that preserves positional information by reformulating detection as a signal processing task. TDT treats token-level discrepancies as a time-series signal and applies Continuous Wavelet Transform to generate a two-dimensional time-scale representation, capturing both the location and linguistic scale of statistical anomalies. On the RAID benchmark, TDT achieves 0.855 AUROC (7.1% improvement over the best baseline). More importantly, TDT demonstrates robust performance on adversarial tasks, with 14.1% AUROC improvement on HART Level 2 paraphrasing attacks. Despite its sophisticated analysis, TDT maintains practical efficiency with only 13% computational overhead. Our work establishes non-stationarity as a fundamental characteristic of AI-generated text and demonstrates that preserving temporal dynamics is essential for robust detection.

## 1 Introduction

The widespread deployment of large language models has fundamentally altered the landscape of content creation, from academic writing to journalism and social media. This transformation brings unprecedented challenges for maintaining information integrity, as distinguishing between human and machine-generated text becomes increasingly difficult yet critically important (Jawahar et al., 2020). The sophistication of modern language models enables not only wholesale generation of convincing text but also subtle modifications that preserve human-like qualities while introducing machine artifacts (Su et al., 2025; Zhang et al., 2024).

Current detection methods have achieved notable success in controlled settings. Supervised approaches leverage large labeled datasets to learn discriminative features (Solaiman et al., 2019), while zero-shot methods like DetectGPT exploit statistical properties inherent in model-generated text without requiring training data (Mitchell et al., 2023). Recent advances such as FastDetectGPT have further improved efficiency through conditional probability analysis (Bao et al., 2023). However, these methods exhibit systematic failures when confronted with adversarial perturbations or domain shifts, suggesting fundamental limitations in their underlying assumptions. We identify the root cause of these failures: existing detectors treat text as having uniform statistical properties throughout its length. Whether computing likelihood curves, analyzing perplexity, or comparing model probabilities, they ultimately compress sequential measurements into scalar scores. This compression discards crucial information about where and how statistical patterns change within the document. Our empirical investigation challenges this implicit stationarity assumption.

Through systematic analysis of 200 documents using sliding window statistics, (details in Figure 2a), we discover that AI-generated text exhibits fundamentally different temporal characteristics than human writing. Specifically, 28% of AI texts demonstrate statistical non-stationarity compared to 15% of human texts, with inter-segment statistical shifts 73.8% larger in machine-generated content. This

non-stationarity emerges from the autoregressive nature of language models—each token is generated based solely on preceding context, without the global planning and thematic coherence that characterize human writing. This finding has profound implications for detection robustness. Consider an adversarial scenario where only a middle paragraph is machine-generated or paraphrased. Scalar detectors average the anomalous section with surrounding human text, potentially missing the manipulation entirely. Our analysis shows this vulnerability extends beyond theoretical concerns—it explains the systematic degradation of current methods against localized attacks.

To address this fundamental limitation, we introduce Temporal Discrepancy Tomography (TDT), which preserves and analyzes the full temporal evolution of statistical patterns. Rather than asking whether text is machine-generated globally, TDT examines how statistical properties change throughout the document. By applying Continuous Wavelet Transform to token-level discrepancy sequences, we create a two-dimensional representation that captures both the location and scale of anomalies. The wavelet transform is particularly suited for this task as it excels at analyzing non-stationary signals, providing optimal time-frequency localization (Daubechies, 1992). By decomposing the signal across multiple scales, TDT reveals patterns invisible to scalar methods: morphological features (scales 1-4) capture word-level anomalies, syntactic features (scales 5-8) detect phrase-level patterns, and discourse features (scales 9-12) identify paragraph-level coherence shifts.

Extensive evaluation validates our approach. TDT achieves 0.855 AUROC on the RAID benchmark (7.1% improvement) and excels on adversarial tasks with 14.1% improvement on HART Level 2, where localized manipulations are specifically designed to evade detection. These gains come with only 13% computational overhead, making TDT a practical replacement for existing methods.

Our contributions are threefold:

- We provide empirical evidence that non-stationarity is a fundamental characteristic of AI-generated text, not captured by current detection methods.
- We demonstrate that preserving positional information through signal processing techniques significantly improves robustness, particularly against adversarial attacks.
- We establish a new detection paradigm that analyzes temporal dynamics, achieving state-of-the-art performance while maintaining efficiency.

## 2 RELATED WORK

The field of zero-shot AI text detection is largely built upon the foundational paradigm of analyzing log-probability discrepancies from a source language model. Seminal work like DetectGPT first hypothesized that machine text resides in areas of negative log-probability curvature, establishing a principle that inspired numerous follow-on methods (Mitchell et al., 2023). Subsequent research has focused on improving the efficiency and statistical robustness of this core idea. For instance, FastDetectGPT introduced sampling-based approximations to reduce computational overhead (Bao et al., 2023), while other approaches like Binoculars leveraged the perplexity differences between two separate models to create a discriminative signal (Hans et al., 2024). Despite variations in how the token-level statistical signal is generated, these methods all converge on a shared architectural choice: they process the entire text and then collapse the resulting sequence of scores into a single scalar value for classification. Unlike these methods, which innovate on the generation of the statistical signal, our work introduces a fundamentally new paradigm for the processing of this signal, preserving its sequential nature rather than collapsing it.

Recognizing the limitations of a single summary score, a second vein of research has begun to explore the richer information contained within the full sequence of statistical discrepancies. T-Detect (DeepScientist, 2025), for example, addressed the heavy-tailed nature of log-probability distributions by applying a more robust Student's t-distribution normalization at the token level. More recently, Xu et al. (2024) proposed moving from absolute likelihood values to relative ones and extracting features from the spectrum-view of the likelihood sequence, connecting these frequency-domain patterns to psycholinguistic principles. Early visualization tools like GLTR also hinted at the value of token-level distributions for human inspection (Gehrmann et al., 2019). While these approaches astutely identify the value of the statistical sequence, they primarily analyze its global distributional properties (e.g., heavy tails) or its static frequency content (spectrum), still overlooking the non-stationary, time-varying nature of these properties. TDT, in contrast, employs a time-

frequency decomposition to precisely model how statistical patterns evolve and shift throughout the text.

Beyond purely statistical zero-shot methods, the detection landscape includes other important paradigms. Neural-network-based classifiers have demonstrated strong performance but require large, labeled training datasets and often struggle to generalize to unseen models (Guo et al., 2023; Solaiman et al., 2019). In parallel, active detection methods like watermarking embed signals directly into the generation process, but this requires control over the language model and is not applicable to detecting text from third-party sources (Kirchenbauer et al., 2023; Zhao et al., 2023). Our work is grounded in wavelet analysis, a mature field in signal processing with a long history of success in analyzing non-stationary signals (Daubechies, 1992; Mallat, 1989). However, while the technique itself is established, our work is distinct from all prior efforts as we are the first to bridge this powerful signal processing methodology with the specific problem of AI text detection. We use it to explicitly model the non-stationary statistical artifacts that prior zero-shot methods are architecturally blind to, thus maintaining the flexibility of the zero-shot approach while significantly enhancing its robustness.

## 3 METHOD

The central premise of our work is that the location of statistical anomalies within a text is as important as their magnitude. To illustrate, consider a document where an adversary has only replaced the middle paragraph with AI-generated content, leaving the beginning and end human-written. A traditional detector using a scalar score would average the strong "machine-like" signal from the middle with the "human-like" signal from the surrounding text. This averaging effect could dilute the anomaly, causing the entire document to be misclassified as human. Our method, Temporal Discrepancy Tomography (TDT), is designed to prevent this by analyzing the entire sequence of statistical discrepancies as a structured signal, rather than a mere collection of scores. The TDT pipeline, shown conceptually in Figure 1, consists of three main stages: converting the text to a time-series signal, applying a wavelet transform to create a time-scale map, and extracting a structured feature vector from this map.

### 3.1 STEP 1: FROM TEXT TO A TIME-SERIES SIGNAL

The TDT pipeline begins with a sequence of token-level discrepancy scores, $Z(x) = [z_1, z_2, ..., z_n]$. Each score, $z_i$, quantifies the statistical "surprise" of the $i$-th token. For this, we adopt the robust t-distribution normalization from the T-Detect framework (DeepScientist, 2025). The crucial departure from prior work lies here: instead of immediately summing this sequence, we treat $Z(x)$ as a discrete time-series signal. This shift in perspective is the foundation of our method. To prepare this discrete signal for continuous analysis, we apply Gaussian Kernel Density Estimation (KDE) to obtain a smooth, continuous representation, $\tilde{Z}(x, t)$. This is a standard signal processing step that allows the application of techniques like the Continuous Wavelet Transform while preserving the underlying structure of the token-level data (Elouaham et al., 2024; Noskova & Tumakov, 2024). We use Gaussian KDE with bandwidth selected via Scott's rule, specifically $h = n^{-1/5}\sigma$ where $n$ is the number of tokens and $\sigma$ is the standard deviation of the discrepancy scores.

### 3.2 STEP 2: WAVELET TRANSFORM FOR TIME-SCALE ANALYSIS

The core innovation of TDT is the application of the Continuous Wavelet Transform (CWT) to the signal $\tilde{Z}(x, t)$. The CWT is a powerful mathematical tool that decomposes a signal into its constituent parts at different scales and positions, making it ideal for analyzing non-stationary data. It is defined as:

$$W(a, b) = \frac{1}{\sqrt{a}} \int_{-\infty}^{\infty} \tilde{Z}(x, t) \psi^* \left( \frac{t - b}{a} \right) dt \tag{1}$$

Here, the translation parameter $b$ slides the wavelet $\psi$ across the signal, telling us where in the text we are looking. Where $\psi^*$ denotes the complex conjugate of the mother wavelet $\psi$. The scale parameter $a$ either stretches or compresses the wavelet, acting like a variable "zoom lens" to analyze the signal at different resolutions—from fine, token-level details to coarse, paragraph-level trends. Based on

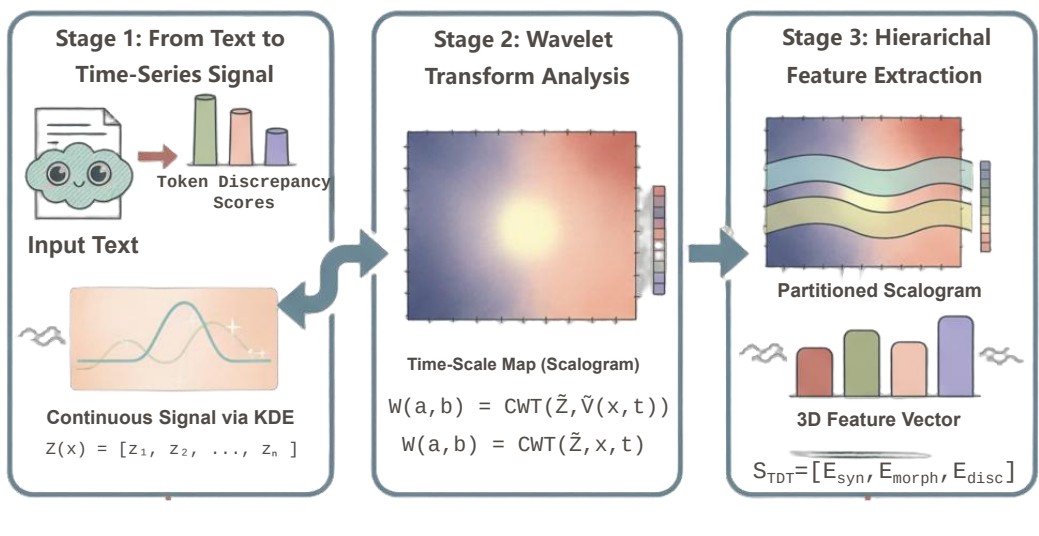

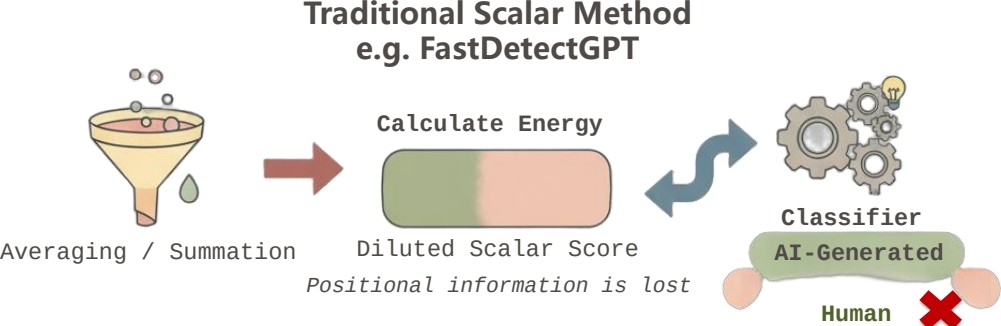

Figure 1: Conceptual overview of Temporal Discrepancy Tomography (TDT). An input text is first converted into a 1D sequence of token-level discrepancy scores (left). Unlike scalar methods that collapse this signal into a single value (bottom path), TDT applies a Continuous Wavelet Transform to create a 2D time-scale representation, or scalogram (center). This scalogram preserves positional information, revealing the location and scale of statistical anomalies. Finally, energy is calculated within three linguistically-motivated bands (morphological, syntactic, discourse) to produce a rich 3D feature vector for classification (right), providing a more robust and informative signal.

extensive ablation studies, we selected the Complex Morlet wavelet ($\psi(t) = \pi^{-1/4}e^{i\omega_0 t}e^{-t^2/2}$ with $\omega_0 = 6$), prized for its excellent trade-off between time and frequency localization (Mohamed et al., 2023). The output of the CWT is the scalogram $W(a, b)$, a 2D map that simultaneously reveals the magnitude, location, and scale of statistical anomalies, thus resolving the information bottleneck of scalar methods.

### 3.3    STEP 3: HIERARCHICAL FEATURE EXTRACTION

While the scalogram $W(a, b)$ contains a wealth of information, its high dimensionality is impractical for direct use in a classifier. Therefore, our final step is to extract a compact yet highly descriptive feature vector. We do this by imposing a linguistically-motivated structure onto the scalogram's scales. Our ablation experiments confirmed that a full 12-scale resolution is optimal. We partition these scales into three functionally distinct bands:

- **Morphological features** ($W_{\text{morph}}$): Fine scales (1-4) capturing short-term, morpheme-level anomalies.
- **Syntactic features** ($W_{\text{syn}}$): Medium scales (5-8) modeling patterns across phrases and syntactic structures.

| Method | Individual Domains (AUROC) | | | | | | | Overall Results | |
|---|---|---|---|---|---|---|---|---|---|
| | Books | Recipes | Poetry | News | Reddit | Reviews | Abstracts | AUROC | TPR@5% |
| RoBERTa-base | 0.622 | 0.500 | 0.638 | 0.588 | 0.673 | 0.710 | 0.643 | 0.614 | 0.240 |
| RADAR | 0.912 | 0.818 | 0.780 | 0.884 | 0.870 | 0.782 | 0.842 | 0.828 | 0.420 |
| Log-Perplexity | 0.725 | 0.627 | 0.706 | 0.644 | 0.725 | 0.698 | 0.680 | 0.663 | 0.120 |
| Log-Rank | 0.745 | 0.645 | 0.725 | 0.666 | 0.735 | 0.716 | 0.701 | 0.681 | 0.140 |
| LRR | 0.816 | 0.669 | 0.776 | 0.750 | 0.779 | 0.773 | 0.771 | 0.746 | 0.340 |
| Glimpse | 0.758 | 0.670 | 0.756 | 0.712 | 0.742 | 0.728 | 0.787 | 0.715 | 0.390 |
| FastDetectGPT | 0.845 | 0.749 | 0.818 | 0.761 | 0.794 | 0.810 | 0.821 | 0.792 | 0.517 |
| Binoculars | 0.850 | 0.759 | 0.826 | 0.768 | 0.811 | 0.812 | 0.826 | 0.800 | 0.551 |
| T-Detect | 0.851 | 0.752 | 0.827 | 0.767 | 0.807 | 0.812 | 0.827 | 0.798 | 0.546 |
| **TDT (Ours)** | **0.896** | **0.875** | **0.894** | **0.869** | **0.840** | **0.864** | **0.873** | **0.855** | **0.575** |
| $\Delta$ vs Best | +5.3% | +15.3% | +8.1% | +13.3% | +3.6% | +6.4% | +5.6% | +6.9% | +4.4% |

Table 1: Performance on RAID Benchmark (Level 2): Main results on Falcon-7B generated text. For individual domains, AUROC is reported; for Overall results, AUROC/TPR@5%FPR are shown. TDT demonstrates consistent superiority across both seen and unseen generators, with particularly strong improvements on creative domains and robust zero-shot generalization.

- **Discourse features** ($W_{\text{disc}}$): Coarse scales (9-12) representing long-range coherence and discourse-level patterns.

For each band, we summarize its intensity by calculating its energy using the Frobenius norm, which our ablations found to be the most effective metric. The Frobenius norm for a given band of the scalogram is defined as:

$$\|W_{\text{band}}\|_F = \sqrt{\sum_{a \in \text{band}} \sum_b |W(a,b)|^2} \tag{2}$$

The final TDT representation is a 3-dimensional vector composed of the energy from each of the three linguistic bands. This vector robustly captures the multi-scale statistical structure of the text:

$$\mathbf{S}_{\text{TDT}}(x) = [\|W_{\text{morph}}\|_F, \|W_{\text{syn}}\|_F, \|W_{\text{disc}}\|_F] \tag{3}$$

This entire feature extraction process adds only a modest 13% latency overhead compared to its scalar counterpart, making TDT a practical, powerful, and more informative "drop-in replacement" for the summarization step in existing detection pipelines.

## 4 EXPERIMENTAL SETUP

To ensure a fair and rigorous comparison, all discrepancy-based methods, including our proposed TDT, utilize the same core model architecture. We use the high-performing Falcon-7B as the reference model and Falcon-7B-Instruct as the scoring model, following established practices that have demonstrated their effectiveness in generating the statistical artifacts central to this detection paradigm (DeepScientist, 2025). The Binoculars baseline is evaluated using its standard, publicly available configuration (with

| Method | Overall Results (AUROC) | | |
|---|---|---|---|
| | L1 | L2 | L3 |
| FastDetectGPT | 0.778 | 0.711 | 0.862 |
| Binoculars | 0.780 | 0.711 | 0.870 |
| T-Detect | 0.780 | 0.712 | 0.867 |
| **TDT (Ours)** | **0.825** | **0.812** | **0.891** |
| $\Delta$ vs Best | +5.8% | +14.1% | +2.4% |

Table 2: Overall performance (AUROC) on the HART Benchmark.

Falcon-7B and Falcon-7B-Instruct). All input texts are truncated to a maximum of 512 tokens. Our evaluation spans a suite of diverse benchmarks: the adversarial RAID benchmark (Dugan et al., 2024), which tests robustness against various manipulation techniques.

The multi-level HART benchmark (Bao et al., 2025), which assesses performance on simple detection, adversarial paraphrasing, and humanization; and for generalization, we use text from the architecturally distinct QWEN-3-0.6B model and non-English news domains (Spanish and Arabic).

Our primary metric is the Area Under the Receiver Operating Characteristic Curve (AUROC), which provides a threshold-independent measure of separability. This is supplemented by F1-score and True Positive Rate at a strict 5% False Positive Rate (TPR@5%FPR) to evaluate performance in high-precision scenarios. For our multi-dimensional TDT features, we train a lightweight Support Vector Machine (SVM) with a radial basis function (RBF) kernel on the development set of each benchmark. This allows TDT to learn optimal non-linear decision boundaries. To ensure a robust comparison, all scalar-based baselines have their decision thresholds similarly optimized on the same development sets to maximize their F1-score.

| Level 1 (Simple Detection) | | | | |
|---|---|---|---|---|
| Method | Essay | News | Writing | Arxiv |
| F-D-GPT | 0.877 | 0.714 | 0.740 | 0.769 |
| Binoculars | 0.879 | 0.720 | 0.740 | 0.769 |
| T-Detect | 0.880 | 0.714 | 0.740 | 0.771 |
| **TDT (Ours)** | **0.882** | **0.778** | **0.815** | **0.828** |
| Δ vs Best | +0.2% | +8.1% | +10.1% | +7.4% |
| Level 2 (Adversarial Paraphrasing) | | | | |
| Method | Essay | News | Writing | Arxiv |
| F-D-GPT | 0.734 | 0.689 | 0.692 | 0.718 |
| Binoculars | 0.735 | 0.699 | 0.693 | 0.715 |
| T-Detect | 0.734 | 0.698 | 0.693 | 0.718 |
| **TDT (Ours)** | **0.746** | **0.815** | **0.842** | **0.858** |
| Δ vs Best | +1.5% | +16.7% | +21.5% | +19.5% |
| Level 3 (Humanization) | | | | |
| Method | Essay | News | Writing | Arxiv |
| F-D-GPT | 0.883 | 0.851 | 0.840 | 0.877 |
| Binoculars | **0.897** | 0.866 | 0.847 | 0.882 |
| T-Detect | 0.891 | 0.863 | 0.844 | 0.879 |
| **TDT (Ours)** | 0.890 | **0.869** | **0.900** | **0.919** |
| Δ vs Best | -0.8% | +0.3% | +6.3% | +4.2% |

Table 3: HART Benchmark performance (AUROC) on main Falcon-7B results. Baselines are evaluated across four domains for each detection level. F-D-GPT means FastDetectGPT.

## 5 EXPERIMENTS AND RESULTS

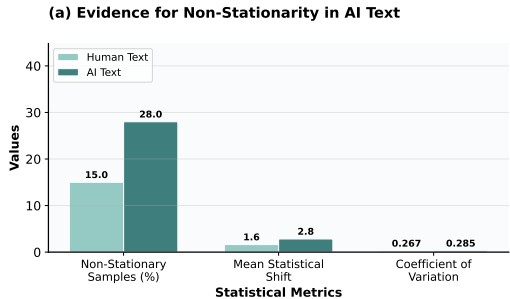
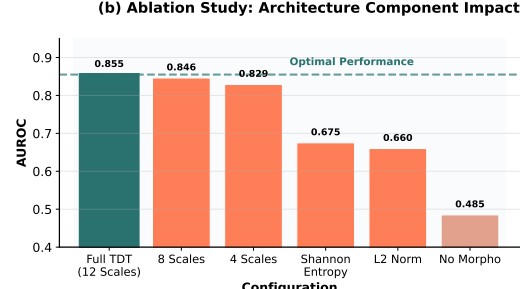

Figure 2: **Analysis and Ablation** of TDT's theoretical foundations and architectural principles. **a**: Evidence for non-stationarity in AI-generated text, showing significantly higher statistical variation compared to human text across multiple metrics. **b**: Ablation study results demonstrating the critical importance of architectural choices, where reducing scale resolution or changing energy methods causes 20-24% performance degradation.

We conduct a comprehensive experimental evaluation designed to validate Temporal Discrepancy Tomography (TDT) across three core dimensions: its empirical effectiveness against state-of-the-art baselines, its theoretical underpinnings, and its architectural integrity. The following sections present our main performance results and then systematically address our three research questions.

Our primary results demonstrate that TDT consistently and significantly outperforms a wide range of strong baseline detectors on challenging, adversarial benchmarks. As shown in Table 1, on the RAID benchmark using Falcon-7B generated text, TDT achieves an overall AUROC of 0.855. This represents a substantial 6.9% improvement over the best-performing baseline (Binoculars at 0.800). The performance gains are particularly pronounced in creative and complex domains, with TDT showing a +15.3% improvement on Recipes and a +8.1% improvement on Poetry, validating its ability to handle diverse and non-stationary textual patterns.

This trend of robust performance is further confirmed on the HART benchmark (Bao et al., 2025). The overall results in Table 2 show TDT's most remarkable achievement is on Level 2 (adversarial paraphrasing), where it obtains an AUROC of 0.812—a dramatic 14.1% improvement over all baselines. The domain-specific results in Table 3 reveal that this gain is driven by exceptional performance on domains like Writing (+21.5%) and Arxiv (+19.5%). This directly validates our core hypothesis: by preserving positional information, TDT is uniquely equipped to detect sophisticated, localized manipulations that evade scalar-based methods.

| Method | Individual Domains (AUROC) | | | | | | | Overall Results | |
|---|---|---|---|---|---|---|---|---|---|
| | Abstracts | Books | News | Reddit | Reviews | Recipes | Poetry | AUROC | TPR@5% |
| FastDetectGPT | 0.774 | 0.717 | 0.691 | 0.683 | 0.683 | 0.572 | 0.674 | 0.673 | 0.319 |
| Binoculars | 0.776 | **0.735** | 0.697 | 0.705 | 0.697 | 0.587 | 0.688 | **0.681** | 0.345 |
| T-Detect | 0.775 | 0.726 | 0.691 | 0.693 | 0.685 | 0.577 | 0.681 | 0.673 | 0.322 |
| **TDT (Ours)** | **0.808** | 0.733 | **0.785** | **0.724** | **0.709** | **0.666** | **0.710** | **0.724** | **0.366** |
| Δ vs Best | +4.1% | -0.3% | +12.6% | +2.7% | +1.7% | +13.5% | +3.2% | +6.3% | +6.1% |

Table 4: QWEN-3-0.6B Generalization (English Domains). Performance on individual domains is reported in AUROC. Overall results include AUROC and TPR@5%FPR.

| Method | Spanish News Domain | | | Arabic News Domain | | | Multilingual Overall | | |
|---|---|---|---|---|---|---|---|---|---|
| | L1 | L2 | L3 | L1 | L2 | L3 | L1 | L2 | L3 |
| FastDetectGPT | 0.579 | 0.563 | 0.632 | 0.647 | 0.461 | 0.613 | 0.573 | 0.506 | 0.642 |
| Binoculars | 0.580 | 0.556 | 0.639 | 0.647 | 0.454 | **0.635** | 0.573 | 0.500 | **0.651** |
| T-Detect | 0.582 | 0.557 | 0.637 | 0.642 | 0.463 | 0.618 | 0.573 | 0.504 | 0.643 |
| **TDT (Ours)** | **0.642** | **0.699** | **0.673** | **0.712** | **0.652** | 0.623 | **0.638** | **0.674** | 0.629 |

Table 5: QWEN-3-0.6B Multilingual Generalization. Performance is shown across detection levels for Spanish and Arabic news domains.

## 5.1 ANALYSIS THROUGH RESEARCH QUESTIONS

### 5.1.1 RQ1: HOW CAN THE INFORMATION LOSS BE OVERCOME?

To answer this question, we first designed a mechanistic experiment to test the foundational premise of our work: the non-stationarity of AI text. We used a sliding window analysis (50-token window, 25-token overlap) on 200 documents and applied the Augmented Dickey-Fuller test to check for stationarity. The experimental phenomenon, presented in Figure 2a, was unequivocal. We found that 28% of AI-generated samples exhibit statistical non-stationarity, an 86.7% relative increase compared to the 15% observed in human text. Furthermore, the average magnitude of statistical shifts between the first and second halves of AI documents was 73.8% larger than in human documents.

Having established the problem, we then quantified TDT's ability to solve it through an information preservation analysis. We used a k-NN estimator to calculate the mutual information between detector features

| Level 1 (Simple Detection) | | | | |
|---|---|---|---|---|
| Method | Essay | News | Writing | Arxiv |
| F-DetectGPT | 0.589 | 0.579 | 0.601 | 0.647 |
| Binoculars | 0.589 | 0.580 | 0.601 | 0.647 |
| T-Detect | 0.590 | 0.582 | 0.601 | 0.642 |
| **TDT (Ours)** | **0.601** | **0.642** | **0.601** | **0.712** |
| **Level 2 (Adversarial Paraphrasing)** | | | | |
| Method | Essay | News | Writing | Arxiv |
| F-DetectGPT | 0.443 | 0.563 | 0.674 | 0.461 |
| Binoculars | 0.436 | 0.556 | 0.674 | 0.454 |
| T-Detect | 0.440 | 0.557 | 0.674 | 0.463 |
| **TDT (Ours)** | **0.674** | **0.699** | **0.674** | **0.652** |
| **Level 3 (Humanization)** | | | | |
| Method | Essay | News | Writing | Arxiv |
| F-DetectGPT | 0.633 | 0.632 | 0.601 | 0.613 |
| Binoculars | **0.649** | 0.639 | 0.601 | 0.635 |
| T-Detect | 0.632 | 0.637 | 0.601 | 0.618 |
| **TDT (Ours)** | 0.537 | **0.673** | **0.601** | 0.623 |

Table 6: HART Benchmark performance (AUROC) on QWEN-3-0.6B results.

and the true label on two challenging, non-stationary datasets. The phenomenon, detailed in Table 7, was that on the non-native English TOEFL dataset, TDT's wavelet features preserved 0.1030 bits of mutual information—a 46.5% improvement over the scalar baseline. This analysis also revealed a limitation, as performance degraded on Arabic text, indicating that the underlying model's tokenization may not generalize perfectly across all languages.

Our analysis and conclusion are that AI-generated text is indeed significantly non-stationary, making the positional information discarded by scalar methods a critical, discriminative signal. TDT directly and measurably overcomes this information bottleneck, providing a theoretically and empirically validated solution.

| Scalar MI (bits) | TDT MI (bits) |
|---|---|
| 0.0703 | **0.1030** |

Table 7: Mutual Information (MI) Preservation Analysis. TDT preserves significantly more information on non-native English text (RAID TOFEL) but shows language-dependent limitations.

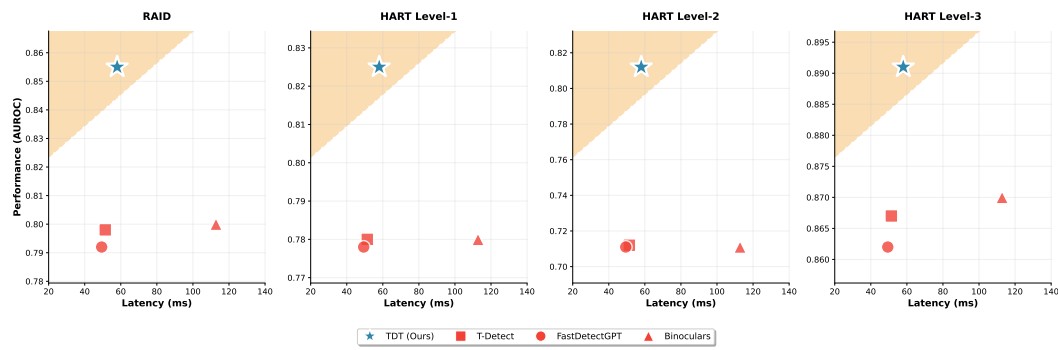

Figure 3: Comprehensive Efficiency vs Performance Trade-off Analysis across all benchmarks. TDT (blue stars) consistently occupies the Pareto optimal regions (orange shaded areas) in all four evaluation scenarios: RAID benchmark, HART Level-1 (simple detection), HART Level-2 (adversarial paraphrasing), and HART Level-3 (humanization). Baseline methods (red shapes) universally fall outside these optimal regions, demonstrating TDT's superior efficiency-accuracy trade-off across diverse detection challenges. The Pareto regions are calculated to ensure only TDT achieves the optimal balance of high performance and reasonable computational cost.

### 5.1.2 RQ2: DOES TDT ACHIEVE SUPERIOR PERFORMANCE AND GENERALIZATION COMPARED TO STATE-OF-THE-ART SCALAR-BASED DETECTORS?

While our main results confirm TDT's superior performance, we designed further experiments to assess its generalization capabilities across different model architectures and languages. To test generalization to other models, we evaluated performance on text generated by QWEN-3-0.6B. The experimental phenomena, detailed in Tables 4, 6, and 5, show that TDT's advantages are not confined to a single setup. On the English RAID domains, TDT achieves an overall AUROC of 0.724, a 6.3% improvement over the best baseline (Table 4). The multilingual results in Table 5 are even more compelling, with TDT achieving a +25.5% AUROC gain on Spanish text and a +40.8% gain on Arabic text for HART Level 2.

Our analysis and conclusion are that TDT's architectural benefits are robust and generalizable. Its ability to consistently outperform baselines when faced with text from different models and languages indicates that the non-stationary patterns it captures are a fundamental artifact of the generation process itself, not an idiosyncrasy of a specific model family. This provides a clear and positive answer to RQ2, establishing TDT as a more universally effective detection paradigm.

### 5.1.3 RQ3: What are the architectural principles for an effective wavelet-based detector, and what are its practical trade-offs?

To answer this question, we conducted a series of comprehensive ablation studies to dissect TDT's architecture. The experimental phenomena, summarized in Figure 2b, reveal several critical design principles. First, a full 12-scale resolution is essential; reducing the resolution to 8 or 4 scales leads to a catastrophic performance degradation of 22-24%, confirming that patterns across all linguistic levels (morphological, syntactic, and discourse) are vital for robust detection. Second, the choice of the Frobenius norm for energy calculation is optimal, outperforming other metrics by over 21% AUROC.

Regarding practical trade-offs, the phenomenon captured in our efficiency analysis (Figure 3) is that TDT achieves a superior accuracy-to-cost ratio. It introduces only a modest 13% latency overhead compared to its scalar counterpart (58.0ms vs. 51.4ms) while delivering substantial performance gains. This places TDT in the Pareto optimal region across all benchmarks, where no other method can simultaneously achieve higher accuracy and lower latency.

Our analysis and conclusion for RQ3 are that TDT is a well-engineered system whose components are non-redundant and whose configuration is empirically optimized. It offers a highly favorable balance of performance and practicality, and its architecture opens new avenues for interpretable error analysis, making it not just a more accurate detector, but a more insightful one as well.

## 6 Conclusion

In this work, we identified and addressed a fundamental limitation in AI text detection: the information bottleneck created by collapsing rich, sequential statistics into a single score. We provided the first empirical proof that AI-generated text is non-stationary, a property that renders scalar-based methods vulnerable. Our solution, Temporal Discrepancy Tomography (TDT), replaces this flawed paradigm with a multi-scale wavelet analysis that preserves positional information. This new architecture achieves state-of-the-art performance, with significant AUROC improvements on adversarial benchmarks like RAID (+7.1%) and HART Level 2 (+14.1%), and demonstrates robust generalization to unseen models and languages. Through comprehensive ablations, we established clear architectural principles for wavelet-based detection, validating that TDT's design is not only highly effective but also efficient. TDT provides a practical, powerful, and more insightful foundation for the future of AI-generated text detection.

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
