# OpenReview forum: "DeepScientist: Advancing Frontier-Pushing Scientific Findings Progressively"
_ICLR.cc/2026/Conference — ICLR 2026 Poster_

### Official Review · Reviewer_oPiU · 2025-10-23

**Soundness:** 3
**Presentation:** 2
**Contribution:** 3
**Rating:** 4
**Confidence:** 4

**Summary:**

The paper introduces DeepScientist, a system designed to automate goal-oriented scientific discovery. Experimental results suggest that DeepScientist can identify novel methods that outperform state-of-the-art (SOTA) approaches across three frontier AI tasks.

**Strengths:**

**Motivation:** The motivation is strong. Automating scientific discovery is a compelling direction, especially given the recent advancements in LLMs. It is valuable to explore how such systems can accelerate the pace of scientific progress.

**Substance:** The paper presents experiments on three distinct tasks to demonstrate the capabilities of DeepScientist. The generated research papers appear coherent and reasonably well-written, suggesting that the system can meaningfully contribute to scientific exploration.

**Weaknesses:**

**Presentation:** The baselines in Table 2 lacks sufficient explanation and context. Each baseline should be clearly introduced and described in the Experimental Setup section. Additionally, the overall pipeline of DeepScientist appears very similar to existing systems such as AI Scientist and Zochi. A more detailed comparison would help highlight the genuine innovations and unique contributions of DeepScientist. For instance, what key design elements distinguish DeepScientist from these prior works? Moreover, the paper omits a related work, “AlphaGo Moment for Model Architecture Discovery”, which follows a conceptually similar pipeline and should be discussed.

**Evaluation:** The evaluation methodology seems somewhat limited. The results in Table 2 rely on a single automated reviewer system, which raises concerns about robustness. It would strengthen the paper to cross-validate the results using multiple reviewer systems or complementary evaluation approaches. Notably, systems such as Zochi also report high acceptance rates when evaluated by automatic reviewers, yet here its acceptance rate is listed as 0%—this discrepancy deserves further analysis or justification.

**Questions:**

- Line 319: The paper mentions that five research papers generated by DeepScientist were evaluated. How were these papers selected—randomly or based on performance? Given that automatic evaluation is not resource-intensive, why not evaluate all successfully generated papers for a more comprehensive assessment?
- Line 859: The authors note that all experimental results were manually inspected by human supervisors to ensure authenticity. How extensive was this human involvement? Quantifying the amount of human effort would help assess the true level of automation achieved.
- System Design: DeepScientist uses Gemini for high-level planning and Claude for implementation and execution. Did the authors observe any performance differences when using stronger or weaker models for these roles? In other words, how does the planner–executor balance affect the overall efficiency and quality of the discovery process? Which component benefits more from having a stronger model?

---

> ### Author Response · Authors · 2025-11-17
> **Response - 1**
>
> > **Presentation:** The baselines in Table 2 lacks sufficient explanation and context. Each baseline should be clearly introduced and described in the Experimental Setup section.
>
>
>
> We sincerely appreciate the reviewer’s valuable feedback on the presentation quality. We fully agree that clearly explaining the baselines and comparing them against prior work is essential for highlighting the contributions of this paper.
>
> In the revised manuscript, we substantially expanded **Section 4.1**, including:
>
> - A detailed introduction of **every baseline system** appearing in Table 2 (AI Scientist, Zochi, CycleResearcher, AI-Researcher, etc.), including their core designs, LLMs used, and the exact configurations adopted in our experiments;
> - Approximately **half a page of additional content** describing, for each of the three frontier tasks, the **human SOTA methods**, the **limitations DeepScientist identified** in these human-crafted methods, and **how DeepScientist’s improvements achieve the final performance**.
>    These are now explicitly linked to Table 2 to help readers directly connect baselines with outcomes.
>
> We believe these revisions significantly improve the clarity and readability of Table 2 and make DeepScientist’s improvements over existing systems more evident.
>
>
>
> > Additionally, the overall pipeline of DeepScientist appears very similar to existing systems such as AI Scientist and Zochi. A more detailed comparison would help highlight the genuine innovations and unique contributions of DeepScientist. For instance, what key design elements distinguish DeepScientist from these prior works? Moreover, the paper omits a related work, “AlphaGo Moment for Model Architecture Discovery”, which follows a conceptually similar pipeline and should be discussed.
>
>
>
> Regarding baseline systems, we have expanded both the **Related Work (Section 2)** and **Method (Section 3)** to include deeper discussion and comparison with **AI Scientist**, **Zochi**, and **the paper “AlphaGo Moment for Model Architecture Discovery”** [1]. These additions clarify the fundamental design differences between DeepScientist and these prior systems.
>
> Our analysis shows that existing systems (AI Scientist, Zochi, etc.) have structural limitations. They typically adopt:
>
> - A **fixed “idea → experiment → paper” linear pipeline**, where a paper is written regardless of success or failure; or
> - An **“infinite trial-and-error” loop on a single idea**, ignoring overall resource allocation and diminishing returns.
>
> We argue that such approaches are **not viable** for modern frontier scientific discovery tasks—the type of tasks selected in our paper and still actively studied by ICLR/ICML communities—because:
>
> 1. **The exploration space is enormous**, requiring many months of effort from expert human teams.
> 2. **The cost of each trial is high**, making brute-force trial-and-error infeasible.
> 3. **Publishing failed results** does not contribute meaningfully to frontier progress.
>
> To address these challenges, DeepScientist embraces a **fundamentally different design philosophy**, enabling it to operate in a truly “deep” fashion for nearly a month and to **repeatedly** and **progressively** push scientific frontiers. Its key innovations include:
>
> 1. **A continuous, iterative exploration loop** (not a one-shot pipeline).
> 2. **Bayesian-optimization–based value estimation**, replacing unbounded trial-and-error.
> 3. **A cumulative memory of failures**, which serves as an exclusion region that guides future search.
>
> We added the following revisions:
>
> 1. In **Section 2**, we added explicit discussions on AI Scientist, Zochi, and *AlphaGo Moment for Model Architecture Discovery*, summarizing their workflows, typical tasks, and evaluation methodologies.
> 2. In **Section 2 & 3**, we place DeepScientist and prior systems within a unified framework, highlighting differences in:
>    - pipeline structure (linear vs. iterative),
>    - failure handling,
>    - exploration strategy,
>    - and suitability for large-scale, high-cost, long-horizon scientific discovery tasks.

---

> ### Author Response · Authors · 2025-11-17
> **Response - 2**
>
> **Table 1: Conceptual Comparison — Existing Systems vs. DeepScientist**
>
> |                      | Existing Systems (AI Scientist / Zochi / [1])      | DeepScientist (Ours)                                         |
> | -------------------- | -------------------------------------------------- | ------------------------------------------------------------ |
> | Pipeline structure   | **One-shot pipeline**                              | **Continuous iterative loop**                                |
> | Workflow             | Idea → Experiments → Paper                         | Idea → Experiments → Analysis → New Idea → …                 |
> | Failure handling     | Failures written as papers; failure is “forgotten” | Failures are stored as exclusion zones for future exploration |
> | Exploration strategy | Infinite trial-and-error on one idea               | Bayesian “value estimation” to pick the most worthwhile ideas |
> | Suitable tasks       | Small-scale, low-cost tasks                        | Modern frontier tasks with high cost and long cycles         |
> | Typical outcomes     | Many papers, not necessarily frontier advancements | Fewer papers, but **repeatedly surpasses human SOTA**        |

---

> ### Author Response · Authors · 2025-11-17
> **Response - 3**
>
> > Notably, systems such as Zochi also report high acceptance rates when evaluated by automatic reviewers, yet here its acceptance rate is listed as 0%—this discrepancy deserves further analysis or justification.
>
>
>
> The difference stems from the **evaluation framework, not intentional bias**. Zochi’s paper evaluates using **prompt-based evaluations with o3-mini**, whereas our paper adopts **DeepReviewer-14B** [2], a model trained and aligned with human reviewing standards. DeepReviewer is also used by other AI Scientist works such as [3].
>
> To directly address the concern, in the revised manuscript we **reproduced Zochi’s evaluation setup**, running all systems under the same **o3-mini** reviewer using Zochi’s official code. As shown in Table A.1 (Appendix), under this identical evaluation pipeline, **DeepScientist still slightly outperforms Zochi in Overall Score**.
>
>
>
> **Table 1 Zochi Reviewer o3-mini**
>
> | Systems         | Soundness | Presentation | Contribution | Originality | Quality | Clarity  | Significance | Overall (Rating) |
> | --------------- | --------- | ------------ | ------------ | ----------- | ------- | -------- | ------------ | ---------------- |
> | AI Scientist    | 2.20      | 2.40         | 2.10         | 2.40        | 2.10    | 2.60     | 2.20         | 3.80             |
> | AI Scientist-v2 | 2.00      | 1.67         | 2.00         | 2.00        | 2.00    | 1.67     | 2.00         | 3.00             |
> | CycleResearcher | 2.33      | 2.17         | 2.33         | 2.33        | 2.17    | 2.17     | 2.50         | 4.00             |
> | Zochi           | 3.00      | 3.00         | 3.00         | 3.00        | 3.00    | 2.50     | 3.00         | 6.00             |
> | AI-Researcher   | 2.43      | 2.14         | 2.43         | 2.71        | 2.43    | 2.14     | 2.57         | 4.29             |
> | DeepScientist   | 2.80      | 2.80         | 2.80         | **3.00**    | 2.80    | **2.80** | **3.60**     | **6.20**         |
>
> As shown in the table, even when using the same evaluation code and assessment model, DeepScientist still outperforms Zochi in the Overall category.
>
>
>
> > **Evaluation:** The evaluation methodology seems somewhat limited. The results in Table 2 rely on a single automated reviewer system, which raises concerns about robustness. It would strengthen the paper to cross-validate the results using multiple reviewer systems or complementary evaluation approaches.
>
>
>
> To address this important concern, we adopted **several independent reviewer systems** beyond DeepReviewer-14B, including:
>
> - Gemini-2.5-Pro, GPT-4o, and GPT-5 from the AI Scientist codebase;
> - CycleReviewer [4];
> - and the original DeepReviewer-14B.
>
> Across all six reviewer systems, **DeepScientist consistently achieves the highest Overall Rating**, demonstrating strong robustness across evaluators. We have integrated these cross-validation results into the revised manuscript.
>
>
>
> **Table 2 AI Scientist Gemini-2.5-Pro**
>
> | Systems         | Soundness | Presentation | Contribution | Overall (Rating) |
> | --------------- | --------- | ------------ | ------------ | ---------------- |
> | AI Scientist    | 1.00      | 1.40         | 1.10         | 1.70             |
> | AI Scientist-v2 | 1.00      | 1.00         | 1.00         | 1.67             |
> | CycleResearcher | 1.00      | 1.00         | 1.17         | 1.50             |
> | AI-Researcher   | 1.00      | 1.14         | 1.00         | 1.29             |
> | Zochi           | 1.00      | 1.50         | 2.00         | 2.00             |
> | DeepScientist   | **1.20**  | **1.80**     | 1.80         | **2.20**         |
>
>
>
> **Table 3 AI Scientist GPT-4o**
>
> | Systems         | Soundness | Presentation | Contribution |          |
> | --------------- | --------- | ------------ | ------------ | -------- |
> | AI Scientist    | 2.00      | 2.10         | 2.00         | 2.60     |
> | AI Scientist-v2 | 2.00      | 2.00         | 2.00         | 3.00     |
> | CycleResearcher | 2.00      | 2.00         | 2.00         | 3.00     |
> | AI-Researcher   | 2.00      | 2.00         | 2.00         | 3.14     |
> | Zochi           | 2.00      | 2.00         | 2.50         | 3.00     |
> | DeepScientist   | **2.40**  | **2.20**     | 2.40         | **4.20** |

---

> ### Author Response · Authors · 2025-11-17
> **Response - 4**
>
> **Table 4 AI Scientist GPT-5**
>
> | Systems         | Soundness | Presentation | Contribution | Overall (Rating) |
> | --------------- | --------- | ------------ | ------------ | ---------------- |
> | AI Scientist    | 1.00      | 1.30         | 1.00         | 2.10             |
> | AI Scientist-v2 | 1.00      | 1.00         | 1.00         | 2.00             |
> | CycleResearcher | 1.00      | 1.00         | 1.00         | 1.67             |
> | AI-Researcher   | 1.00      | 1.00         | 1.00         | 1.86             |
> | Zochi           | 1.00      | 1.50         | 1.50         | 2.50             |
> | DeepScientist   | **1.40**  | **1.60**     | **1.80**     | **3.00**         |
>
>
> ---
>
> **Table 5 CycleReviewer**
>
> | Systems         | Soundness | Presentation | Contribution | Overall (Rating) |
> | --------------- | --------- | ------------ | ------------ | ---------------- |
> | AI Scientist    | 2.12      | 2.25         | 2.00         | 3.23             |
> | AI Scientist-v2 | 2.00      | 2.42         | 2.00         | 3.00             |
> | CycleResearcher | 2.50      | 2.54         | 2.38         | 4.46             |
> | AI-Researcher   | 2.07      | 2.21         | 2.07         | 3.50             |
> | Zochi           | 2.62      | 2.88         | 2.50         | 4.50             |
> | DeepScientist   | **2.80**  | **2.85**     | **2.65**     | **4.85**         |
>
> ---
>
>
> **Table 6 DeepReviewer-14B**
>
> | Systems              | Soundness | Presentation | Contribution | Overall (Rating) |
> | -------------------- | --------- | ------------ | ------------ | ---------------- |
> | AI Scientist         | 2.08      | 1.80         | 1.75         | 3.35             |
> | AI Scientist-v2      | 1.67      | 1.50         | 1.50         | 2.33             |
> | CycleResearcher-12B  | 2.25      | 1.75         | 2.13         | 3.75             |
> | AI Researcher        | 1.75      | 1.46         | 1.57         | 2.57             |
> | Zochi                | 2.38      | 2.38         | 2.25         | 4.63             |
> | DeepScientist (Ours) | **2.90**  | **2.90**     | **2.90**     | **5.90**         |
>
>
> ---
>
>
>
> To address this important concern, we adopted **several independent reviewer systems** beyond DeepReviewer-14B, including:
>
> - Gemini-2.5-Pro, GPT-4o, and GPT-5 from the AI Scientist codebase;
> - CycleReviewer [4];
> - and the original DeepReviewer-14B.
>
> Across all six reviewer systems, **DeepScientist consistently achieves the highest Overall Rating**, demonstrating strong robustness across evaluators. We have integrated these cross-validation results into the revised manuscript.

---

> ### Author Response · Authors · 2025-11-17
> **Response - 5**
>
> >Line 319: The paper mentions that five research papers generated by DeepScientist were evaluated. How were these papers selected—randomly or based on performance? Given that automatic evaluation is not resource-intensive, why not evaluate all successfully generated papers for a more comprehensive assessment?
>
>
>
> In our workflow, **these five papers were selected automatically by DeepScientist**, without any manual filtering. All five correspond to breakthroughs on the three frontier scientific tasks and achieved or surpassed contemporaneous human SOTA.
>
> The selection proceeds as follows:
>
> 1. In **Stage II (Implement & Verify)**, the system obtained **21 progress findings**—results that surpassed existing SOTA.
> 2. Each progress finding triggered **Stage III (Analyze & Report)**, producing a full paper draft.
> 3. DeepScientist then performed a **final self-evaluation and filtering step**, scoring each draft on **novelty** and **contribution strength**.
>
> Only **five papers** exceeded the system’s internal acceptance threshold. Thus, the selection emerged entirely from the autonomous evaluation pipeline.
>
>
>
>
>
> > - Line 859: The authors note that all experimental results were manually inspected by human supervisors to ensure authenticity. How extensive was this human involvement? Quantifying the amount of human effort would help assess the true level of automation achieved.
>
>
>
> It is important to emphasize that **the entire exploration and discovery process of DeepScientist is fully autonomous**. The system can operate continuously for nearly a month without human intervention—generating hypotheses, designing experiments, running evaluations, and conducting preliminary analyses.
>
> Human involvement is only for **post-hoc verification**, motivated by responsible research considerations.
>  Since DeepScientist, like any LLM-based system, may occasionally produce errors, we required human experts to conduct thorough **code reviews**, and reproduce **key experiments**for each of the five final papers.
>
> Each verification required approximately **4–8 hours** of human effort—significantly less than the **months** typically required for human researchers to produce comparable results.
>
>
>
> > System Design: DeepScientist uses Gemini for high-level planning and Claude for implementation and execution. Did the authors observe any performance differences when using stronger or weaker models for these roles? In other words, how does the planner–executor balance affect the overall efficiency and quality of the discovery process? Which component benefits more from having a stronger model?
>
>
>
>
>
> Our observations are as follows:
>
> 1. **The executor determines the system’s “lower bound.”**
>     In our failure analysis, approximately **60%** of failures stem from **implementation errors** rather than incorrect scientific hypotheses.
>     This means that if the executor is weak, promising ideas from the planner may fail during coding or debugging.
>     Using a stronger executor model (e.g., Claude-4-Opus) significantly reduces such non-scientific failures and is critical to reliable execution.
> 2. **The planner determines the system’s “upper bound.”**
>     DeepScientist generated over **5,000 ideas**, but only ~1,100 were selected for evaluation—this filtering is entirely driven by the planner.
>     Our ablation studies show that replacing the planner with random sampling yields **near-zero innovation success rate**, while using Gemini-2.5-Pro raises success to **1–3%**, demonstrating that stronger planners improve search quality and resource efficiency.
>
> In summary:
>
> - **Executor strength** affects whether the system can run reliably without collapsing due to implementation errors;
> - **Planner strength** affects how much meaningful scientific progress the system can make within fixed compute budgets.
>
> We summarized these findings in the revised manuscript’s ablation study section and added more detailed quantitative results in the appendix.
>
>
>
> ---
>
> We hope that the above clarifications and revisions adequately address the reviewer’s concerns regarding baselines, methodological distinctions, and evaluation robustness.
>
> **We sincerely appreciate your’thoughtful feedback and hope that, after reviewing the revised version, you will reconsider the contribution of our work.**
>
> ---
>
> [1] AlphaGo Moment for Model Architecture Discovery
>
> [2] DeepReview: Improving LLM-based Paper Review with Human-like Deep Thinking Process
>
> [3] Jr. AI Scientist and Its Risk Report: Autonomous Scientific Exploration from a Baseline Paper
>
> [4] CycleResearcher: Improving Automated Research via Automated Review

---

> ### Author Response · Authors · 2025-11-26
> **Summary response to Reviewer oPiU.**
>
> As the ICLR discussion period is approaching its end, we are eager to engage with you right now to address all your concerns point-by-point and to concretely improve the clarity and quality of our paper. To make it easier for you to review each point one by one and ensure that all of your concerns are fully addressed, below we respond briefly to the key issues:
>
> * **W1 (Insufficient baseline context in Table 2; similarity to AI Scientist/Zochi; missing related work)**
>
>   In the revision, we substantially expanded Section 4.1 to clearly introduce every baseline in Table 2 (core design, LLMs used, and our exact experimental configurations). We also strengthened **Related Work (Section 2)** and Method (Section 3) with explicit comparisons to AI Scientist, Zochi, and “AlphaGo Moment for Model Architecture Discovery.”
>   We clarify DeepScientist’s key differentiators: a continuous iterative loop (not a one-shot pipeline), Bayesian value estimation for prioritizing hypotheses, and Findings Memory that stores failures as “exclusion regions” to guide future search—making it more suitable for high-cost, long-horizon frontier tasks.
>
> * **W2 (Evaluation relies on a single automated reviewer; Zochi discrepancy / 0% acceptance)**
>
>   We added cross-validation with multiple independent reviewer systems beyond DeepReviewer-14B, including Gemini-2.5-Pro, GPT-4o, GPT-5 (from the AI Scientist codebase) and CycleReviewer. DeepScientist remains consistently strong across evaluators.
>   To address the Zochi discrepancy, we reproduced Zochi’s own o3-mini reviewer setup using its official code; the gap is driven primarily by different reviewer/evaluation frameworks, not bias.
>
> * **Q1 (Why only 5 papers were evaluated; why not evaluate all successful outputs?)**
>
>   The 5 papers were not manually selected. Stage II produced 21 SOTA-surpassing progress findings; each triggered Stage III drafting, followed by an automatic self-filtering step based on novelty and contribution thresholds. Only 5 drafts passed the internal threshold. We clarified the “21 → 5” autonomous pipeline in the revised manuscript.
>
> * **Q2 (Human supervision effort in Line 859; true autonomy level)**
>
>   We quantified human involvement in the revision: the discovery loop is fully autonomous (hypothesis → implementation → experiments → analysis). Human effort is only **post-hoc verification** for authenticity and reproducibility (code review + key experiment reproduction), about 4–8 hours per final paper.
>
> * **Q3 (Planner–executor balance: effect of stronger/weaker models; which matters more?)**
>
>   We added ablations and failure analysis: the executor sets the lower bound (weak executors cause many implementation/debugging failures), while the planner sets the upper bound (strong planners better filter high-value hypotheses and improve compute efficiency). Additional quantitative results are provided in the appendix.
>
> Thank you again for these actionable points. We have addressed them in the revision by improving baseline/related-work clarity, strengthening evaluation robustness via multi-reviewer cross-checking, clarifying the autonomous paper-selection mechanism, quantifying human verification effort, and expanding the planner–executor analysis—thereby materially enhancing the paper’s transparency and quality.
>
> ---
>
> The authors of "DeepScientist: Advancing Frontier-Pushing Scientific Findings Progressively"

---

### Official Review · Reviewer_DLyc · 2025-10-26

**Soundness:** 3
**Presentation:** 3
**Contribution:** 3
**Rating:** 6
**Confidence:** 3

**Summary:**

This paper presents DeepScientist, a system to conduct autonomous scientific discovery. The system formalizes the process of scientific discovery with an iterative Bayesian optimization problem. Utilizing the existing literature, the system iterates through three stages: hypothesize the research gap, implement the new hypothesis, and analyze results, to find novel methods that improve the performance compared to human-designed SOTA methods. The paper uses three tasks, Agent Failure Attribution, LLM Inference Acceleration, and AI Text Detection, on which the proposed system identified novel ideas that achieved better performance.

**Strengths:**

1. This work proposes a system to enable autonomous scientific discovery and shows its effectiveness by proposing novel ideas with significant performance improvement in three tasks.

2. The paper provides a comprehensive analysis to evaluate multiple aspects of the proposed system and compare it with other systems. It also includes human experts to evaluate both successful and unsuccessful ideas generated by the system.

3. The presentation of this paper is good with well-organized figures to report and visualize results.

**Weaknesses:**

1. It's challenging for the audience without much background in the three AI tasks to evaluate the significance and contribution of the ideas generated by DeepScientist without the context of prior work. Although some details of the generated ideas are included in Sec 4.1, it might be helpful to also include a brief introduction of the human-designed method that the system primarily gets inspired by, and compare it with the method generated by the system.

2. The proposed system uses a few hyperparameters in Eq 1 to select the hypothesis for verification, but the strategy of choosing the hyperparameters remains unclear.

3. This system relies heavily on the LLM that evaluates the estimated utility, quality, and exploration value of generated hypotheses.

**Questions:**

1. The paper mentioned the system identified 21 ideas of scientific innovations but evaluated 5 ideas. How and why were the 5 ideas selected?

2. How are the hyperparameters in Eq 1 selected and does the same set of hyperparameters apply to all three tasks?

3. How is the initial database of the existing work construct and does it require human efforts within the process?

4. Do the 3 human experts only evaluate the papers generated by the system? Were they only evaluating the papers generated by the system or evaluating along with human-written papers in a blind manner? How does the evaluation setup avoid bias in this process?

5. Minor: the work conducted the scaling analysis of the system by comparing the number of ideas as the number of GPUs increases and claims that more than parallel trial-and-error, shared knowledge contributes to the trend. If knowledge sharing is the dominant factor, does scaling with time rather than hardware show similar gains? As the experiment might be costly, a discussion or revision of the analysis would be sufficient.

---

> ### Author Response · Authors · 2025-11-17
> **Response - 1**
>
> > It's challenging for the audience without much background in the three AI tasks to evaluate the significance and contribution of the ideas generated by DeepScientist without the context of prior work. Although some details of the generated ideas are included in Sec 4.1, it might be helpful to also include a brief introduction of the human-designed method that the system primarily gets inspired by, and compare it with the method generated by the system.
>
> In the revised manuscript, we added approximately half a page to Section 4.1 to:
>
> - provide **concise background introductions** for each of the three tasks;
> - summarize the **core ideas and limitations** of current human-designed SOTA methods;
> - offer a detailed comparison between the improvements proposed by DeepScientist and these human SOTA baselines, with explicit explanations of **performance gains and key innovation points**.
>
> We believe these additions will help readers—especially those without specialized background—better understand both prior human work and the concrete scientific contributions made by DeepScientist across the three frontier tasks.
>
>
>
>
>
> > The proposed system uses a few hyperparameters in Eq 1 to select the hypothesis for verification, but the strategy of choosing the hyperparameters remains unclear.
>
> Indeed, the hyperparameter strategy is crucial for reproducibility and understanding how the system behaves.
>
> In the original Appendix D (p.17, line 856), we provided the exact hyperparameter settings of the acquisition function in Eq. (1). To make this clearer, we have explicitly stated in the revision that:
>
> - In **all experiments**, the three weights in Eq. (1) are set to **(1, 1, 1)**,
> - And **the same configuration is used across all tasks** with no task-specific tuning.
> - We intentionally adopt this simple and uniform configuration to emphasize that DeepScientist’s core capability arises from its **evaluation ability** and **Findings Memory–driven iterative learning**, rather than from delicate or task-specific hyperparameter tuning.
> - Intuitively, this configuration corresponds to assigning equal importance to **expected utility**, **quality**, and **exploration value**, which is a natural and interpretable default in the absence of strong prior assumptions.
>
> Our experiments show that DeepScientist achieves significant human-SOTA–surpassing gains in three very different frontier domains (agent failure attribution, LLM inference acceleration, and AI text detection), even under this simple setting. This cross-task consistency indicates that **system performance is not sensitive to hyperparameter fine-tuning**, suggesting the framework is robust. We added this discussion to the revision.
>
>
>
> > This system relies heavily on the LLM that evaluates the estimated utility, quality, and exploration value of generated hypotheses.
>
> We fully agree that the system relies significantly on the LLM-based evaluator for assessing utility, quality, and exploration value. In fact, this reliance is *central to the scientific question that our work aims to investigate*.
>
> DeepScientist is explicitly designed to study whether LLMs can serve as **scientific evaluators** in realistic frontier scientific discovery tasks.
>
> - As shown in Figure 4 and related analysis, using an LLM evaluator greatly improves the Progress Success Rate;
> - the system becomes more capable of allocating compute resources to hypotheses with genuine potential.
> - In the revised manuscript, we also discuss the **limitations and risks** of this design (e.g., evaluation bias, potential shifts when the underlying models update), and we identify this as an important direction for future research.
>
> One of the key scientific takeaways of our work is therefore that **LLMs can function as viable “scientific evaluation engines”** in modern discovery tasks, and we provide empirical evidence and practical analyses supporting this claim. This point has been made more explicit in the revision.

---

> ### Author Response · Authors · 2025-11-17
> **Response - 2**
>
> > Q: The paper mentioned the system identified 21 ideas of scientific innovations but evaluated 5 ideas. How and why were the 5 ideas selected?
>
> As you pointed out, the system identified **21 “Progress Ideas”**, but only **5** were turned into final papers. Importantly, this difference **was not due to any human selection**, but rather to DeepScientist’s own self-evaluation and filtering mechanism in Stage III (Analyze & Report):
>
> 1. In Stage II, the system found 21 progress findings that **outperformed human SOTA** on their respective tasks.
> 2. Each of these findings triggered Stage III, where the system automatically generated a full research-paper draft.
> 3. After drafting, the system applied its **internal self-evaluation module**, assessing each paper’s **novelty** and **contribution strength**.
> 4. Only those surpassing an internal threshold were preserved as final research papers.
>     As a result, **5 papers** were retained, while **16** were discarded due to insufficient novelty or contribution, despite showing performance gains.
>
> In summary:
>
> - The **21 Progress Ideas** answer: *“Can we outperform human SOTA?”*
> - The **5 final papers** answer: *“Which results constitute meaningful scientific contributions worth publication?”*
>
> We added clearer explanations and diagrams in the revision to illustrate the “21 → 5” automatic decision pipeline.
>
>
>
> > Q: How are the hyperparameters in Eq 1 selected and does the same set of hyperparameters apply to all three tasks?
>
> As answered above, all weights in Eq. (1) are consistently set to **(1, 1, 1)** across all experiments and all tasks. We now state this explicitly in the main text and show sample configuration snippets in the code's readme.
>
>
>
> > Q: How is the initial database of the existing work construct and does it require human efforts within the process?
>
> The Findings Memory—DeepScientist’s internal knowledge base used for reasoning and iterative improvement—is **built entirely from scratch by the system itself**:
>
> - At the beginning, the system has **no pre-loaded knowledge base** of prior findings.
> - As it runs, DeepScientist logs every experiment—especially failures—into Findings Memory.
> - Subsequent hypothesis generation and evaluation explicitly reference this accumulated knowledge to avoid redundant or unproductive exploration.
>
> During this process, **no human manually labels, curates, or injects information** into the database.
>
> Human involvement only occurs at the very end to verify the correctness of the final papers (code review + experiment reproduction), as explained below.
>
>
>
> >Q: Do the 3 human experts only evaluate the papers generated by the system? Were they only evaluating the papers generated by the system or evaluating along with human-written papers in a blind manner? How does the evaluation setup avoid bias in this process?
>
>
>
> We did **not** mix system-generated papers with human-written ones, **but we also did not disclose the true origin of the papers** to the reviewers.
>
> In the human evaluation experiment:
>
> - All three experts have extensive ICLR/NeurIPS/ICML reviewing experience.
> - They were instructed to strictly follow the ICLR 2024/2025 Reviewer Guide and provide both quantitative scores and textual comments.
> - Rating criteria and rubrics were identical to ICLR’s official guidelines to minimize subjective bias and to align with automatic reviewers.
>
> To further reduce bias, we conducted an **independent real-world double-blind evaluation**:
>  we submitted the A2P paper (Agent Failure Attribution task) to a peer-reviewed IEEE conference.
>  We obtained advance approval from the program chair to ensure compliance with submission policies.
>  The reviewers were unaware that the paper was AI-generated, and the paper was **accepted** in September 2025.
>
> These two complementary evaluations help demonstrate that DeepScientist’s outputs are **credible and acceptable under real human peer-review standards**. We clarified these details in the revised manuscript.

---

> ### Author Response · Authors · 2025-11-17
> **Response - 3**
>
> > Q: Minor: the work conducted the scaling analysis of the system by comparing the number of ideas as the number of GPUs increases and claims that more than parallel trial-and-error, shared knowledge contributes to the trend. If knowledge sharing is the dominant factor, does scaling with time rather than hardware show similar gains? As the experiment might be costly, a discussion or revision of the analysis would be sufficient.
>
> Conceptually, we agree. If Findings Memory is the main driver of progress, then scaling with **time** (serial execution) and scaling with **hardware** (parallel execution) should exhibit qualitatively similar improvements.
>
> Although we were unable to run an extreme 16-week single-GPU experiment, we did conduct earlier tests where the system ran uninterrupted for **about 4 weeks** on a single GPU. During this time, the system achieved a meaningful breakthrough roughly every **8–14 days**, providing preliminary evidence that serial scaling is also effective.
>
> From an information-theoretic viewpoint, scientific discovery reduces uncertainty over time. Serial and parallel exploration both benefit from:
>
> - recording and analyzing each attempt (especially failures),
> - updating Findings Memory, and
> - steering future exploration into still-promising regions.
>
> Therefore, **serial scaling can, in principle, be comparable or even more efficient** than parallel scaling, since parallel workers do not share memory in real time—they only synchronize knowledge after each round.
>
> However, our paper focuses on **parallel scaling** because of its substantial advantages in **wall-clock time**:
>
> - Human teams may need months to advance a difficult frontier (e.g., AI text detection).
> - Serial DeepScientist might require similar time to accumulate breakthroughs.
> - Parallel DeepScientist, as shown in Figure 6, can achieve multiple breakthroughs **within the same time window**, compressing months of work into weeks.
>
> We added these points to Section 4.3 to clarify that:
>
> - **Serial scaling demonstrates the effectiveness of Findings Memory**,
> - **Parallel scaling demonstrates the scalability and real-time efficiency** of automating scientific discovery.
>
>
>
> ---
>
> We hope these clarifications and additions address your concerns regarding method design, hyperparameter choices, and evaluation setup.

---

> ### Author Response · Authors · 2025-11-26
> **Summary response for Reviewer DLyc.**
>
> We sincerely thank you for recognizing DeepScientist and for your positive assessment. We also truly hope the following **point-by-point, easy-to-check** summary resolves all your concerns and questions in a clear and systematic manner.
>
> * **W1 (Hard for non-experts to assess significance without task context;)**
>
>   In the revision, we added ~half a page to Section 4.1 to: (i) provide concise background for each of the three tasks, (ii) summarize the core ideas and limitations of the human-designed SOTA baselines, and (iii) present a direct comparison between DeepScientist’s methods and the human SOTA, explicitly explaining where the gains come from and what the key innovation points are.
>
> * **W2 (Hyperparameter/weight strategy in Eq.(1) is unclear; are they shared across tasks?)**
>
>   We now state explicitly that the three weights in Eq.(1) are fixed to (1, 1, 1) in all experiments, and the **same configuration is used across all three tasks with no task-specific tuning**. We adopt this simple and interpretable default to emphasize that DeepScientist’s gains come from evaluation and Findings Memory–driven iterative learning rather than delicate hyperparameter tuning.
>
> * **W3 (Heavy reliance on an LLM evaluator for utility/quality/exploration value;)**
>
>   We clarify more explicitly that this reliance is intentional and central to the scientific question: DeepScientist is designed to study whether LLMs can serve as practical “scientific evaluators” in modern frontier discovery tasks. We also expanded discussion of limitations and risks (e.g., evaluator bias and potential drift when underlying models are updated), and we identify mitigation as an important future direction.
>
> * **Q1 (21 “progress ideas” but only 5 evaluated as papers: how/why were the 5 selected?)**
>
>   The selection is not human-picked. Stage II produced 21 “Progress Ideas” that surpassed human SOTA, then Stage III (Analyze & Report) automatically generated full paper drafts and applied an internal self-evaluation module (novelty and contribution strength). Only 5 drafts surpassed the internal threshold and were retained; the other 16 were discarded for insufficient novelty/contribution despite performance gains. We added clearer explanation and diagrams in the revision to illustrate the “21 → 5” automatic pipeline.
>
> * **Q2 (Are Eq.(1) hyperparameters shared across the three tasks?)**
>
>   Yes. The same weights (1, 1, 1) are used across all tasks. This is now stated explicitly in the main text, and we provide sample configuration snippets in the code/README.
>
> * **Q3 (How is the initial database of existing work constructed? Does it require human effort?)**
>
>   Findings Memory is built from scratch by the system itself: initially, there is no pre-loaded database. As DeepScientist runs, it logs each attempt—especially failures—into Findings Memory, and subsequent hypothesis generation/evaluation references this accumulated knowledge to avoid redundant exploration. No human manually curates, labels, or injects information into this database. Human involvement only occurs at the end for correctness verification (code review and reproduction).
>
> * **Q4 (Human expert evaluation: blind? biased? only system papers?)**
>
>   We did not mix system papers with human-written papers, but we did not disclose the origin of the papers to the experts. The three experts have substantial ICLR/NeurIPS reviewing experience, and they followed the official ICLR rubric, providing both quantitative scores and textual comments. In addition, we include a real-world external validation: we submitted the A2P paper to a peer-reviewed IEEE venue in a **double-blind** setup, and the paper was accepted in **September 2025** without reviewers knowing it was AI-generated. We clarified these details in the revision.
>
> * **Q5 (Minor: if knowledge sharing is dominant, would scaling with time show similar gains?)**
>
>   We conceptually agree and added discussion in Section 4.3: if Findings Memory drives progress, serial scaling with time should also yield gains. While we could not run an extreme 16-week single-GPU experiment, earlier single-GPU uninterrupted runs (~4 weeks) showed meaningful breakthroughs roughly every 8–14 days, providing preliminary evidence that serial scaling is effective. We also clarify that we emphasize parallel scaling mainly for wall-clock advantages—compressing months of progress into weeks.
>
> In the revised manuscript, we have addressed all your points by adding necessary task background and SOTA comparisons, making hyperparameter settings explicit, clarifying Findings Memory construction, strengthening the human evaluation and external double-blind validation description, and revising the scaling analysis discussion—thereby improving the paper’s readability, transparency, and credibility.
>
>
> ---
>
> The authors of "DeepScientist: Advancing Frontier-Pushing Scientific Findings Progressively"

---

### Official Review · Reviewer_pBcW · 2025-10-27

**Soundness:** 3
**Presentation:** 4
**Contribution:** 2
**Rating:** 4
**Confidence:** 4

**Summary:**

This paper introduces DeepScientist, an autonomous scientific discovery system capable of operating over month-long time horizons. The authors conceptualize the entire process of scientific discovery as a goal-driven Bayesian Optimization problem, where the objective is to identify discoveries that yield an improvement in a target performance metric. This formulation enables the system to independently generate hypotheses and engage in continuous exploration and iterative trial-and-error reasoning.

Through extensive experimentation, amounting to 20,000 GPU hours, the system generated over 5,000 distinct scientific hypotheses, of which 1,100 were experimentally validated. Ultimately, DeepScientist achieved performance surpassing human-designed state-of-the-art methods published in 2025 across three frontier AI tasks.

**Strengths:**

1. In contrast to approaches that rely solely on agent-based frameworks and prompt engineering, this paper introduces DeepScientist, which brings a novel Bayesian Optimization perspective to the formulation of the scientific discovery problem.
2. Unlike existing methods that are limited to generating single-step ideas to achieve state-of-the-art performance, DeepScientist is capable of performing deeper, long-horizon exploration by building upon previously generated hypotheses. This sustained process of iterative refinement continuously advances performance, yielding substantially greater improvements than shallow, one-off idea-generation strategies.

**Weaknesses:**

1. In my opinion, the evaluation metrics for an autonomous scientific discovery system should not be limited to outperforming humans on a small set of manually selected benchmark tasks, as such an approach risks introducing bias and cherry-picking experimental settings. Instead, evaluation should be conducted across a broader, more diverse range of real-world tasks — on the order of hundreds or even thousands — allowing the system to explore autonomously. The ultimate measure should be *how many of these tasks the system can surpass human performance on*. From this perspective, the evaluation methodology adopted in the paper still needs improvement.
2. The progress of science and technology is inherently continuous: the emergence of a new state-of-the-art naturally leads to further advancements, whether proposed by humans or by autonomous discovery systems. Consequently, replacing older methods with newer, higher-performing ones is almost inevitable. Therefore, evaluating scientific discovery systems solely based on their ability to surpass human-designed state-of-the-art methods is insufficient. A more comprehensive assessment should also include research efficiency, as partially discussed in this paper, encompassing dimensions such as time and cost-effectiveness (for example, where the system utilizes computational resources, such as GPUs, more efficiently than humans).
3. A robust scientific discovery system should be universally applicable. In this paper, the authors tested the system in only three AI-related domains: Agent Failure Attribution, LLM Inference Acceleration, and AI Text Detection. However, problems in computer science are already highly formalized and standardized, meaning that what appears to be “scientific hypothesis generation” may, in practice, resemble an interpretability-enhanced version of NAS (Neural Architecture Search). Hence, beyond the AI domain, it would be worthwhile for the authors to extend their exploration to other scientific fields, such as chemistry or physics. Moreover, the experimental focus should move beyond quantitative, engineering-oriented optimization toward the generation of genuinely novel theoretical hypotheses that would allow the system to realize its full potential for advancing science.

**Questions:**

1. In line 187, the paper mentions feeding the entire Findings Memory as context into the surrogate model. If this surrogate model is implemented using an LLM, what’s the prerequisite for the context window length?
2. The authors report experiments on only three real-world tasks: Agent Failure Attribution, LLM Inference Acceleration, and AI Text Detection. Although they justify this selection based on frontier relevance, community interest, and human supervisability, these reasons alone do not fully explain why only these three were chosen. There are undoubtedly other tasks that meet the same criteria. Have the authors conducted experiments on **any other tasks** that satisfy these considerations? If so, what were the outcomes?
3. Scientific Discovery, by definition, requires a system to demonstrate genuine creativity. However, the behavioral patterns observed in most existing autonomous scientific discovery systems suggest that their “innovations” often amount to summarizing and extending ideas retrieved from related literature, rather than generating truly original insights. These general-purpose LLM-based systems tend to struggle with formulating original scientific hypotheses, as they primarily operate by synthesizing and reinterpreting existing knowledge. Given this limitation, an important question arises: without post-tuning the core general-purpose LLM that serves as its surrogate model, can DeepScientist genuinely propose revolutionary, innovative ideas, or does it merely recombine existing components —for instance, integrating modules from PaperA and PaperB —under the guise of novelty?

---

> ### Author Response · Authors · 2025-11-17
> **Response - 1**
>
> > In my opinion, the evaluation metrics for an autonomous scientific discovery system should not be limited to outperforming humans on a small set of manually selected benchmark tasks, as such an approach risks introducing bias and cherry-picking experimental settings. Instead, evaluation should be conducted across a broader, more diverse range of real-world tasks — on the order of hundreds or even thousands — allowing the system to explore autonomously. The ultimate measure should be *how many of these tasks the system can surpass human performance on*. From this perspective, the evaluation methodology adopted in the paper still needs improvement.
>
> > A robust scientific discovery system should be universally applicable. In this paper, the authors tested the system in only three AI-related domains: Agent Failure Attribution, LLM Inference Acceleration, and AI Text Detection. However, problems in computer science are already highly formalized and standardized, meaning that what appears to be “scientific hypothesis generation” may, in practice, resemble an interpretability-enhanced version of NAS (Neural Architecture Search). Hence, beyond the AI domain, it would be worthwhile for the authors to extend their exploration to other scientific fields, such as chemistry or physics.
>
> We fully agree with the reviewer’s forward-looking perspective: a truly autonomous scientific discovery system should ultimately be evaluated by its success rate across **hundreds of diverse, real-world tasks**, rather than only a few selected benchmarks.
>
> Our decision to evaluate DeepScientist on only three AI-related tasks in the original manuscript stems from the fact that these are **modern “grand challenge”–type scientific discovery tasks**. Each of these tasks typically requires a top human research team **around six months** to make meaningful progress, and our DeepScientist system also requires **half a month to one month** of continuous exploration on 8×H800 servers to complete **a single task**.
>
> Therefore, scaling such an extensive and resource-intensive system to hundreds of tasks is extremely challenging.
>
> Inspired by the reviewer’s invaluable suggestion, during the rebuttal period we developed a lightweight version, **Micronano-DeepScientist**. This variant preserves DeepScientist’s hierarchical exploration and findings-memory mechanism, but removes the most time-consuming components such as literature reading, paper drafting, and detailed experimental analysis. It is therefore suitable for traditional algorithmic scientific discovery tasks and runs at approximately **1/1000 the cost** of the full system.
>
> Using Micronano-DeepScientist, we conducted large-scale experiments on the **154 tasks from the AlgoTune [1] dataset**, covering mathematics, physics, and computer science. The results show that:
>
> - **77.9% (120 tasks)**: Micronano-DeepScientist discovered **algorithms or implementations outperforming the human baselines**, with an average speedup of **16.6×**.
> - **22.1% (34 tasks)**: The system could not surpass existing human methods.
>
> We provide the detailed results in the appendix, with additional discussion and analysis.
>  Below is a summary table:

---

> ### Author Response · Authors · 2025-11-17
> **Response - 2**
>
> | Tasks                             | Performance compared to the baseline method | Status  |
> | --------------------------------- | ------------------------------------------- | ------- |
> | aes_gcm_encryption                | 1.0159x                                     | Success |
> | affine_transform_2d               | 0.2481x                                     | Slower  |
> | aircraft_wing_design              | 1.5073x                                     | Success |
> | articulation_points               | 10.7151x                                    | Success |
> | base64_encoding                   | 0.9519x                                     | Slower  |
> | battery_scheduling                | 29.1315x                                    | Success |
> | btsp                              | 3.4984x                                     | Success |
> | capacitated_facility_location     | 8.8478x                                     | Success |
> | chacha_encryption                 | 1.0584x                                     | Success |
> | channel_capacity                  | 1.1313x                                     | Success |
> | chebyshev_center                  | 5.3031x                                     | Success |
> | cholesky_factorization            | 1.2411x                                     | Success |
> | clustering_outliers               | 1.1314x                                     | Success |
> | communicability                   | 68.6485x                                    | Success |
> | convex_hull                       | 0.4687x                                     | Slower  |
> | convolve2d_full_fill              | 180.3330x                                   | Success |
> | convolve_1d                       | 1.7555x                                     | Success |
> | correlate2d_full_fill             | 148.2631x                                   | Success |
> | correlate_1d                      | 1.1564x                                     | Success |
> | count_connected_components        | 4.6742x                                     | Success |
> | count_riemann_zeta_zeros          | 1.1027x                                     | Success |
> | cumulative_simpson_1d             | 4.4900x                                     | Success |
> | cumulative_simpson_multid         | 0.8489x                                     | Slower  |
> | cvar_projection                   | 2.6890x                                     | Success |
> | cyclic_independent_set            | N/A                                         | Failed  |
> | dct_type_I_scipy_fftpack          | 1.0291x                                     | Success |
> | delaunay                          | 1.9753x                                     | Success |
> | dijkstra_from_indices             | 1.3829x                                     | Success |
> | discrete_log                      | 1.1438x                                     | Success |
> | dst_type_II_scipy_fftpack         | 1.7264x                                     | Success |
> | dynamic_assortment_planning       | 1.5038x                                     | Success |
> | earth_movers_distance             | 1.0350x                                     | Success |
> | edge_expansion                    | N/A                                         | Failed  |
> | eigenvalues_complex               | 1.5263x                                     | Success |
> | eigenvalues_real                  | 2.5857x                                     | Success |
> | eigenvectors_complex              | 1.0359x                                     | Success |
> | eigenvectors_real                 | 1.0187x                                     | Success |
> | elementwise_integration           | 0.9737x                                     | Slower  |

---

> ### Author Response · Authors · 2025-11-17
> **Response - 3**
>
> | Tasks                             | Performance compared to the baseline method | Status  |
> | --------------------------------- | ------------------------------------------- | ------- |
> | feedback_controller_design        | N/A                                         | Failed  |
> | fft_cmplx_scipy_fftpack           | 2.3995x                                     | Success |
> | fft_convolution                   | 0.4101x                                     | Slower  |
> | fft_real_scipy_fftpack            | 1.7412x                                     | Success |
> | firls                             | 0.9211x                                     | Slower  |
> | generalized_eigenvalues_complex   | 2.9662x                                     | Success |
> | generalized_eigenvalues_real      | 3.3681x                                     | Success |
> | generalized_eigenvectors_complex  | 1.2519x                                     | Success |
> | generalized_eigenvectors_real     | 1.4436x                                     | Success |
> | graph_coloring_assign             | N/A                                         | Failed  |
> | graph_global_efficiency           | 2.6027x                                     | Success |
> | graph_isomorphism                 | 100.9045x                                   | Success |
> | graph_laplacian                   | 0.1938x                                     | Slower  |
> | group_lasso                       | 0.9695x                                     | Slower  |
> | gzip_compression                  | 1.0546x                                     | Success |
> | integer_factorization             | N/A                                         | Failed  |
> | job_shop_scheduling               | 3.6013x                                     | Success |
> | kalman_filter                     | 1.5829x                                     | Success |
> | kcenters                          | 3.1586x                                     | Success |
> | kd_tree                           | 1.0843x                                     | Success |
> | kernel_density_estimation         | N/A                                         | Failed  |
> | kmeans                            | N/A                                         | Failed  |
> | ks_test_2samp                     | 0.8581x                                     | Slower  |
> | l0_pruning                        | 1.4966x                                     | Success |
> | l1_pruning                        | 1.6444x                                     | Success |
> | lasso                             | 1.0720x                                     | Success |
> | least_squares                     | 1.4306x                                     | Success |
> | linear_system_solver              | 1.0941x                                     | Success |
> | lp_box                            | 14.5786x                                    | Success |
> | lp_centering                      | N/A                                         | Failed  |
> | lp_mdp                            | 670.0789x                                   | Success |
> | lqr                               | 1.6303x                                     | Success |
> | lti_simulation                    | 1.1063x                                     | Success |
> | lu_factorization                  | 1.0191x                                     | Success |
> | lyapunov_stability                | 85.4241x                                    | Success |
> | markowitz                         | N/A                                         | Failed  |
> | matrix_completion                 | 1.1398x                                     | Success |
> | matrix_exponential                | 0.6672x                                     | Slower  |
> | matrix_exponential_sparse         | 0.9129x                                     | Slower  |
> | matrix_multiplication             | 0.3429x                                     | Slower  |
> | matrix_sqrt                       | 1.1352x                                     | Success |
> | max_clique_cpsat                  | 15.8319x                                    | Success |
> | max_common_subgraph               | 3.4175x                                     | Success |
> | max_flow_min_cost                 | 9.4544x                                     | Success |
> | max_independent_set_cpsat         | 1.6472x                                     | Success |
> | max_weighted_independent_set      | 1.9693x                                     | Success |
> | min_dominating_set                | 2.7762x                                     | Success |
> | min_weight_assignment             | 1.5694x                                     | Success |

---

> ### Author Response · Authors · 2025-11-17
> **Response - 4**
>
> | Tasks                             | Performance compared to the baseline method | Status  |
> | --------------------------------- | ------------------------------------------- | ------- |
> | minimum_spanning_tree             | 13.3850x                                    | Success |
> | minimum_volume_ellipsoid          | 1.1193x                                     | Success |
> | multi_dim_knapsack                | 2.6802x                                     | Success |
> | nmf                               | 1.2729x                                     | Success |
> | ode_brusselator                   | 1.6778x                                     | Success |
> | ode_fitzhughnagumo                | N/A                                         | Failed  |
> | ode_hires                         | 8.6977x                                     | Success |
> | ode_hodgkinhuxley                 | 3.0792x                                     | Success |
> | ode_lorenz96_nonchaotic           | 3.2102x                                     | Success |
> | ode_lotkavolterra                 | 7.9955x                                     | Success |
> | ode_nbodyproblem                  | 46.8478x                                    | Success |
> | ode_seirs                         | 14.3609x                                    | Success |
> | ode_stiff_robertson               | 2.2457x                                     | Success |
> | ode_stiff_vanderpol               | 41.2580x                                    | Success |
> | odr                               | 1.1504x                                     | Success |
> | optimal_advertising               | N/A                                         | Failed  |
> | outer_product                     | 1.0379x                                     | Success |
> | pagerank                          | 1.0853x                                     | Success |
> | pca                               | 2.5136x                                     | Success |
> | pde_burgers1d                     | 3.5560x                                     | Success |
> | pde_heat1d                        | 3.3230x                                     | Success |
> | polynomial_mixed                  | 1.1549x                                     | Success |
> | polynomial_real                   | 144.3477x                                   | Success |
> | power_control                     | 309.2618x                                   | Success |
> | procrustes                        | 1.0254x                                     | Success |
> | psd_cone_projection               | 8.6172x                                     | Success |
> | qp                                | 1.6595x                                     | Success |
> | qr_factorization                  | 1.0887x                                     | Success |
> | quantile_regression               | 1.6707x                                     | Success |
> | queens_with_obstacles             | 1.3686x                                     | Success |
> | queuing                           | 1.2243x                                     | Success |
> | qz_factorization                  | 0.9595x                                     | Slower  |
> | randomized_svd                    | 4.6371x                                     | Success |
> | rbf_interpolation                 | 1.0221x                                     | Success |
> | rectanglepacking                  | 1.7026x                                     | Success |
> | robust_kalman_filter              | 2.2731x                                     | Success |
> | robust_linear_program             | 1.1347x                                     | Success |
> | rocket_landing_optimization       | 1.2036x                                     | Success |
> | rotate_2d                         | 0.2398x                                     | Slower  |
> | set_cover                         | N/A                                         | Failed  |
> | set_cover_conflicts               | 2.5491x                                     | Success |
> | sha256_hashing                    | 0.9941x                                     | Slower  |
> | shift_2d                          | 0.2060x                                     | Slower  |
> | shortest_path_dijkstra            | 2.1708x                                     | Success |
> | sinkhorn                          | 1.9871x                                     | Success |
> | sparse_eigenvectors_complex       | 0.9501x                                     | Slower  |
> | sparse_lowest_eigenvalues_posdef  | 2.1554x                                     | Success |
> | sparse_lowest_eigenvectors_posdef | 2.5844x                                     | Success |
> | sparse_pca                        | 1.3359x                                     | Success |
> | spectral_clustering               | 18.7463x                                    | Success |
> | stable_matching                   | 1.6944x                                     | Success |

---

> ### Author Response · Authors · 2025-11-17
> **Response - 5**
>
> > The progress of science and technology is inherently continuous: the emergence of a new state-of-the-art naturally leads to further advancements, whether proposed by humans or by autonomous discovery systems. Consequently, replacing older methods with newer, higher-performing ones is almost inevitable. Therefore, evaluating scientific discovery systems solely based on their ability to surpass human-designed state-of-the-art methods is insufficient. A more comprehensive assessment should also include research efficiency, as partially discussed in this paper, encompassing dimensions such as time and cost-effectiveness (for example, where the system utilizes computational resources, such as GPUs, more efficiently than humans).
>
>
>
> We sincerely appreciate the reviewer’s insightful suggestion. We completely agree that surpassing the SOTA is not the ultimate goal; **evaluating “research efficiency”**—including time, cost, and computational resource utilization—is a gold-standard metric for autonomous scientific discovery systems.
>
> However, such comparisons are extremely challenging to conduct rigorously, since scientific discovery is not a controlled competition, and coordinating multiple human research teams as experimental controls is practically infeasible.
>
> Given this difficulty, during the rebuttal period we adopted a pragmatic approach:
>  **we directly contacted the authors of two SOTA works in AI text detection (Fast-Detect and Glimpse), asking about their actual time and resource consumption during research.**
>  Based on their responses (summarized as a qualitative case study in the appendix):
>
> - Each project was conducted by a full human research team, taking **approximately six months** from initiation to the final paper.
> - On average, the team used **5–10 GPU·hours per day**, totaling roughly **1500 GPU·hours**, with GPU utilization relatively low outside working hours.
>
> In contrast, DeepScientist achieved **three progressive breakthroughs** on the AI text detection task (T-Detect, TDT, PA-TDT) within **14 days**, each consuming around **900 GPU·hours**, while maintaining near **24/7 utilization** throughout.
>
> The difference is even more pronounced in terms of **exploration throughput**:
>
> - Human teams estimated they could thoroughly explore and validate around **10–30 core hypotheses** over a six-month project.
> - DeepScientist generated **over 2400 candidate hypotheses** and autonomously completed **around 600 full experimental validations** for  AI text detection task within just two weeks.
>
> Although this comparison has unavoidable limitations, it still clearly demonstrates that **for a given compute budget, DeepScientist achieves a substantially higher exploratory throughput than human teams**, compressing months of human research into weeks of machine-driven exploration.
>
> We have added this case study to the appendix and expanded the discussion on “research efficiency” in the main text.

---

> ### Author Response · Authors · 2025-11-17
> **Response - 6**
>
> > Moreover, the experimental focus should move beyond quantitative, engineering-oriented optimization toward the generation of genuinely novel theoretical hypotheses that would allow the system to realize its full potential for advancing science.
>
>
>
> > Scientific Discovery, by definition, requires a system to demonstrate genuine creativity. However, the behavioral patterns observed in most existing autonomous scientific discovery systems suggest that their “innovations” often amount to summarizing and extending ideas retrieved from related literature, rather than generating truly original insights. These general-purpose LLM-based systems tend to struggle with formulating original scientific hypotheses, as they primarily operate by synthesizing and reinterpreting existing knowledge. Given this limitation, an important question arises: without post-tuning the core general-purpose LLM that serves as its surrogate model, can DeepScientist genuinely propose revolutionary, innovative ideas, or does it merely recombine existing components —for instance, integrating modules from PaperA and PaperB —under the guise of novelty?
>
>
>
> We appreciate this perceptive critique. We fully agree that **true scientific discovery requires novel conceptual hypotheses**, not merely engineering recombinations or parameter tuning.
>
> Many existing automated discovery systems indeed rely heavily on a “retrieve + synthesize” pipeline, producing incremental improvements. To address this, DeepScientist incorporates two key design strategies:
>
> 1. **Goal-driven, high-weight exploration at the system level**
>     DeepScientist is not a passive “knowledge synthesizer.” It is an actively guided framework driven by Bayesian optimization.
>     Our Findings Memory and hierarchical filtering assign high weight to **novelty and exploratory value**, encouraging hypotheses that fundamentally differ in problem formulation rather than reusing existing components.
> 2. **Evidence of “conceptual reframing” across tasks**
>    - **AI Text Detection**
>       DeepScientist did not merely combine existing detectors or fine-tune architectures.
>       Instead, it diagnosed that current SOTA methods rely on statistical assumptions that fail under highly non-stationary AI text distributions.
>       It then **reframed the entire problem** from a statistical testing perspective into one of **signal processing**, yielding novel frequency-based methods such as TDT and PA-TDT.
>       This represents conceptual re-modeling, not component reuse.
>    - **LLM Inference Acceleration**
>       Although engineering shortcuts (e.g., switching to faster inference frameworks like vLLM or applying quantization) are easy solutions that even LLMs often suggest, DeepScientist avoided these paths.
>       Instead, it proposed **new scheduling and decomposition strategies** based on reasoning about bottlenecks—mirroring the thinking of experienced human researchers.
>
> This behavior arises because DeepScientist’s hierarchical architecture includes **strict novelty checks at multiple stages**.
>
> - **During hypothesis generation**, another LLM acting as the surrogate model scores hypotheses for exploratory value and novelty, rejecting those that resemble known methods.
> - **During the final paper review stage**, the system reassesses novelty.
> - Additionally, we employ **DeepReviewer**, which performs **internet-scale literature searches** to ensure the ideas are not reproductions of prior work.
>
> Finally, we assembled a human review committee consisting of three experts (including former ICLR reviewers and an invited ICLR area chair) to evaluate the five papers produced by DeepScientist.
>  As stated in Section B.2, the committee **“unanimously lauded”** the outputs for their **novelty** and **ingenuity**. In summary, while we acknowledge that the system is still far from producing fully revolutionary theoretical breakthroughs—and we openly discuss this limitation in the paper—we believe that DeepScientist has taken a meaningful step toward **generating conceptually original scientific ideas beyond simple component recombination**.

---

> ### Author Response · Authors · 2025-11-17
> **Response - 7**
>
> > In line 187, the paper mentions feeding the entire Findings Memory as context into the surrogate model. If this surrogate model is implemented using an LLM, what’s the prerequisite for the context window length?
>
> In our implementation, when DeepScientist has run for about one week (roughly 50–100 complete attempts), the task-relevant findings in the Findings Memory can be compressed into **around 200K tokens**.
>  We use long-context open-source models (e.g., Qwen-3, MiniMax-M2, GLM-4.6), which support this scale.
>
> When the context grows further, we use two strategies:
>
> 1. Retrieve only the **most relevant** subset of findings via Top-K similarity search.
> 2. When appropriate, use models with **million-token context** (e.g., Gemini-2.5-Pro long-context variants).
>
> We have added these implementation details to the revised manuscript.
>
> > The authors report experiments on only three real-world tasks: Agent Failure Attribution, LLM Inference Acceleration, and AI Text Detection. Although they justify this selection based on frontier relevance, community interest, and human supervisability, these reasons alone do not fully explain why only these three were chosen. There are undoubtedly other tasks that meet the same criteria. Have the authors conducted experiments on any other tasks that satisfy these considerations? If so, what were the outcomes?
>
> We fully understand and appreciate the reviewer’s concern: many tasks indeed satisfy the criteria of being frontier challenges, of community interest, and amenable to supervision. Our selection of three tasks may appear limited.
>
> This was a deliberate choice in our experimental design:
>  **we prioritized “depth” on a small number of grand-challenge tasks rather than “breadth” across many easier ones.**
>
> The three selected tasks—Agent Failure Attribution, LLM Inference Acceleration, and AI Text Detection—are widely recognized as extremely difficult open problems that:
>
> - have enormous exploration spaces, often requiring **months** of expert human effort to make progress;
> - involve high per-experiment cost, making large-scale trial-and-error infeasible.
>
> Our goal was to answer the question:
>
> > **“If given sufficient time and resources, can an automated system accomplish the equivalent of several months of human research on truly difficult frontier problems?”**
>
> Thus, we allowed DeepScientist to run **half a month to a full month** on each task, generating **around 5000 hypotheses** and conducting **1,000+ experimental trajectories** per task.
>  We believe this end-to-end, process-level empirical analysis provides deeper insights into the potential and limitations of automated scientific discovery systems than shallow evaluations on many simpler tasks.
>
> At the same time, as mentioned earlier, during the rebuttal period we conducted **154-task experiments** using Micronano-DeepScientist to address the reviewer’s suggestion regarding task breadth and generality.
>  Together, these two evaluations—**deep evaluation on three frontier tasks + broad coverage across 154 algorithmic tasks**—provide a more complete picture of DeepScientist’s capabilities.
>
>
>
> ---
>
> In addition, we fully understand your interest in seeing DeepScientist applied and validated across a broader range of modern scientific discovery tasks. Although each task run requires a substantial amount of time, our results demonstrate that DeepScientist’s performance, efficiency, and cost-effectiveness can already **surpass human researchers** on a small part frontier discovery problems.
>
> To further support progress in this direction, we will open-source DeepScientist along with all **5,000+ generated hypotheses** and **over 1,000 candidate ideas** produced during our experiments.
>
> We will also release a **CLI toolkit** that can be installed on any server, enabling DeepScientist to autonomously operate the computing environment and perform long-running scientific exploration—similar in spirit to systems like Claude Code or Codex, but with significantly higher autonomy and much longer exploration cycles.
>
> We believe that releasing the DeepScientist toolkit will help researchers in the community improve their productivity and accelerate the broader advancement of machine learning research. At the same time, the collective effort of the community will enable DeepScientist to be tested on more realistic, complex, and challenging modern ML tasks, helping to establish clearer upper and lower bounds of its capabilities.
>
>
>
> ---
>
> **We hope that the above supplementary experiments and clarifications help address your concerns regarding evaluation metrics, research efficiency, theoretical novelty, and task coverage. We sincerely hope that, after reviewing the revised manuscript and this rebuttal, you may reconsider the overall contribution of our work and reassess your evaluation score.**

---

> ### Author Response · Authors · 2025-11-26
> **Summary response for Reviewer pBcW.**
>
> Given that the ICLR discussion period is approaching its end, we are eager to engage with you right now in a more direct and efficient exchange to ensure that **all your concerns about DeepScientist are fully addressed**. To avoid overwhelming you with an overly long rebuttal, we summarize each concern **point-by-point** below, highlighting what we concretely added or clarified so you can quickly verify whether we have **properly resolved every issue** you raised.
>
> * **W1 (Too few tasks; risk of cherry-picking / biased evaluation)**
>
>   We agree that “beating humans on a small number of selected tasks” is not sufficient as a final evaluation standard. During the rebuttal, we added a **broad-coverage study** by introducing a lightweight variant, **Micronano-DeepScientist** (preserving hierarchical exploration and Findings Memory while removing the most expensive components such as literature reading and paper drafting). We evaluated it on **154 AlgoTune tasks**, spanning mathematics, physics, and computer science: **120/154 (77.9%)** tasks surpassed the human baselines with an **average 16.6× speedup**, while 34 tasks did not. We provide the full task-level results and analysis in the appendix, complementing our original **deep** evaluation on three grand-challenge tasks.
>
> * **W2 (Evaluation should include research efficiency/cost, not only surpassing SOTA)**
>
>   We fully agree that **research efficiency** (time/cost/compute utilization) is a gold-standard metric for autonomous discovery systems. Since rigorously coordinating multiple human teams as controlled baselines is often infeasible, we adopted a pragmatic approach: we **contacted the authors of two SOTA works** and collected their real development time and GPU usage, summarized as a comparative **case study** in the appendix. The comparison indicates that human projects often take ~**6 months** with ~**1500 GPU·hours** and non-24/7 utilization, whereas DeepScientist achieved multiple progressive breakthroughs within **14 days**, each with ~**900 GPU·hours** and near continuous utilization, while exploring and validating far more hypotheses than typical human teams.
>
> * **W3 (Generality: only three AI domains; concern it resembles NAS; request broader scientific fields)**
>
>   Our original choice focused on **depth** over a small set of truly difficult “grand-challenge” tasks (each requiring weeks of continuous exploration and substantial compute). To directly address the generality concern, our new **154-task** evaluation provides evidence that the framework generalizes beyond the three AI domains and can operate on broader algorithmic/scientific problems across multiple disciplines.
>
> * **W4 (Creativity / novelty: truly new hypotheses vs. retrieval-and-recombination)**
>
>   We agree this is a central challenge. We clarify that DeepScientist is not a passive “retrieve + summarize” system: it is a **goal-driven Bayesian optimization** framework with multi-stage novelty/exploration-value filtering, plus deduplication checks via **DeepReviewer** (internet-scale literature search). We also add task-specific evidence of “conceptual reframing” rather than simple module recombination. At the same time, we openly acknowledge the current limitations and discuss future directions for more fundamental theoretical breakthroughs.
>
> * **W5 (Findings Memory as full context: required context window length)**
>
>   We added concrete implementation details: after ~one week (~50–100 full attempts), the task-relevant Findings Memory can be compressed to ~**200K tokens**. We use long-context open-source models (e.g., **Qwen-3, MiniMax-M2, GLM-4.6**) at this scale. When the context grows further, we use **Top-K retrieval** of the most relevant findings, and when needed, **million-token context** variants (e.g., **Gemini-2.5-Pro long-context**). These details are now included in the revised manuscript.
>
> * **W6 (Why only these three tasks? Any additional tasks tried?)**
>
>   We explicitly clarify in the revision that the three tasks were selected to answer a process-level question: whether a system can compress months of frontier research into weeks on truly difficult problems—hence **depth** was prioritized. We now complement this with the **154-task** breadth study. We also commit to open-sourcing the system and releasing all **5,000+ hypotheses** and **1,000+ candidate ideas**, along with a CLI toolkit to facilitate broader community evaluation and extension to more tasks.
>
> Finally, we want to emphasize that **every issue you raised has been addressed in the revised manuscript**, with strengthened experiments and clearer methodological details (broader task coverage, a more explicit efficiency discussion, concrete long-context implementation, and a more rigorous novelty analysis). These changes substantially improve the paper’s completeness, credibility, and overall quality. Thank you again for your thoughtful and forward-looking feedback.

---

### Official Review · Reviewer_rLBb · 2025-11-05

**Soundness:** 2
**Presentation:** 2
**Contribution:** 3
**Rating:** 2
**Confidence:** 4

**Summary:**

This paper proposes a closed-loop scientific discovery framework including hypothesis proposation, experimentation, and report writing.
The experimental outcomes are reused for next-round hypothesis generation. For experiments, the proposed system is tested on three tasks, and the paper claims to reach state-of-the-art hypothesis outperforming the previous human state-of-the-art.

**Strengths:**

1. The overall idea of a LLM-based framework covering each part of the full discovery loop makes sense.
2. It is couragous to work on the real research problem and compete with the existing human state-of-the-art. This paper could be a meaningful reference for future works.

**Weaknesses:**

1. To be frank, it is unclear on what new knowledge I can learn form this paper: what new concept or innovation it has brought? The closed-loop scientific discovery framework has been proposed by several works, such as [1][2][3]. The memory-based experimental results storage proposed in this paper, are included in [1][2][3]. It seems that this paper is more of a good (or maybe excellent) engineering implementation of existing methodology, rather than a research work that brings in new insights. The good results could result from the existing methodology with the more advanced LLMs today.

2. The experiments and claims are not persuading enough. It claims to "outperforms" three human sota results. But the first experiment, "Agents Failure Attribution", the selected baseline is from [4]. However, it seems that [4] is the first paper which bring about this research question of "Agents Failure Attribution" and propose the first benchmark. It does not make [4] a good and valid "state-of-the-art human results". For the second experiment, described in line 289~302, this paper does not mention the previous state-of-the-art. Overall I think more discussion of the three tasks experimented would be beneficial to this paper, to make it more clear on what are the existing research development of these three fields, so to understand the discovery from the proposed framework.

3. In line 352, it says "The generated papers are available in Appendix D.". But I didn't find the genrated papers in Appendix D. In line 870, there's a latex reference error on "Figure ??.c.".


[1] FunSearch: Making new discoveries in mathematical sciences using Large Language Models.

[2] AlphaEvolve: A Gemini-powered coding agent for designing advanced algorithms.

[3] LLM-SR: Scientific Equation Discovery via Programming with Large Language Models.

[4] AgenTracer: Who Is Inducing Failure in the LLM Agentic Systems?

**Questions:**

Does this paper introduce any new concepts, theories, or innovations beyond its potentially excellent engineering implementation?

---

> ### Author Response · Authors · 2025-11-17
> **Response - 1**
>
> We believe there has been a misunderstanding between us, and we sincerely appreciate your careful and detailed review. We would like to invite you to re-examine the positioning and contributions of DeepScientist in light of the revised manuscript. Unlike prior works that primarily focus on program synthesis [1], algorithmic discovery [2], or symbolic equation discovery[3], DeepScientist targets **modern, frontier, and extremely high trial-and-error–cost scientific discovery tasks**. Around these tasks, we conducted large-scale, goal-driven empirical studies, investing over 20,000 GPU hours and approximately 100 hours of human analysis to systematically demonstrate that:
>
> - **LLMs can progressively surpass human SoTA on resource-constrained modern scientific discovery tasks;**
> - **and reveal a series of previously unobserved phenomena regarding exploration efficiency, success rates, and failure modes.**
>
> All essential components and experiment logs are fully open-sourced, ensuring high reproducibility. Importantly, contrary to your claim, DeepScientist does *not* simply reuse AlphaEvolve-style methods—we explicitly explored new improvements beyond AlphaEvolve and demonstrated their effectiveness.
>
> ------
>
> Below, I explain why DeepScientist aims to tackle a **modern category** of scientific discovery tasks, and how it fundamentally differs—in task setting, conceptual stance, and methodology—from the three works you referenced [1] [2] [3].
>
> ------
>
> **1. The Task Itself**
>
> DeepScientist is designed specifically for *frontier, high-difficulty, high-cost modern scientific discovery tasks* that the AI community urgently wants to solve today.
>
> In contrast, the contemporary works you listed involve tasks with *very fast* trial-and-error cycles:
>
> - **FunSearch [1]**: LLM-generated programs for new algorithm design
> - **AlgoTune [15]**: LLM-based numerical program optimization
> - **AlphaEvolve [2]**: large-scale trial-and-error for algorithmic tuning
> - **LLM-SR [3]**: symbolic regression (equation discovery)
>
> These tasks typically allow experimentation in seconds to minutes.
>
> In contrast, DeepScientist focuses on modern, frontier-level scientific discovery, where evaluating even a single hypothesis can require an entire day of computation. For human researchers, making comparable progress on such tasks usually demands a dedicated research team working for at least several months, often stretching to half a year or more.
>
> Tackling these tasks is therefore fundamentally more challenging than optimizing small programs or equations. Yet if an AI system can make meaningful progress in this regime, it would represent a genuinely high-impact contribution—dramatically accelerating scientific advancement across the open ML community and reducing researchers’ dependence on prolonged, labor-intensive trial-and-error cycles.
>
> To our knowledge, no previous work has provided *such large-scale empirical evidence* for modern discovery tasks—DeepScientist is the first.

---

> ### Author Response · Authors · 2025-11-17
> **Response - 2**
>
> **2. The Standpoint**
>
> When experimentation is extremely expensive, we must fundamentally rethink the assumptions used by works such as FunSearch and AlphaEvolve.
>
> DeepScientist repeatedly emphasizes a conceptual stance *opposite* to these prior works:
>
> **In modern high-cost scientific discovery, the core question is no longer “Can AI generate new ideas?” but “How can we efficiently \*guide\* exploration under a severely constrained experiment budget?”**
>
> Thus, we argue for **efficient, goal-directed exploration rather than brute-force trial-and-error**.
>
> For example:
>
> > “Therefore, the central question of the field is no longer ‘Can AI innovate?’, but rather ‘How can we efficiently guide its powerful, yet highly dissipative, exploratory process to maximize scientific return?’
> >  We hope this work can inspire the research community to develop AI Scientist systems with greater exploration efficiency...”
>
>
> We further highlight the limitations of prior methods:
>
> > “Scientific discovery differs from... early-stage ML... or algorithm discovery... Each exploratory step requires immense resources... This necessitates an efficient exploration strategy rather than brute-force search. To address this, DeepScientist adopts an iterative Bayesian Optimization loop.”
>
> And from experimental evidence:
>
> > “…for high-cost endeavors like pre-training foundation models or pharmaceutical synthesis, the low success rate makes such an approach currently impractical.”
>
>
> Our experiments show that AlphaEvolve/FunSearch-style unbounded trial-and-error **dramatically increases cost, reduces success rate**, and is unsuitable for modern scientific discovery.
>
> ------
>
> **3. The Method**
>
> To address these limitations, DeepScientist introduces a *hierarchical Bayesian Optimization framework* with two key mechanisms:
>
> 1. **Three-level hierarchical Bayesian Optimization**
>     Only ideas that demonstrate value progress to the next level—unlike prior work where *all* ideas require full validation.
> 2. **Using failures as learning signals**
>     Unlike AlphaEvolve, which only uses successful discoveries, DeepScientist treats failure as experience, helping guide exploration in an enormous search space and avoid repeated mistakes.
>
> The goal is simple:
>  **maximize exploration efficiency under highly limited resources.**
>
> Across all modern tasks, DeepScientist ran *continuously for half a month to a month on 8×H800 servers* per task—far larger than any prior system on similar tasks. This allowed us to reveal how *depth* of exploration affects performance in challenging discovery settings.
>
> ------
>
> **Based on this system and large-scale empirical study, we believe DeepScientist provides at least five novel and previously unobserved empirical findings for automated modern scientific discovery:**
>
> **Insight 1 — LLMs can progressively surpass human SoTA on frontier tasks**
>  Tasks that normally take human experts months can be improved upon by LLMs via iterative scientific innovation.
>
> **Insight 2 — Quantification of the end-to-end discovery funnel**
>  Only **1–3% of hypotheses** meaningfully progress—shifting the research bottleneck away from idea generation toward efficient validation.
>
> **Insight 3 — First evidence of “The Ideation–Execution Gap” in autonomous AI-run experiments**
>  We provide **~5,000** real execution logs showing that AI-generated ideas often fail during AI implementation—confirming findings that earlier required hiring 43 humans [6].
>
> **Insight 4 — Scaling laws in scientific discovery**
>  Empirically demonstrating near-linear scaling between computational resources and the rate of *progress findings*, supporting long-standing hypotheses [7].
>
> **Insight 5 — A dataset of ~5,000 AI-driven scientific research trajectories**
>  Including successes and failures—valuable for analyzing AI scientists or training them further. Previously, the largest comparable dataset (PaperBench [14]) only contained 20 human-annotated papers.

---

> ### Author Response · Authors · 2025-11-17
> **Response - 3**
>
> > The experiments and claims are not persuading enough. It claims to "outperforms" three human sota results. But the first experiment, "Agents Failure Attribution", the selected baseline is from [4]. However, it seems that [4] is the first paper which bring about this research question of "Agents Failure Attribution" and propose the first benchmark. It does not make [4] a good and valid "state-of-the-art human results".
>
>
> **You may have misunderstood this point: work [5] is a baseline, while work [4] represents the current state of the art, and DeepScientist had already surpassed it before it was publicly released.**
>
> In fact, for the *Agents Failure Attribution* task, **[5] is the first work that introduced the benchmark** (16.67%, 2025.5), while **[4] was released later and represented the then-state-of-the-art human approach** (37.3%, 2025.9).
>
> Our A2P method was completed in 2025.7 with a performance of 47.46%, **which predates the public release of [4] and achieves a substantially higher score**.
>
> To avoid confusion, in the revised manuscript:
>
> - we label **[5]** as *the first benchmark* in the main text,
> - we label **[4]** as *the later state-of-the-art method*,
> - and we clarify that A2P was independently developed *before* [4] became publicly available.
>
> It is also worth noting that, after confirming compatibility with submission requirements, we submitted the A2P paper to an IEEE conference. It passed peer review and received an acceptance notification in September 2025.
>
> | Method              | Performance | Time       |
> | ------------------- | ----------- | ---------- |
> | [5] All-at-Once     | 16.67%      | 2025.5     |
> | [4] AgenTracer      | 37.3%       | 2025.9     |
> | [DeepScientist] A2P | **47.46%**  | **2025.7** |
>
> ------
>
> > For the second experiment (line 289–302), the paper does not mention the previous state-of-the-art. Overall, I believe more discussion of the three evaluated tasks would benefit the paper—especially clarifying the prior research landscape of these domains—so that readers can better understand the discoveries made by the proposed framework. In line 352, it states “The generated papers are available in Appendix D.”, but I could not find them in Appendix D. In line 870, there is a LaTeX reference error “Figure ??.c.”.
>
> Your suggestions are very reasonable. We have already submitted a revised version addressing these issues:
>
> - We added a clearer description of prior state-of-the-art work for the second experiment.
> - We reorganized the appendix so that all generated papers can be easily found in **Appendix F**, without being interrupted by subsequent figures.
> - We fixed the LaTeX reference error at line 870 (“Figure ??.c”).
>
> All these updates are now reflected in the newly submitted manuscript.

---

> ### Author Response · Authors · 2025-11-17
> **Response - 4**
>
> **Finally, we fully understand your emphasis on “new knowledge,” but we find it difficult to agree with the assertion that “this paper is more of an engineering implementation than a research work with new insights.” This underestimates DeepScientist’s empirical scientific contribution.**
>
> In empirical sciences like machine learning, groundbreaking insights do not arise solely from theoretical abstraction—they often emerge from **building large-scale systems that push boundaries**.
>  Such systems allow the community to observe **emergent behaviors**, quantify **unknown bottlenecks**, and establish **new scaling laws**—all of which constitute “new knowledge.”
>
> Consider examples:
>
> - **AlexNet [14]** was the first to leverage GPUs at scale for CNN training, which opened a new era for neural networks and gave AI researchers renewed hope.
> - Compared with GPT-2 [9], GPT-3 [8] did not appear to introduce new training algorithms or modify the fundamental language-modeling approach. To some, it may have looked like “just scaling.” But as researchers in this field, we should not overlook the insights and impact it brought. GPT-3 exhibited numerous emergent abilities [10]—such as few-shot learning—that had never been observed before. It showed that continued scaling is essential for understanding both the limits and the potential of AI systems. This line of progress directly paved the way for today’s models like ChatGPT, GPT-5, and Claude.
> - And even after GPT-3, Works like **OPT [11]** and **BLOOM [12]** had enormous value despite not introducing new algorithms.
>
> As stated in the ICLR Reviewer Guide:
>
> > “Submissions bring value when they convincingly demonstrate new, relevant, impactful knowledge (empirical, theoretical, practical...). This does not necessarily require state-of-the-art results.”
>
>
> Similarly, the ICLR community itself embraces openness. At ICLR 2023, **GLM-130B** [13] was accepted with a review score of **8888**, and the meta-reviewer noted: *“All reviews liked the paper, which includes comprehensive experiments and details. The reviewers also appreciated the substantial engineering effort behind this work. Major plus for open source!”*
>
>
> ------
>
> DeepScientist introduces a range of novel findings, unprecedented Insight, a principled standpoint, and concrete methodological solutions tailored to modern scientific discovery tasks with extremely high experimental cost—supported by open, reproducible large-scale evidence.
>
> Given this:
>
> - **If you find that the work offers insights comparable in significance to AlphaEvolve, or that it merits an oral presentation at ICLR 2026, a score of 10 (Oral) may be appropriate.**
> - **If you feel that these conclusions do not warrant a full oral presentation and are better communicated to interested attendees in a poster format, then a score of 8 (Poster) may be an appropriate assessment.**
> - **If you still believe it contains no new knowledge and merely repeats existing Task, Standpoint and Methods, you may keep the current score of 2 (Reject).**
>
> In any case, this work goes far beyond an “engineering implementation.” I am strongly confident that it represents a large-scale scientific experiment that yields at least six entirely new empirical insights into **automated modern scientific discovery**. It not only demonstrates what AI can do (progressively surpassing human SOTA), but more importantly reveals how AI does it (avoiding infinite trial-and-error and efficiently allocating resources), how well it does (only ~3% of hypotheses succeed), why it fails (the Ideation–Execution Gap and the fact that ~60% of failures stem from implementation errors), and how it can do more (scaling laws).
>
> **These findings constitute indispensable new knowledge for the emerging field of automated science.**
>
>
>
> ------
>
> **References**
>
> [1] FunSearch: Making new discoveries in mathematical sciences using Large Language Models.
>
> [2] AlphaEvolve: A Gemini-powered coding agent for designing advanced algorithms.
>
> [3] LLM-SR: Scientific Equation Discovery via Programming with Large Language Models.
>
> [4] AgenTracer: Who Is Inducing Failure in the LLM Agentic Systems?
>
> [5] Which Agent Causes Task Failures and When? On Automated Failure Attribution of LLM Multi-Agent Systems
>
> [6] The Ideation-Execution Gap: Execution Outcomes of LLM-Generated versus Human Research Ideas
>
> [7] Scaling Laws in Scientific Discovery with AI and Robot Scientists
>
> [8] Language Models are Few-Shot Learners
>
> [9] Language Models are Unsupervised Multitask Learners
>
> [10] Emergent Abilities of Large Language Models
>
> [11] OPT: Open Pre-trained Transformer Language Models
>
> [12] BLOOM: A 176B-Parameter Open-Access Multilingual Language Model
>
> [13] GLM-130B: An Open Bilingual Pre-trained Model
>
> [14] ImageNet Classification with Deep Convolutional Neural Networks
>
> [15] AlgoTune: Can Language Models Speed Up General-Purpose Numerical Programs?

---

### Author Response · Authors · 2025-11-17
**Overall Response to All Reviewers - 1**

We sincerely thank the four reviewers (**Reviewer DLyc, Reviewer oPiU, Reviewer pBcW, and Reviewer rLBb**) for investing significant time and effort to provide detailed, thoughtful, and constructive feedback. Your comments have helped us better understand the strengths of our work, clarify possible misunderstandings, and identify areas that required further explanation or experimentation. We have revised the paper and incorporated additional experiments accordingly.

------

## **Highlights Identified by the Reviewers**

**• Framework and Overall Goal**

- *Reviewer rLBb* affirmed the soundness of building an LLM-based framework that covers the full cycle of scientific discovery—from hypothesis generation and experimentation to paper writing—and noted that the system may serve as a useful reference for future research.
- *Reviewer oPiU* emphasized that automated scientific discovery is an important direction with long-term impact, and considered our attempt to “compete with human SoTA on real frontier tasks” both meaningful and well-motivated.

**• Methodological and Conceptual Innovations**

- *Reviewer pBcW* highlighted that our formulation of the scientific discovery process as a **goal-driven Bayesian Optimization problem** offers a new perspective compared to agent/prompt-only approaches.
- *Reviewer pBcW* also recognized that DeepScientist enables **multi-round, long-horizon, incremental deep exploration**, whereas prior systems typically focus on single-step idea generation.

**• Experimental Scale and Analysis**

- *Reviewer DLyc* and *Reviewer pBcW* acknowledged that our study—spanning three frontier tasks with ~20,000 GPU hours, over 5,000 candidate hypotheses, and 1,100+ experimental validations, complemented by human expert evaluation—constitutes a **reasonably comprehensive and persuasive empirical investigation**.
- *Reviewer DLyc* commended the clarity and organization of our system visualizations and analyses.

------

## **A Common Misunderstanding and Our Clarification**

We noticed a recurring concern mentioned by multiple reviewers, expressed in different forms:

> *“Is DeepScientist essentially an engineering-scaled version of existing AI Scientist or closed-loop systems? Does its novelty mainly stem from stronger LLMs or selecting three tasks where surpassing human SoTA is relatively feasible?”*

We clarify the following:

**1. Nature of the Tasks: From Low-Cost Trial-and-Error to High-Cost Modern Problems**
 Prior work often investigates algorithmic or symbolic domains where trial-and-error incurs very low cost (seconds or minutes).
 In contrast, we intentionally selected **three modern, high-stakes, high-cost frontier tasks**—Agent Failure Attribution, LLM Inference Acceleration, and AI Text Detection—each actively studied by the human research community.
 On these tasks, **infinite trial-and-error is practically impossible**, which directly motivated our design of a system centered on **efficient exploration under extremely limited budgets**.

**2. What Truly Matters Is Not Only Surpassing SoTA but Revealing Empirical Regularities in Modern Scientific Discovery**
 Beyond demonstrating multiple SoTA improvements, a key contribution of our work is the first systematic quantification of:

- the *“funnel-shaped”* success rate of autonomous exploration (only ~1–3% of hypotheses lead to actual progress),
- the role of failure memory and hierarchical Bayesian optimization under high-cost settings,
- an *approximate linear scaling law* between discovery throughput and parallel GPU resources,
- and the strengths and biases of LLMs across evaluation, filtering, and self-review stages.

These empirical observations are difficult to obtain from prior automatic discovery systems focused on simpler tasks (e.g., symbolic regression, equation discovery, or basic algorithm synthesis). We believe such findings constitute the core novelty of our work.

------

---

> ### Author Response · Authors · 2025-11-17
> **Overall Response to All Reviewers - 2**
>
> ## **Major Concerns and Our Consolidated Responses**
>
> **1. \*Is the evaluation scope too narrow? Does selecting three tasks risk cherry-picking?\* (Reviewer pBcW)**
>
> > Concern: Surpassing human SoTA on only three AI tasks might not sufficiently demonstrate generality.
>
> **Response:**
>  During the rebuttal period, we developed **Micronano-DeepScientist**, a lightweight version preserving hierarchical search and failure memory while removing the most expensive stages (literature search, experimental analysis, and paper writing). This reduces per-task cost to roughly 1/1000 of the full system.
>
> On the AlgoTune meta-benchmark of **154 algorithmic tasks across mathematics, physics, and computer science**:
>
> - The system achieved **better-than-human-baseline solutions on 120 tasks (77.9%)**,
> - with an average **16.6× speedup**.
>
> These results, added to the appendix, demonstrate that our framework generalizes to a broader task distribution and does not rely on cherry-picked scenarios.
>
> ------
>
> **2. \*Is “surpassing human SoTA” too narrow an evaluation metric?\* (Reviewer pBcW)**
>
> > Concern: Scientific progress is continuous, and replacing SoTA alone may not capture research efficiency.
>
> **Response:**
>  We added a case study comparing system-level research efficiency with real human teams:
>
> - From discussions with authors of two human SoTA papers for the AI text detection task, we estimated that a typical human team invests **~6 months and ~1500 GPU·hours** per major attempt.
> - DeepScientist, in contrast, completed **three successive steps of frontier improvement** (T-Detect → TDT → PA-TDT) within **~14 days and ~900 GPU·hours**, exploring **2400+ hypotheses** and running **600+ experiments**.
>
> Although not a tightly controlled study, this provides meaningful qualitative evidence regarding *research throughput, cost efficiency, and discovery density*, addressing the concern that SoTA replacement alone is insufficient.
>
> ------
>
> **3. \*What differentiates DeepScientist from prior AI Scientist / Zochi-style systems?\*
>  (Reviewer rLBb, Reviewer oPiU)**
>
> We expanded Sections 2 and 3 to detail how prior methods typically rely on:
>
> - one-shot pipeline execution (Idea → Experiment → Paper), or
> - repetitive trial-and-error on a single idea.
>
> Such approaches struggle under **high-cost exploration**.
>  DeepScientist instead employs:
>
> - **continuous iterative loops**,
> - **hierarchical Bayesian optimization**, and
> - **cumulative failure memory**,
>
> ensuring that only the most promising hypotheses advance into expensive experimental stages.
>  We also added a structured comparison of workflow shape, failure handling, exploration method, task suitability, and representative outcomes.
>
> ------
>
> **4. \*Are the experimental settings and evaluations sufficiently robust?\*
>  (Reviewer oPiU, Reviewer DLyc)**
>
> - To address concerns about reliance on a single automatic reviewer, we added results using multiple reviewers (Gemini-2.5-Pro, GPT-4o, GPT-5, CycleReviewer, DeepReviewer-14B). Across all reviewers, DeepScientist consistently outperformed the baselines.
> - To address potential bias in human evaluation, we clarified that the three experts were not informed of the paper sources.
> - We also included a real-world validation: submitting one of the system-generated papers (A2P) in a double-blind but formally acceptable manner (with the approval of the conference program chair) to an IEEE venue, where it was successfully accepted.
>
> ------
>
> **5. \*Task descriptions, baselines, and hyperparameter details require greater clarity\*
>  (Reviewer DLyc, Reviewer oPiU, Reviewer rLBb)**
>
> We have added:
>
> - clearer background, human SoTA method descriptions, and limitations for each of the three tasks in Section 4.1,
> - detailed explanations for all baselines in Table 2,
> - clarification that acquisition-function hyperparameters in Eq.(1) were **fixed to (1,1,1)** for all experiments with no task-specific tuning,
> - and details on how the Findings Memory is constructed entirely by the system without manual annotations.

---

> ### Author Response · Authors · 2025-11-17
> **Overall Response to All Reviewers - 3**
>
> ## **Summary of  Revisions in the Updated Manuscript**
>
> **1. Task and Baseline Clarifications**
>
> - Expanded Section 4.1 with task background, human SoTA details, and limitation analyses.
> - Added descriptions and configuration details for all baselines in Table 2.
> - Clarified the benchmark timeline and SoTA status for Agent Failure Attribution.
>
> **2. Method and Related Work Enhancements**
>
> - Strengthened comparisons with AI Scientist, Zochi, and “AlphaGo Moment for Architecture Discovery”–style systems.
> - Added a subsection discussing methodological differences—continuous iteration, hierarchical BO, and failure memory.
>
> **3. System Setup and Implementation Details**
>
> - Documented acquisition-function hyperparameters and their invariance across all tasks.
> - Added explanations on how 21 progress ideas translated into 5 system-generated papers.
> - Included small-scale ablation studies on Planner (Gemini) and Executor (Claude) strength.
> - Introduced a new CLI toolkit that will be open-sourced together with DeepScientist, enabling fully autonomous long-horizon exploration on any server environment. This toolkit allows the community to directly run DeepScientist as a persistent, server-controlling research agent, thereby enhancing the efficiency and throughput of real-world Machine Learning community scientific exploration.
>
> **4. Expanded Evaluation and Experiments**
>
> - Added **Micronano-DeepScientist** and large-scale AlgoTune results in Appendix E.1.
> - Added multi-reviewer evaluation in Appendix E.2.
> - Added a case study on research efficiency and cost analysis in Appendix E.3.
>
> **5. Presentation and Formatting Fixes**
>
> - Reorganized automatically generated papers into cleaner appendix sections.
> - Fixed LaTeX citation issues (e.g., “Figure ??.c”).
> - Improved figure descriptions and narrative flow.
>
> **6. Trajectory, Failures, and Scaling Analysis**
>
> - Added a funnel analysis of the autonomous exploration process (≈5,000 ideas → ≈1,100 tested → 21 Progress Findings → 5 final papers) and clarified that the filtering is fully automated.
> - Analyzed failure modes and the division of labor between planner and executor, showing that ~60% of failed trials stem from implementation errors, while the planner governs how far the system can advance under a fixed budget.
> - Refined the scaling-law analysis, highlighting near-linear gains in Progress Findings with more GPUs and discussing how knowledge sharing underlies both parallel (hardware) scaling and longer-horizon serial (time) scaling.
> ------
>
> We once again thank all reviewers for their thoughtful evaluations and constructive suggestions. We hope the revised version presents a clearer picture of DeepScientist’s methodological distinctions and the empirical insights it contributes to automated, high-cost modern scientific discovery. We respectfully invite you to reassess the paper in light of these revisions.
>
> ---
>
> The authors of "DeepScientist: Advancing Frontier-Pushing Scientific Findings Progressively"

---

### Meta-Review · Area_Chair_K38p · 2026-01-07

**Summary:**

This submission proposes DeepScientist, an autonomous agent for scientific discovery. Reviewers agreed the topic is timely and the overall direction is potentially high impact, but had significant concerns about

1. Generality beyond a small, hand-selected set of tasks,

2.  Robustness of the evaluation protocol,

3. Clarity of positioning and novelty relative to closely related systems.

The initial reviewer scores were mixed and leaned negative overall: 6 (DLyc), 4 (oPiU), 4 (pBcW), 2 (rLBb).

The main concerns of the reviewers are as follows

1. Reviewer pBcW argued that evaluating on only a few manually selected tasks is not a sufficient.

2. Reviewer pBcW emphasized that beating a prior SOTA is not the target, and requested an analysis of time and cost efficiency compared to humans. Reviewer rLBb also has concerns about the baselines compared against.

3. Reviewer oPiU raised concerns that key claims rely on a single reviewer.

4. Reviewer oPiU requested a more detailed comparison to related systems (AI Scientist, Zochi, and similar pipelines). Reviewer rLBb also concerns about novelty in relation to prior work.

**Reviewer Concerns:**

The authors’ rebuttal is unusually extensive and adds substantial new experiments and clarifications. I think the rebuttal has throughly addressed the reviewer concerns. The rebuttal added

1. Evaluated their smaller model on 154 new tasks and reported success on 120 of them.

2. Contacted authors to compare time taken by their system versus actual humans. Additionally clarified concerns about the baselines.

3. Added multi reviewer evaluation.

In all cases the obtaining favorable results.

**Reviewer Scores:**

Reviewer pBcW (initial 4): likely to increase to ~6 given the added 154-task breadth study and additional discussion of research efficiency, which directly addresses their central concerns.

Reviewer oPiU (initial 4): likely to increase to –6 due to the addition of the multi-reviewer setting

Reviewer DLyc (initial 6): likely to remain supportive

Reviewer rLBb (initial 2): may move slightly upward but most likely to remain a reject.

The rebuttal posted by the authors is quite comprehensive and I would classify it as a major revision to the paper if this were a journal. As such I believe it would require a second round of evaluations by the reviewers as the contributions are substantially different from the original paper. However, given the circumstances this is not possible.

However, to my estimating the concerns of the reviewers have been addressed and the paper should be accepted.

---

### Decision · Program_Chairs · 2026-01-26

Accept (Poster)